# Identifiability Guarantees for Causal Disentanglement from Soft Interventions

**Jiaqi Zhang**
LIDS, MIT
Broad Institute of MIT and Harvard

**Kristjan Greenewald**
MIT-IBM Watson AI Lab
IBM Research

**Chandler Squires**
LIDS, MIT
Broad Institute of MIT and Harvard

**Akash Srivastava**
MIT-IBM Watson AI Lab
IBM Research

**Karthikeyan Shanmugam**[*]
IBM Research

**Caroline Uhler**
LIDS, MIT
Broad Institute of MIT and Harvard

## Abstract

Causal disentanglement aims to uncover a representation of data using latent variables that are interrelated through a causal model. Such a representation is identifiable if the latent model that explains the data is unique. In this paper, we focus on the scenario where unpaired observational and interventional data are available, with each intervention changing the mechanism of a latent variable. When the causal variables are fully observed, statistically consistent algorithms have been developed to identify the causal model under faithfulness assumptions. We here show that identifiability can still be achieved with unobserved causal variables, given a generalized notion of faithfulness. Our results guarantee that we can recover the latent causal model up to an equivalence class and predict the effect of unseen combinations of interventions, in the limit of infinite data. We implement our causal disentanglement framework by developing an autoencoding variational Bayes algorithm and apply it to the problem of predicting combinatorial perturbation effects in genomics.

## 1 Introduction

The discovery of causal structure from observational and interventional data is important in many fields including statistics, biology, sociology, and economics [42, 17]. Directed acyclic graph (DAG) models enable scientists to reason about causal questions, e.g., predicting the effects of interventions or determining counterfactuals [46]. Traditional causal structure learning has considered the setting where the causal variables are observed [21]. While sufficient in many applications, this restriction is limiting in most regimes where the available datasets are either perceptual (e.g., images) or very-high dimensional (e.g., the expression of $> 20k$ human genes). In an imaging dataset, learning a causal graph on the pixels themselves would not only be difficult since there is no common coordinate system across images (pixel $i$ in one image may have no relationship with pixel $i$ in another image) but of questionable utility due to the relative meaninglessness of interventions on individual pixels. Similar problems are also present when working with very high-dimensional data. For example,

---

[*]Currently at Google Research. Contributions to this work were made when affiliated with IBM research.

37th Conference on Neural Information Processing Systems (NeurIPS 2023).

in a gene-expression dataset, subsets of genes (e.g. belonging to the same pathway) may function together to induce other variables and should therefore be aggregated into one causal variable.

These issues mean that the causal variables need to be learned, instead of taken for granted. The recent emerging field of causal disentanglement [9, 65, 30] seeks to remedy these issues by recovering a causal representation in *latent* space, i.e., a small number of variables $U$ that are mapped to the observed samples in the ambient space via some mixing map $f$. This framework holds the potential to learn more semantically meaningful latent factors than current approaches, in particular factors that correspond to interventions of interest to modelers. Returning to the image and the genomic examples, latent factors could, for example, be abstract functions of pixels (corresponding to objects) or groups of genes (corresponding to pathways).

Despite a recent flurry of interest, causal disentanglement remains challenging. First, it inherits the difficulties of causal structure learning where the number of causal DAGs grows super-exponentially in dimension. Moreover, since we only observe the variables after the unknown mixing function but never the latent variables, it is generally impossible to recover the latent causal representations with only observational data. Under the strong assumption that the causal DAG is the empty graph, such unidentifiability from observational data has been discussed in previous disentanglement works [28].

However, recent advances in many applications enable access to interventional data. For example, in genomics, researchers can perturb single or multiple genes through CRISPR experiments [13]. Such interventional data can be used to identify the causal variables and learn their causal relationships. When dealing with such data, it is important to note that single-cell RNA sequencing and other biological assays often destroy cells in the measurement process. Thus, the available interventional data is *unpaired*: for each cell, one only obtains a measurement under a single intervention.

In this work, we establish identifiability for *soft* interventions on general structural causal models (SCMs), when the latent causal variables are observed through a class of (potentially non-linear) polynomial mixing functions proposed by [3]. Prior works [57, 20, 67] show that the causal model can be identified under *faithfulness* assumptions, when all the causal variables are observed. We here demonstrate that idenfiability can still be achieved when the causal variables are *unobserved* under a generalized notion of faithfulness. The identifiability is up to an equivalence class and guarantees that we can predict the effect of unseen combinations of interventions, in the limit of infinite data. It then remains to design an algorithmic approach to estimate the latent causal representation from data. We propose an approach based on autoencoding variational Bayes [29], where the decoder is composed of a deep SCM (DSCM) [45] followed by a deep mixing function. Finally, we apply our approach to a real-world genomics dataset to find genetic programs and predict the effect of unseen combinations of genetic perturbations.

## 1.1 Related Work

**Identifiable Representation Learning.** The identifiability of latent representations from observed data has been a subject of ongoing study. Common assumptions are that the latent variables are independent [11], are conditionally independent given some observed variable [23, 28], or follow a known distribution [71]. In contrast, we do not make any independence assumptions on the latent variables or assume we know their distribution. Instead, we assume that the variables are related via a causal DAG model, and we use data from interventions in this model to identify the representation.

**Causal Structure Learning.** The recovery of a causal DAG from data is well-studied for the setting where the causal representation is directly observed [21]. Methods for this task take a variety of approaches, including exact search [12] and greedy search [10] to maximize a score such as the posterior likelihood of the DAG, or an approximation thereof. These scores can be generalized to incorporate interventional data [62, 67, 33], and methods can often be naturally extended by considering an augmented search space [43]. Indeed, interventional data is generally necessary for identifiability without further assumptions on the functions relating variables [54].

**Causal Disentanglement.** The task of identifying a causal DAG over latent causal variables is less well-studied, but has been the focus of much recent work [9, 65, 30]. These works largely do not consider interventions, and thus require restrictions on functional forms as well as structural assumptions on the map from latent to observed variables. Among works that do not restrict the map, [2] and [6] assume access to *paired* counterfactual data. In contrast, we consider only *unpaired* data, which is more common in applications such as biology [55]. Unpaired interventional data

is considered by [3], [53], and as a special case of [37]. These works do not impose structural restrictions on the map from latent to observed variables but assume functional forms of the map, such as linear or polynomial. Our work builds on and complements these results by providing identifiability for *soft* interventions and by offering a learning algorithm based on variational Bayes. We remark here that the task of causal disentanglement is sometimes called *causal representation learning* in literature. We adopted the term causal disentanglement mainly following [26], as causal representation learning also includes methods such as Invariant Risk Minimization (IRM) [4] which do not completely learn latent variables. We discuss contemporaneous related work in Appendix I.

## 2 Problem Setup

We now formally introduce the causal disentanglement problem of identifying latent causal variables and causal structure between these variables. We consider the *observed* variables $X = (X_1, ..., X_n)$ as being generated from *latent* variables $U = (U_1, ..., U_p)$ through an unknown deterministic (potentially non-linear) mixing function $f$. In the observational setting, the latent variables $U$ follow a joint distribution $\mathbb{P}_U$ that factorizes according to an unknown directed acyclic graph (DAG) $\mathcal{G}$ with nodes $[p] = \{1, ..., p\}$. Concisely, we have the following data-generating process:

$$X = f(U), \quad U \sim \mathbb{P}_U = \prod_{i=1}^{p} \mathbb{P}(U_i \mid U_{\mathrm{pa}_{\mathcal{G}}(i)}), \tag{1}$$

where $\mathrm{pa}_{\mathcal{G}}(i) = \{j \in [p] : j \to i\}$ denotes the parents of $i$ in $\mathcal{G}$. We also use $\mathrm{ch}_{\mathcal{G}}(i)$, $\mathrm{de}_{\mathcal{G}}(i)$ and $\mathrm{an}_{\mathcal{G}}(i)$ to denote the children, descendants and ancestors of $i$ in $\mathcal{G}$. Let $\mathbb{P}_X$ denote the induced distribution over $X$.

We consider atomic (i.e., single-node) interventions on the latent variables. While our main focus is on general types of soft interventions, our proof also applies to hard interventions. In particular, an *intervention $I$* with target $T_{\mathcal{G}}(I) = i \in [p]$ modifies the joint distribution $\mathbb{P}_U$ by changing the conditional distribution $\mathbb{P}(U_i \mid U_{\mathrm{pa}_{\mathcal{G}}(i)})$. A *hard* intervention sets the conditional distribution as $\mathbb{P}^I(U_i)$, removing the dependency of $U_i$ on $U_{\mathrm{pa}_{\mathcal{G}}(i)}$, whereas a *soft* intervention is allowed to preserve this dependency but changes the mechanism into $\mathbb{P}^I(U_i \mid U_{\mathrm{pa}_{\mathcal{G}}(i)})$. An example of a soft intervention is as a shift intervention [50, 69], which modifies the conditional distribution as $\mathbb{P}^I(U_i = u + a_i \mid U_{\mathrm{pa}_{\mathcal{G}}(i)}) = \mathbb{P}(U_i =$

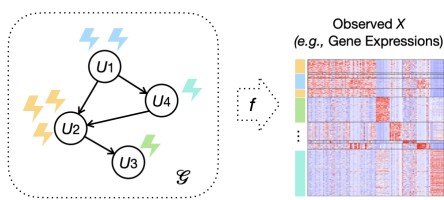

Figure 1: Example of the data-generating process, where observed gene expressions $X = f(U)$ and the distribution of $U$ factorizes with respect to an unknown DAG $\mathcal{G}$.

$u \mid U_{\mathrm{pa}_{\mathcal{G}}(i)})$ for some shift value $a_i$. In the following, we will use $\mathbb{P}_U^I = \prod_{i=1}^{p} \mathbb{P}^I(U_i \mid U_{\mathrm{pa}_{\mathcal{G}}(i)})$ to denote the interventional distribution, where $\mathbb{P}^I(U_j \mid U_{\mathrm{pa}_{\mathcal{G}}(j)}) = \mathbb{P}(U_j \mid U_{\mathrm{pa}_{\mathcal{G}}(j)})$ for $j \neq T_{\mathcal{G}}(I)$. We denote the induced distribution over $X$ by $\mathbb{P}_X^I$. In cases where the referred random variable is clear from the context, we abbreviate the subscript and use $\mathbb{P}^I$ instead.

We consider the setting where we have unpaired data from observational and interventional distributions, i.e., $\mathcal{D}, \mathcal{D}^{I_1}, ..., \mathcal{D}^{I_K}$. Here, $\mathcal{D}$ denotes samples of $X = f(U)$ where $U \sim \mathbb{P}_U$; $\mathcal{D}^{I_k}$ denotes samples of $X$ where $U \sim \mathbb{P}_U^{I_k}$. We focus on the scenario where we have *at least* one intervention per latent node. In the worst case, one intervention per node is necessary for identifiability in linear SCMs [53]. We note that having *at least* one intervention per latent node is a strict generalization of having *exactly* one intervention per latent node, since we assume no knowledge of which interventions among $I_1, ..., I_K$ target the same node. Throughout the paper, we assume latent variables $U$ are unobserved and their dimension $p$, the DAG $\mathcal{G}$, and the interventional targets of $I_1, ..., I_K$ are unknown. The goal is to identify these given samples of $X$ in $\mathcal{D}, \mathcal{D}^{I_1}, ..., \mathcal{D}^{I_K}$.

## 3 Equivalence Class for Causal Disentanglement

In this section, we characterize the equivalence class for causal disentanglement, i.e., the class of latent models that can generate the same observed samples of $X$ in $\mathcal{D}, \mathcal{D}^{I_1}, ..., \mathcal{D}^{I_K}$. Since we only have access to this data, the latent model can only be identified up to this equivalence class.

First, note that the data-generation process is agnostic to the re-indexing of latent variables, provided that we change the mixing function to reflect such re-indexing. More precisely, consider an arbitrary permutation $\pi$ of $[p]$. Denote $U_\pi = (U_{\pi(1)}, ..., U_{\pi(p)})$ and $f_\pi$ as the mixing function such that $f_\pi(U_\pi) = f(U)$. We define $\mathcal{G}_\pi$ as the DAG with nodes in $[p]$ and edges $i \to j$ if and only if $\pi(i) \to \pi(j) \in \mathcal{G}$. Then the following data-generating process,

$$X = f_\pi(U_\pi), \quad U_\pi \sim \mathbb{P}_{U_\pi} = \prod_{i=1}^p \mathbb{P}\big((U_\pi)_i \mid (U_\pi)_{\mathrm{pa}_{\mathcal{G}_\pi}(i)}\big),$$

satisfies $X = f(U)$. The same argument holds when $U$ is generated from an interventional distribution $\mathbb{P}^I$, where this process generates the same $X$ when $U_\pi$ is sampled from $\mathbb{P}_{U_\pi}^{I_\pi}$. Here $I_\pi$ is such that $T_{\mathcal{G}_\pi}(I_\pi) = \pi^{-1}(T_{\mathcal{G}}(I))$ and the mechanism $\mathbb{P}^{I_\pi}\big((U_\pi)_i \mid (U_\pi)_{\mathrm{pa}_{\mathcal{G}_\pi}(i)}\big) = \mathbb{P}^I\big(U_{\pi(i)} \mid U_{\mathrm{pa}_{\mathcal{G}}(\pi(i))}\big)$.

We would also observe the same data if each $U_i$ is affinely transformed into $\lambda_i U_i + b_i$ for constants $\lambda_i \neq 0$ and $b_i$ and the mixing function is rescaled element-wise to accommodate this transformation. To account for these two types of equivalences, we define the following notion of causal disentanglement (CD) equivalence class.

**Definition 1** (CD-Equivalence). *Two sets of variables, $\langle U, \mathcal{G}, I_1, ..., I_K \rangle$ and $\langle \hat{U}, \hat{\mathcal{G}}, \hat{I}_1, ..., \hat{I}_K \rangle$ are* CD-equivalent *if and only if there exists a permutation $\pi$ of $[p]$, non-zero constants $\lambda_1, ..., \lambda_p \neq 0$, and $b_1, ..., b_p$ such that*

$$\hat{U}_i = \lambda_{\pi(i)} U_{\pi(i)} + b_{\pi(i)}, \ \forall i \in [p], \quad \hat{\mathcal{G}} = \mathcal{G}_\pi, \quad \text{and } \hat{I}_k = (I_k)_\pi, \ \forall k \in [K].$$

*The same definition applies to $\langle \mathcal{G}, I_1, ..., I_k \rangle$ and $\langle \hat{\mathcal{G}}, \hat{I}_1, ..., \hat{I}_k \rangle$, where we say they are* CD-equivalent *if and only if $\hat{\mathcal{G}} = \mathcal{G}_\pi$, and $\hat{I}_k = (I_k)_\pi$ for some permutation $\pi$.*

For simplicity, we refrain from talking about transformations on the mixing function $f$ and mechanisms of latent variables. These can be obtained once $U, \mathcal{G}, T_{\mathcal{G}}(I_1), ..., T_{\mathcal{G}}(I_K)$ are identified. In particular, $f$ is the map from $U$ to the observed $X$; and the joint distribution $\mathbb{P}_U$ (and $\mathbb{P}_U^{I_k}$) can be decomposed with respect to $\mathcal{G}$ to obtain the mechanisms $\mathbb{P}_U(U_i \mid U_{\mathrm{pa}_{\mathcal{G}}(i)})$ (and $\mathbb{P}_U^{I_k}(U_i \mid U_{\mathrm{pa}_{\mathcal{G}}(i)})$).

## 4 Identifiability Results

In this section, we present our main results, namely the identifiability guarantees for causal disentanglement from soft interventions. For this discussion, we consider the *infinite-data* regime where enough samples are obtained to exactly determine the observational and interventional distributions $\mathbb{P}_X, \mathbb{P}_X^{I_1}, ..., \mathbb{P}_X^{I_K}$. Detailed proofs are deferred to Appendices A and B.

### 4.1 Preliminaries

Following [3], we pose assumptions on the support of $U$ and on the function class of the map $f$. Our support assumption is for example satisfied under the common additive Gaussian structural causal model [47], and our assumption on the function class is for example satisfied if $f$ is linear and injective (Lemma 2 in Appendix A), a setting considered in many identifiability works (e.g., [11, 1, 53]).

**Assumption 1.** *Let $U$ be a $p$-dimensional random vector. Following [3], we assume that the interior of the support of $\mathbb{P}_U$ is a non-empty subset of $\mathbb{R}^p$, and that $f$ is a full row rank polynomial.*[2]

Under this assumption, the authors in [3] showed that if $p$ is known, $U$ is identifiable up to a linear transformation. This remains true when $p$ is unknown, as summarized in the following lemma.

**Lemma 1.** *Under Assumption 1, we can identify the dimension $p$ of $U$ as well as its linear transformation $U\Lambda + b$ for some non-singular matrix $\Lambda$ and vector $b$. In fact, with observational data, we can only identify $U$ up to such linear transformations.*

Denote all pairs of $\mathbb{P}_U, f$ that satisfy this assumption as $\mathcal{F}_p$. The proof of this lemma is provided by solving the following constrained optimization problem:

$$\min_{(\mathbb{P}_{\hat{U}}, \hat{f}) \in \mathcal{F}_{\hat{p}}} \hat{p} \quad \text{subject to } \mathbb{P}_{\hat{f}(\hat{U})} = \mathbb{P}_X.$$

---

[2]There exists some integer $d$, a full row rank $H \in \mathbb{R}^{(p+...+p^d) \times n}$ and a vector $h \in \mathbb{R}^n$ such that $f(U) = (U, \bar{\otimes}U^2, ..., \bar{\otimes}U^d)H + h$, where $\bar{\otimes}U^k$ denotes the size-$p^k$ vector with degree-$k$ polynomials of $U$ as its entries.

In other words, let $\hat{p}$ be the smallest dimension such that there exists a pair of $\mathbb{P}_{\hat{U}}, \hat{f}$ in $\mathcal{F}_{\hat{p}}$ that generates the observational distribution $\mathbb{P}_X$. Then $p = \hat{p}$ and we recover the latent factors up to linear transformation. The intuition is that (1) the support with non-empty interior guarantees that we can identify $p$ by checking its geometric dimension, and (2) the full-rank polynomial assumption ensures that we search for $f$ (and consequently $U$) in a constrained subspace.

On the other hand, to show we cannot identify more than linear transformations, we construct a mixing function $\hat{f}$ for $\hat{U} := U\Lambda + b$ such that the induced distribution $\mathbb{P}_X$ is the same under both representations. This also means that we cannot identify the underlying DAG $\mathcal{G}$ up to any nontrivial equivalence class; we give an example showing that any causal DAG can explain the observational data in Appendix A. Next, we discuss how identifiability can be improved with interventional data.

## 4.2 Identifying ancestral relations

Lemma 1 guarantees identifiability up to linear transformations from solely observational data. This reduces the problem to the case where an unknown invertible linear mixing of the latent variables $X = f(U) = U\Lambda + b$ is observed. Without loss of generality, we thus work with this reduction for the remainder of the section.

When the causal variables are *fully observed*, we can identify causal relationships from the changes made by interventions [57]. In particular, an intervention on a node will not alter the marginals of its non-descendants as compared to the observational distribution, i.e., $\mathbb{P}(U_j) = \mathbb{P}^I(U_j)$ for $T_{\mathcal{G}}(I) = i$ and $j \notin \deg_{\mathcal{G}}(i)$. However, it is possible that $\mathbb{P}(U_j) = \mathbb{P}^I(U_j)$ for some $j \in \deg_{\mathcal{G}}(i)$ in degenerate cases where the change made by $U_i$ is canceled out on the path from $i$ to $j$. Hence, prior works[3] defined *influentiality* or *interventional faithfulness* [57, 66], which avoids such degenerate cases by assuming that intervening on a node will always change the marginals of all its descendants, i.e., $\mathbb{P}(U_j) \neq \mathbb{P}^I(U_j)$ for $j \in \deg_{\mathcal{G}}(i)$. Under this assumption, we can identify the descendants of an intervention target in $\mathcal{G}$, by testing if a node has a changed marginal interventional distribution.

However, if we only observe a linear mixing of the causal variables, interventional faithfulness is not enough to identify such ancestral relations. Consider the following example.

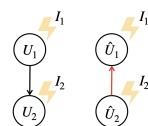

**Example 1.** *Let $\mathcal{G} = \{1 \to 2\}$ with $\mathbb{P}(U_1) = \mathcal{N}(0,1)$ and $\mathbb{P}(U_2 \,|\, U_1) = \mathcal{N}(U_1, 1)$. Suppose that $T_{\mathcal{G}}(I_1) = 1$, with $\mathbb{P}^{I_1}(U_1) = \mathcal{N}(1,1)$, and that $T_{\mathcal{G}}(I_2) = 2$, with $\mathbb{P}^{I_2}(U_2 \,|\, U_1) = \mathcal{N}(U_1 + 1, 1)$. Note that this model satisfies interventional faithfulness.*

Figure 2

*Let $f$ be the identity map, i.e., $X = U$. Consider latent variables $\hat{U} = (U_2, U_2 - U_1)$ and $\hat{f}(\hat{U}) = (\hat{U}_1 - \hat{U}_2, \hat{U}_1)$. Then $X = \hat{f}(\hat{U}) = f(U)$. However, we have $\hat{\mathcal{G}} = \{2 \to 1\}$ with $\mathbb{P}(\hat{U}_2) = \mathcal{N}(0,1)$ and $\mathbb{P}(\hat{U}_1 \,|\, \hat{U}_2) = \mathcal{N}(\hat{U}_2, 1)$, $T_{\hat{\mathcal{G}}}(I_1) = 1$ with $\mathbb{P}^{I_1}(\hat{U}_1 \,|\, \hat{U}_2) = \mathcal{N}(\hat{U}_2 + 1, 1)$, and $T_{\hat{\mathcal{G}}}(I_2) = 2$ with $\mathbb{P}^{I_2}(\hat{U}_2) = \mathcal{N}(1,1)$. We thus may reverse ancestral relations between the intervention targets, as illustrated in Figure 2.*

This example shows that the effect on $U_2$ from intervening on $U_1$ can be canceled out by linearly combining $U_2$ with $U_1$. In other words, intervening on $U_1$ does not change the marginal distribution of $U_2 - U_1$, even under interventional faithfulness. Thus, we need a stronger faithfulness assumption to account for the effect of linear mixing. In general, we want to avoid the case that the effect of an intervention on a downstream variable $U_j$ can be canceled out by combining $U_j$ linearly with other variables.

**Assumption 2.** *Intervention $I$ with target $i$ satisfies* linear interventional faithfulness *if for every $j \in \{i\} \cup \mathrm{ch}_{\mathcal{G}}(i)$ such that $\mathrm{pa}_{\mathcal{G}}(j) \cap \deg_{\mathcal{G}}(i) = \varnothing$, it holds that $\mathbb{P}(U_j + U_S C^\top) \neq \mathbb{P}^I(U_j + U_S C^\top)$ for all constant vectors $C \in \mathbb{R}^{|S|}$, where $S = [p] \setminus (\{j\} \cup \deg_{\mathcal{G}}(i))$.*

This assumption ensures that an intervention on $U_i$ not only affects its children, but that the effect remains even when we take a linear combination of a child with certain other variables. Note that the condition need only hold for the most upstream children of $U_i$, which may be arbitrarily smaller than the set of all children of $U_i$. To illustrate this assumption, we give a simple example on a 2-node

---

[3]A more detailed discussion of interventional faithfulness can be found in Appendix B.1.

DAG where this assumption is generically satisfied. In general, we show in Appendix B that a large class of non-linear SCMs and soft interventions satisfy this assumption.

**Example 2.** *Consider $\mathcal{G} = \{1 \rightarrow 2\}$. Let $\mathbb{P}(U_2 \mid U_1) = \mathcal{N}(\beta U_1^2, \sigma_2^2)$ and $\mathbb{P}(U_1) = \mathcal{N}(0, \sigma_1^2)$. Intervention $I$ that changes $\mathbb{P}(U_1)$ into $\mathcal{N}(0, \sigma_1'^2)$ satisfies Assumption 2 as long as $\beta \neq 0$. To see this, note that $\mathbb{P}(U_2 + U_1 C) \neq \mathbb{P}^I(U_2 + U_1 C)$ for any $C$, since $\mathbb{E}_{\mathbb{P}}(U_2 + U_1 C) = \beta \sigma_1^2 \neq \beta \sigma_1'^2 = \mathbb{E}_{\mathbb{P}^I}(U_2 + U_1 C)$.*

Under Assumption 2, we can show that we can identify causal relationships by detecting marginal changes made by interventions. In particular, consider an easier setting where $K = p$, i.e., we have exactly one intervention per latent node. For a source node[4] $i$ of $\mathcal{G}$, $\mathbb{P}(U_i) \neq \mathbb{P}^I(U_i)$ if and only if $T_{\mathcal{G}}(I) = i$. Therefore the source node will have its marginal changed under one intervention amongst $\{I_1, ..., I_p\}$. This is a property of the latent model that we can utilize when solving for it.

Since we have access to $X = U\Lambda + b$, we solve for $U_i$ in the form of $XC^\top + c$ with $C \in \mathbb{R}^n, c \in \mathbb{R}$, or equivalently, $UC^\top + c$ with $C \in \mathbb{R}^p$. By enforcing that $V = UC^\top + c$ only has $\mathbb{P}(V) \neq \mathbb{P}^I(V)$ for one $I \in \{I_1, ..., I_p\}$, Assumption 2 guarantees that $V$ can only be an affine transformation of a source node and that this $I$ corresponds to intervening on this source node. Otherwise: (1) if $C_j \neq 0$ for a non-source node $j$, take $j$ to be the most downstream node with $C_j \neq 0$, then $\mathbb{P}(V) \neq \mathbb{P}^I(V)$ for at least two $I$'s targeting $j$ and its most downstream parents in $\text{pa}_{\mathcal{G}}(j)$; (2) if $C_{i_1} \neq 0$ and $C_{i_2} \neq 0$ for two source nodes $i_1, i_2$, then $\mathbb{P}(V) \neq \mathbb{P}^I(V)$ for two $I$'s targeting $i_1$ and $i_2$.

In general, we can apply this argument to identify all interventions in $I_1, ..., I_K$ that target source nodes of $\mathcal{G}$. Then using an iterative argument, we can identify all interventions that target source nodes of the subgraph of $\mathcal{G}$ after removing its source nodes. This procedure results in the ancestral relations between the targets of $I_1, \ldots, I_K$. Namely, if $T_{\mathcal{G}}(I_k) \in \text{an}_{\mathcal{G}}(T_{\mathcal{G}}(I_j))$, then $I_j$ is identified in a later step than $I_k$ in the above procedure. We thus have the following theorem.

**Theorem 1.** *Under Assumption 1 and Assumption 2 for $I_1, ..., I_K$, we can identify $\langle \hat{\mathcal{G}}, \hat{I}_1, ..., \hat{I}_K \rangle$, where $\hat{\mathcal{G}} = \mathcal{TS}(\mathcal{G}_\pi)$, and $\hat{I}_k = (I_k)_\pi$ for some permutation $\pi$.*

Here $\mathcal{TS}$ denotes the transitive closure of a DAG [57], where $i \rightarrow j \in \mathcal{TS}(\mathcal{G})$ if and only if $i \in \text{an}_{\mathcal{G}}(j)$. Note that this limitation is not due to the linear mixing of the causal variables. It was shown in [57] that with fully observed causal variables, one can only identify a DAG up to its transitive closure by detecting marginal distribution changes. In the next section, we show how to reduce $\mathcal{TS}(\mathcal{G}_\pi)$ to $\mathcal{G}_\pi$, i.e., identifying the CD-equivalence class of $\langle \mathcal{G}, I_1, ..., I_k \rangle$.

### 4.3 Identifying direct edges

DAGs with the same transitive closure can span a spectrum of sparsities; for example, a complete graph and a line graph with the same topological ordering have the same transitive closure. The following example shows that under Assumption 2, in some cases we cannot identify more than the transitive closure.

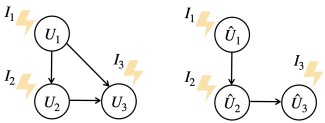

Figure 3

**Example 3.** *Let $\mathcal{G}$ be the 3-node DAG shown on the left in Figure 3. Suppose that $\mathbb{P}(U_1)$ is $\mathcal{N}(0, 1)$, $\mathbb{P}(U_2|U_1)$ is $\mathcal{N}(U_1^2, 1)$, and $\mathbb{P}(U_3|U_1, U_2)$ is $\mathcal{N}((U_1 + U_2)^2, 1)$. Let $f$ be the identity map and $I_1, I_2, I_3$ target nodes $1, 2, 3$, respectively, each changing their conditional variances to 2.[5]*

*Now consider a different model with variables $\hat{U} = (U_1, U_1 + U_2, U_3)$ and mixing function $\hat{f}(\hat{U}) = (\hat{U}_1, \hat{U}_2 - \hat{U}_1, \hat{U}_3)$. Then $\hat{f}(\hat{U}) = U = f(U) = X$. The distributions $\mathbb{P}(\hat{U})$, $\mathbb{P}^{I_1}(\hat{U})$, $\mathbb{P}^{I_2}(\hat{U})$, and $\mathbb{P}^{I_3}(\hat{U})$ each factorizes according to the DAG $\hat{\mathcal{G}}$ that is missing the edge $1 \rightarrow 3$ (Figure 3), where we let $I_1, I_2$ and $I_3$ change the conditional variances of $\hat{U}_1, \hat{U}_2$, and $\hat{U}_3$ to 2, respectively.*

This example shows that we cannot identify $1 \rightarrow 3$ since $U_1 \perp\!\!\!\perp U_3 | U_1 + U_2$. In the case when the causal variables are fully observed, $1 \rightarrow 3$ can be identified by assuming $U_1 \not\perp\!\!\!\perp U_3 | U_2$. However, when allowing for linear mixing, we need to avoid cases such as $U_1 \perp\!\!\!\perp U_3 | U_1 + U_2$ in order to be able to identify $1 \rightarrow 3$. We will show that the following assumption guarantees identifiability of $\mathcal{G}$.

---

[4]A source node is a node without parents.

[5]We show in Appendix B that this model satisfies Assumptions 1 and 2.

When $\mathcal{G}$ is a polytree (a DAG whose skeleton is a tree), this assumption is implied by Assumption 2 under mild regularity conditions (proven in Appendix B). Thus if $\mathcal{G}$ is the sparsest DAG within its transitive closure, we can always identify it with just the Assumptions 1 and 2.

**Assumption 3.** *For every edge $i \to j \in \mathcal{G}$, there do not exist constants $c_j, c_k \in \mathbb{R}$ for $k \in S$ such that $U_i \perp\!\!\!\perp U_j + c_j U_i \mid \{U_l\}_{l \in \mathrm{pa}_{\mathcal{G}}(j) \setminus (S \cup \{i\})}, \{U_k + c_k U_i\}_{k \in S}$, where $S = \mathrm{pa}_{\mathcal{G}}(j) \cap \mathrm{de}_{\mathcal{G}}(i)$.*

**Theorem 2.** *Under Assumptions 1,2,3, $\langle \mathcal{G}, I_1, ..., I_K \rangle$ is identifiable up to its CD-equivalence class.*

### 4.4 Further remarks

Next we discuss if it is possible to recover $U$ along with $\langle \mathcal{G}, I_1, \ldots, I_k \rangle$ up to their CD-equivalence class. Note a simple contradiction with $\mathcal{G} = \{1 \to 2\}$: since we consider general soft interventions, there will always be a valid explanation if we add $U_1$ to $U_2$. Therefore even when we can identify $\langle \mathcal{G}, I_1, \ldots, I_k \rangle$ up to its CD-equivalence class, we still cannot identify $U$ in an element-wise fashion. However, our identifiability results still allow us to draw causal explanations and predict the effect of unseen combinations of interventions, as we discuss below.

**Application of Theorem 1 and 2.** Given unpaired data $\mathcal{D}, \mathcal{D}^{I_1}, ... \mathcal{D}^{I_K}$, these two theorems guarantee that we can identify which $I_1, ..., I_K$ correspond to intervening on the same latent node. Furthermore, Theorem 1 shows that we are able to identify ancestral relationships between the intervention targets of $I_1, ..., I_K$, while Theorem 2 guarantees identifiability of the exact causal structure.

For example, given high-dimensional single-cell transcriptomic readout from a genome-wide knockdown screen, we can under Assumption 1 identify the number of latent causal variables (which we can interpret as the programs of a cell), under Assumption 2 identify which genes belong to the same program, and under Assumption 3 identify the full regulatory relationships between the programs.

**Extrapolation to unseen combinations of interventions.** Theorems 1 and 2 also guarantee that we can predict the effect of unseen combinations of interventions. Namely, consider a combinatorial intervention $\mathcal{I} \subset \{I_1, ..., I_K\}$, where $T_{\mathcal{G}}(I) \neq T_{\mathcal{G}}(I')$ for all $I \neq I' \in \mathcal{I}$. In other words, $\mathcal{I}$ is an intervention with multiple intervention targets that is composed by combining interventions among $I_1, ..., I_K$ with different targets.

Denote by $\langle \hat{U}, \hat{\mathcal{G}}, \hat{I}_1, ..., \hat{I}_K \rangle$ the latent model identified from the interventions $\{I_1, ..., I_K\}$. Recall from Section 3 that we can also infer the mixing function $\hat{f}$ and mechanisms from $\hat{U}, \hat{\mathcal{G}}, \hat{I}_1, ..., \hat{I}_K$. From this, we can infer the interventional distribution under the combinatorial intervention $\mathcal{I}$:

$$X = \hat{f}(\hat{U}), \ \hat{U} \sim \mathbb{P}_{\hat{U}}^{\hat{\mathcal{I}}} = \prod_{\hat{I} \notin \mathcal{I}} \mathbb{P}_{\hat{U}}\big(\hat{U}_{T_{\hat{\mathcal{G}}}(\hat{I})} \mid \hat{U}_{\mathrm{pa}_{\hat{\mathcal{G}}}(T_{\hat{\mathcal{G}}}(\hat{I}))}\big) \cdot \prod_{\hat{I} \in \mathcal{I}} \mathbb{P}_{\hat{U}}^{\hat{I}}\big(\hat{U}_{T_{\hat{\mathcal{G}}}(\hat{I})} \mid \hat{U}_{\mathrm{pa}_{\hat{\mathcal{G}}}(T_{\hat{\mathcal{G}}}(\hat{I}))}\big). \quad (2)$$

We state the conditions for this result informally in the following theorem. A formal version of this theorem together with its proof are given in Appendix B.5.

**Theorem 3** (Informal). *Let $\mathcal{I}$ be a combinatorial intervention (i.e., with multiple intervention targets) combining several interventions among $I_1, ..., I_K$ with different targets. The above procedure allows sampling $X$ according to the distribution $X = f(U), U \sim \mathbb{P}^{\mathcal{I}}$.*

## 5 Discrepancy-based VAE Formulation

Having shown identifiability guarantees for causal disentanglement, we now focus on developing a practical algorithm for recovering the CD-equivalence class from data. As indicated by our proof of Theorem 2, the latent causal graph can be identified by taking the sparsest model compatible with the data. This characterization suggests maximizing a penalized log-likelihood score, a common method for model selection in causal structure learning [10]. The resulting challenging combinatorial optimization problem has been tackled using a variety of approaches, including exact search using integer linear programming [12], greedy search [10, 48, 52], and more recently, gradient-based approaches where the combinatorial search space is relaxed to a continuous search space [70, 34, 38, 61].

Gradient-based approaches offer several potential benefits, including scalability, ease of implementation in automatic differentiation frameworks, and significant flexibility in the choice of components. In light of these benefits, we opted for a gradient-based approach to optimization. In

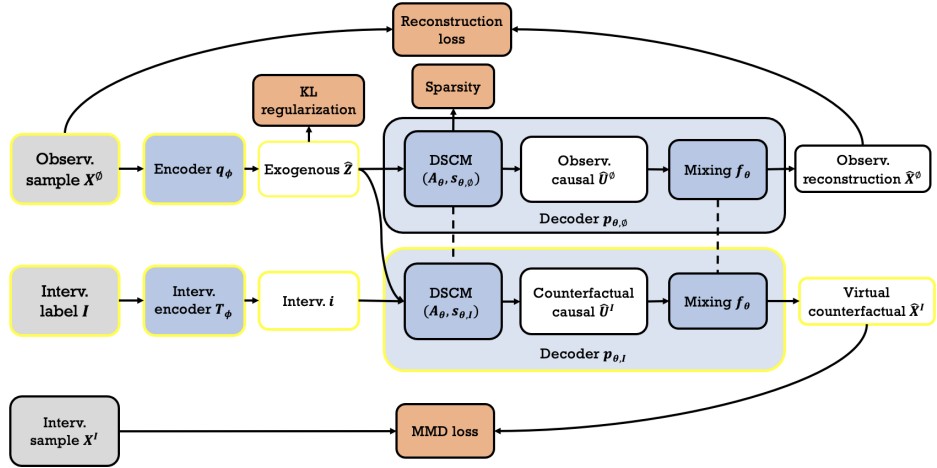

Figure 4: **Our proposed CausalDiscrepancyVAE architecture**. Gray boxes represent inputs, white boxes the generated values, blue boxes the trainable modules, and orange boxes the terms of the loss function. Dashed lines indicate copies of the same module or related modules. Highlighted boxes show the procedure to generate virtural counterfactual samples.

particular, we replace the log-likelihood term of our objective function with a variational lower bound by employing the framework of autoencoding variational Bayes (AVB), widely used in prior works for causal disentanglement [36, 6]. To employ AVB, we re-parameterize each distribution $\mathbb{P}(U_i \mid U_{\mathrm{pa}_{\mathcal{G}}(i)})$ in Eq. (1) into $U_i = s_i(U_{\mathrm{pa}_{\mathcal{G}}(i)}, Z_i)$, where $Z_i$ is an independent exogenous noise variable and $s_i$ denotes the causal mechanism that generates $U_i$ from $U_{\mathrm{pa}_{\mathcal{G}}(i)}$ and $Z_i$. We let $p(Z)$ be a prior distribution over $Z$ and $p_{\theta,\varnothing}(X \mid Z)$ be the conditional distribution of $X$ given $Z$ under no intervention, thereby defining the marginal distribution $p_{\theta,\varnothing}(X)$. Given an arbitrary distribution $q_\phi(Z \mid X)$, we have the following well-known inequality (often called the Evidence Lower Bound or ELBO) for any sample $x$:

$$\log p_{\theta,\varnothing}(x) \geq \mathcal{L}_{\theta,\phi}^{\mathrm{recon}}(x) + \mathcal{L}_\phi^{\mathrm{reg}}(x), \qquad \text{where} \quad \mathcal{L}_{\theta,\phi}^{\mathrm{recon}}(x) := \mathbb{E}_{q_\phi(Z|x)} \log p_{\theta,\varnothing}(x \mid Z),$$

$$\mathcal{L}_\phi^{\mathrm{reg}}(x) := -D_{\mathrm{KL}}(q_\phi(Z \mid x) \| p(Z)).$$

Putting this into the framework of an autoencoder, we call the distribution $q_\phi$ the *encoder* and the distribution $p_{\theta,\varnothing}$ the *decoder*. In our case, the decoder is composed of two functions. First, a deep structural causal model $(A_\theta, s_{\theta,\varnothing})$ maps the exogenous noise $Z$ to the causal variables $U^\varnothing$. In particular, the adjacency matrix $A_\theta$ defines the parent set for each variable, while $s_{\theta,\varnothing} = \{(s_{\theta,\varnothing})_i\}_{i=1}^p$ denotes the learned causal mechanisms. Second, a mixing function $f_\theta$ maps the causal variables $U^\varnothing$ to the observed variables $X^\varnothing$. Because of the permutation symmetry of CD-equivalence, we can fix $A_\theta$ to be upper triangular without loss of generality. We add a loss term $\mathcal{L}_\theta^{\mathrm{sparse}} := -\|A_\theta\|_1$ to encourage $A_\theta$ to be sparse.

While the observational samples are generated from the distribution $p_{\theta,\varnothing}$, the interventional samples are drawn from a different but related distribution $p_{\theta,I}$. The modularity of our decoder allows us to replace $(A_\theta, m_{\theta,\varnothing})$ with an interventional counterpart $(A_\theta, m_{\theta,I})$, while keeping the mixing function $f_\theta$ constant. This is illustrated by the highlighted boxes in Fig. 4. For each intervention label $I$, the corresponding intervention target $i$ and a shift[6] $a_i$ is determined by an intervention encoder $T_\phi$, which uses softmax normalization to approximate a one-hot encoding of the intervention target. Given these intervention targets, we generate "virtual" counterfactual samples for each observational sample. Such samples follow the distribution $\mathbb{P}_{\theta,\phi}(\widehat{X}^{\hat{I}_k})$, the pushforward of $\mathbb{P}_X^\varnothing$ under the action of the encoder $q_\phi$ and decoder $p_{\theta,I}$. These samples are compared to real samples from the corresponding interventional distribution. A variety of discrepancy measures can be used for this comparison. To avoid the saddle point optimization challenges that come with adversarial training, we do not consider adversarial methods (e.g. the dual form of the Wasserstein distance in [5]).

---

[6]For simplicity, we parameterize interventions in DSCM as shifts, though the theoretical results hold for general nonparameteric interventions.

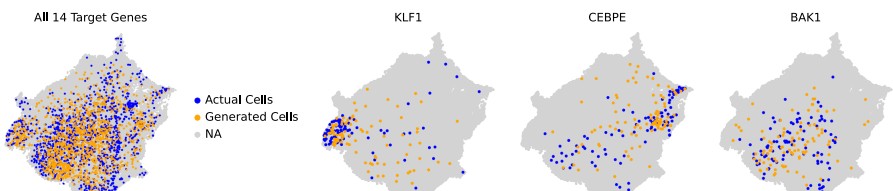

Figure 5: **The distribution of generated samples mirrors the distribution of actual samples.** Samples are visualized using UMAP. *Left:* Samples from the 14 single target-node interventions with more than 800 cells. *Middle-Right:* Samples for target genes SET, CEBPE, and KLF1.

This leaves non-adversarial discrepancy measures, such as the MMD (Maximum Mean Discrepancy) [18], the entropic Wasserstein distance [15], and the sliced Wasserstein distance [63]. In this work, we focus on the MMD measure, whose empirical estimate we recall in Appendix C.1. We take $\mathcal{L}_{\theta,\phi}^{\mathrm{discrep}} := -\sum_{k=1}^{K} \mathrm{MMD}(\mathbb{P}_{\theta,\phi}(\widehat{X}^{\hat{I}_k}), \mathbb{P}_X^{I_k})$. Thus, the full loss function used during training is

$$\mathcal{L}_{\theta,\phi}^{\alpha,\beta,\lambda} := \mathbb{E}_{X^\varnothing}\left[\mathcal{L}_{\theta,\phi}^{\mathrm{reg}}(X) + \beta\mathcal{L}_{\phi}^{\mathrm{recon}}(X)\right] + \alpha\mathcal{L}_{\theta,\phi}^{\mathrm{discrep}} + \lambda\mathcal{L}_{\theta}^{\mathrm{sparse}}. \tag{3}$$

A diagram of the proposed architecture is shown in Fig. 4. Values of the hyperparameters $\alpha, \beta, \lambda$ used in our loss function as well as other hyperparameters are described in Appendix F.

Our loss function exhibits several desirable properties. First, as we show in Appendix D, the unpaired data loss function lower bounds the *paired* data log-likelihood that one would directly optimize in the oracle setting where true counterfactual pairs were available. Second, as we show in Appendix E, this procedure is *consistent*, in the sense that optimizing the loss function in the limit of infinite data will recover the generative process (under suitable conditions). This consistency result also guarantees that the learned model can consistently predict the effect of multi-node interventions; see Appendix E.2.

## 6 Experiments

We now demonstrate our method on a biological dataset. We use the large-scale Perturb-seq study from [44]. After pre-processing, the data contains 8,907 unperturbed cells (observational dataset $\mathcal{D}$) and 99,590 perturbed cells. The perturbed cells underwent CRISPR activation [16] targeting one or two out of 105 genes (interventional datasets $\mathcal{D}^1,...,\mathcal{D}^K$, $K = 217$). CRISPR activation experiments modulate the expression of their target genes, which we model as a shift intervention. Each interventional dataset comprises 50 to 2,000 cells. Each cell is represented as a 5,000-dimensional vector (observed variable $X$) measuring the expressions of 5,000 highly variable genes.

To test our model, we set the latent dimension $p = 105$, corresponding to the total number of targeted genes. During training, we include all the unperturbed cells from $\mathcal{D}$ and the perturbed cells from the single-node interventional datasets $\mathcal{D}^1, ..., \mathcal{D}^{105}$ that target one gene. For each single-node interventional dataset with over 800 cells, we randomly extract 96 cells and reserve these for testing. The double-node interventions (112 distributions $\mathcal{D}^{106}, ..., \mathcal{D}^{217}$) targeting two genes are entirely reserved for testing. The following results summarize the model with the best training performance. Extended evaluations and detailed implementation can be found in Appendix F and G. In additional, we also provide ablation studies on biological data and a simple simulation study in Appendix H.

**Single-node Interventional Distributions.** To study the generative capacity of our model for interventions on single genes, we produce 96 samples for each single-node intervention with over 800 cells (14 interventions). We compare these against the left-out 96 cells of the corresponding distributions. Figure 5 illustrates this for 3 example genes in 2 dimensions using UMAP [41] with all other cells in the dataset as background (labeled by 'NA'). Our model is able to discover subpopulations of the interventional distributions (e.g., for KLF1, the generated samples are concentrated in the middle left corner). We provide a quantitative evaluation for all 105 single-node interventions in Figure 6. The model is able to obtain close to perfect $R^2$ (on average 0.99 over all genes and 0.95 over most differentially expressed genes).

**Double-node Interventional Distributions.** Next, we analyze the generalization capabilities of our model to the 112 double-node interventions. Despite *never* observing any cells from these interventions during training, we obtain reasonable $R^2$ values (on average 0.98 over all genes and

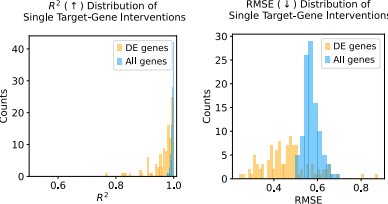

Figure 6: **Our model accurately predicts the effect of single-node interventions.** 'All genes' indicates measurements using the entire 5000-dimensional vectors; 'DE genes' indicates measurements using the 20-dimensional vectors corresponding to the top 20 most differentially expressed genes.

0.88 over most differentially expressed genes). However, when looking at the generated samples for individual pairs of interventions, it is apparent that our model performs well on many pairs, but recovers different subpopulations for some pairs (examples shown in Figure 13 in Appendix G). The wrongly predicted intervention pairs could indicate that the two target genes act non-additively, which needs to be further evaluated and is of independent interest for biological research [22, 44].

**Structure Learning.** Lastly, we examine the learned DAG between the intervention targets. Specifically, this corresponds to a learned gene regulatory network between the learned programs of the target genes. For this, we reduce $p$ from 105 until the learned latent targets of $\mathcal{D}^1, ..., \mathcal{D}^{105}$ cover all $p$ latent nodes. This results in $p = 7$ groups of genes, where genes are grouped by their learned latent nodes. We then run our algorithm with fixed $p = 7$ multiple times and take the learned DAG with the least number of edges. This DAG over the groups of

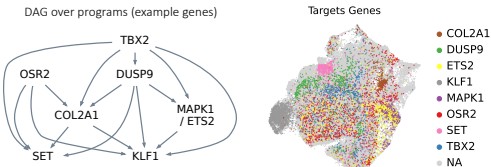

Figure 7: **Structure learning on the biological dataset.** *Left*: learned DAG between target genes (colors indicate edge weights). *Right*: UMAP visualization of the distributions.

targeted genes is shown with example genes in Figure 7 (left). This learned structure is in accordance with previous findings. For example, we successfully identified the edges DUSP9→MAPK1 and DUSP9→ETS2, which is validated in [44] (see their Fig. 5). We also show the interventional distributions targeting these example genes in Figure 7 (right). Among these, MAPK1 and ETS2 correspond to clusters that are heavily overlapping, which explains why the model maps both distributions to the same latent node.

# 7   Conclusion

We derived identifiability results for causal disentanglement from single-node interventions, and presented an autoencoding variational Bayes framework to estimate the latent causal representation from interventional samples. Identification of the latent causal structure and generalization to multi-node interventions was demonstrated experimentally on genetic data.

**Limitations and Future Work.** This paper has various limitations that may be useful to address in future work. We provide an overview here, where an extended discussion can be found in Appendix I. In addition, we also provide a brief summary of concurrent related works in Appendix I.

First, we focused on the setting where single-node interventions on each latent node are available. This is overly optimistic for example in the case of chemical perturbations where the available drugs could all have multiple targets. The VAE framework can still be applied in such settings; however, its theoretical guarantees are subject to further investigations. The key techniques in our proofs can be generalized to the multi-node setting, but further assumptions on the set of interventions needed for identifiability are required. Second, we have focused on the infinite data regime for analyzing identifiability; given that obtaining interventional samples tends to be expensive in practice, there is much room for further investigations in terms of sample complexity.

## Acknowledgements

We thank the Causal Representation Learning Workshop at Bellairs Institute for helpful discussions. All authors acknowledge support by the MIT-IBM Watson AI Lab. In addition, J. Zhang, C. Squires and C. Uhler acknowledge support by the NSF TRIPODS program (DMS-2022448), NCCIH/NIH (1DP2AT012345), ONR (N00014-22-1-2116), the United States Department of Energy (DOE), Office of Advanced Scientific Computing Research (ASCR), via the M2dt MMICC center (DE-SC0023187), the Eric and Wendy Schmidt Center at the Broad Institute, and a Simons Investigator Award.

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

# Contents of Appendix

# A  Useful Lemmas

## A.1  Remarks on Assumption 1

Here we show that the assumption on the functional class of $f$ is satisfied if $f$ is linear and injective, whenever the support of $\mathbb{P}_U$ has non-empty interior. Recall Assumption 1.

**Assumption 1.** *Let $U$ be a $p$-dimensional random vector. Following [3], we assume that the interior of the support of $\mathbb{P}_U$ is a non-empty subset of $\mathbb{R}^p$, and that $f$ is a full row rank polynomial.[7]*

Denote the support of $\mathbb{P}_U, \mathbb{P}_X$ as $\mathbb{U}, \mathbb{X}$ respectively. Let $\mathbb{U}^\circ$ be the interior of $\mathbb{U}$.

**Lemma 2.** *Suppose $\mathbb{U}^\circ$ is a non-empty subset of $\mathbb{R}^p$. If $f : \mathbb{U} \to \mathbb{X}$ is linear and injective, then it must be a full row rank polynomial.*

*Proof.* Since $f$ is linear, it can be written as $f(U) = UH + h$ for some $H \in \mathbb{R}^{p \times n}$ and $h \in \mathbb{R}^p$. If $H$ is not of full row rank, then there exists a non-zero vector $V \in \mathbb{R}^p$ such that $VH = 0$. Let $U \in \mathbb{U}^\circ$, then there exists $\epsilon > 0$ such that $U + \epsilon V \in \mathbb{U}$. We have $f(U + \epsilon V) = f(U)$, which violates $f$ being injective. Therefore $H$ must have full row rank. $\qquad\square$

## A.2  Proof of Lemma 1

The proof of Lemma 1 follows from [3]. For completeness, we present a concise proof here. Then we state a few remarks. Recall Lemma 1.

**Lemma 1.** *Under Assumption 1, we can identify the dimension $p$ of $U$ as well as its linear transformation $U\Lambda + b$ for some non-singular matrix $\Lambda$ and vector $b$. In fact, with observational data, we can only identify $U$ up to such linear transformations.*

*Proof.* We solve for the smallest integer $\hat{p}$ such that there exists a full row rank polynomial $\hat{f} : \mathbb{R}^{\hat{p}} \to \mathbb{R}^n$ where $\hat{U} := \hat{f}^{-1}(X)$ for $X \in \mathbb{X}$ has non-empty support $\hat{\mathbb{U}}^\circ \subseteq \mathbb{R}^{\hat{p}}$. In other words, denote all pairs of $\mathbb{P}_U, f$ that satisfy Assumption 1 as $\mathcal{F}_p$, we solve for

$$\min_{(\mathbb{P}_{\hat{U}}, \hat{f}) \in \mathcal{F}_{\hat{p}}} \hat{p} \quad \text{subject to } \mathbb{P}_{\hat{f}(\hat{U})} = \mathbb{P}_X. \tag{4}$$

Note that $\hat{f}(\hat{U}) = X = f(U)$ for all $U \in \mathbb{U}$. Since $\hat{f}, f$ are full row rank polynomials, there exist full row rank matrices $\hat{H} \in \mathbb{R}^{(p + \ldots + p^{\hat{d}}) \times n}, H \in \mathbb{R}^{(p + \ldots + p^d) \times n}$ and vectors $\hat{h}, h \in \mathbb{R}^n$ such that

$$(\hat{U}, \bar{\otimes}\hat{U}^2, ..., \bar{\otimes}\hat{U}^d)\hat{H} + \hat{h} = \hat{f}(\hat{U}) = X = f(U) = (U, \bar{\otimes}U^2, ..., \bar{\otimes}U^d)H + h. \tag{5}$$

Since $\hat{H}, H$ are of full rank, they have pseudo-inverses $\hat{H}^\dagger, H^\dagger$ such that $\hat{H}\hat{H}^\dagger = \mathbf{I}_{p + \ldots + p^{\hat{d}}}$ and $HH^\dagger = \mathbf{I}_{p + \ldots + p^d}$. Multiplying $\hat{H}^\dagger$ to Eq. (5), we have

$$(\hat{U}, \bar{\otimes}\hat{U}^2, ..., \bar{\otimes}\hat{U}^d) = (U, \bar{\otimes}U^2, ..., \bar{\otimes}U^d)H\hat{H}^\dagger + (h - \hat{h})\hat{H}^\dagger.$$

Therefore $\hat{U}$ can be written as a polynomial of $U$, i.e., $\hat{U} = poly_1(U)$. Similarly, we have $U = poly_2(\hat{U})$. Therefore $U = poly_2(poly_1(U))$ for all $U \in \mathbb{U}$. Since $\mathbb{U}^\circ$ is non-empty, we know that $U = poly_2(poly_1(U))$ on some open set. By the fundamental theorem of algebra [14], we know that $poly_1$ and $poly_2$ must have degree 1. Thus $\hat{U} = U\Lambda + b$ for some full row rank matrix $\Lambda$ and vector $b$. Since $\Lambda \in \mathbb{R}^{p \times \hat{p}}$ is of full row rank, it indicates that $p \leq \hat{p}$. Since $\mathbb{P}_U, f \in \mathcal{F}_p$ satisfy $\mathbb{P}_{f(U)} = \mathbb{P}_X$, by Eq. (4), we must have $\hat{p} \leq p$. Thus $\hat{p} = p$ and $\hat{U} = U\Lambda + b$ for some non-singular matrix $\Lambda$ and vector $b$.

This proof also shows that we can only identify $U$ up to such linear transformations with observational data. Since for any non-singular matrix $\Lambda$ and vector $b$, let $\hat{f}(\hat{U}) = f((\hat{U} - b)\Lambda^{-1})$. We have $\mathbb{P}_{\hat{U}}, \hat{f}$ satisfy Assumption 1 and they generate the same observational data. $\qquad\square$

---

[7]There exists some integer $d$, a full row rank $H \in \mathbb{R}^{(p + \ldots + p^d) \times n}$ and a vector $h \in \mathbb{R}^n$ such that $f(U) = (U, \bar{\otimes}U^2, ..., \bar{\otimes}U^d)H + h$, where $\bar{\otimes}U^k$ denotes the size-$p^k$ vector with degree-$k$ polynomials of $U$ as its entries.

**Remark 1.** *With observational data $X = f(U) \in \mathcal{D}$, we can identify $\hat{U} = \hat{g}(X)$ such that $\hat{U} = U\Lambda + b$ for non-singular $\Lambda$. Then for any interventional data $X = f(U) \in \mathcal{D}^I$, the analytic continuation of $\hat{g}$ to $\mathcal{D}^I$ satisfies $\hat{U} := \hat{g}(X) = U\Lambda + b$ for all $X \in \mathcal{D}^I$.*

*Proof.* The proof follows immediately by writing $\hat{g}, f^{-1}$ as polynomial functions. $\qquad\square$

Next, we discuss identifiability of the underlying DAG $\mathcal{G}$. First, we give an example showing that any causal DAG can explain the observational data.

**Example 4.** *Suppose the ground-truth DAG is an empty graph $\mathcal{G} = \varnothing$. With observational data alone, any DAG can explain the data.*

*Proof.* Let $\hat{\mathcal{G}}$ be an arbitrary DAG with topological order $\tau(1), ..., \tau(p)$, i.e., $\tau(j) \in \mathrm{pa}_{\hat{\mathcal{G}}}(\tau(i))$ only if $j < i$. Let $\Lambda$ be the permutation matrix such that $\hat{U} = U\Lambda$ satisfies $\hat{U}_{\tau(i)} = U_i$ for any $i \in [p]$. Then $\hat{U}$ factorizes as $\mathbb{P}(\hat{U}) = \mathbb{P}(U) = \prod_{i=1}^{p} \mathbb{P}(U_i) = \prod_{i=1}^{p} \mathbb{P}(\hat{U}_{\tau(i)})$. This implies $\hat{U}_{\tau(i)} \perp\!\!\!\perp \hat{U}_{\tau(j)}$ for $j \le i - 1$. Therefore $\mathbb{P}(\hat{U}_{\tau(i)}) = \mathbb{P}(\hat{U}_{\tau(i)} \mid \hat{U}_{\mathrm{pa}_{\hat{\mathcal{G}}(i)}})$ and $\mathbb{P}(\hat{U}) = \prod_{i=1}^{p} \mathbb{P}(\hat{U}_{\tau(i)} \mid \hat{U}_{\mathrm{pa}_{\hat{\mathcal{G}}(i)}})$ factorizes with respect to $\hat{\mathcal{G}}$. Thus $\hat{\mathcal{G}}$ can explain the data. $\qquad\square$

Therefore with observational data alone, we cannot identify the underlying DAG $\mathcal{G}$ up to any nontrivial equivalence class. In [3], it was shown that with a *do* intervention[8] per latent node and assuming the interior of the support of the non-targeted variables is non-empty, one can identify $U$ up to a finer class of linear transformations. Namely, one can identify $U$ up to CD-equivalence (permutation and element-wise affine transformation); see Definition 1. Then, assuming for example faithfulness and influentiality [57], one can identify $\mathcal{G}$.

While several extensions beyond do-interventions are discussed in [3], they all involve manipulating the *support* of the intervention targets. In the case where the support of the intervention targets remains unchanged (e.g., additive Gaussian SCMs with shift interventions), a completely new approach and theory needs to be developed.

## B   Proof of Identifiability with Soft Interventions

In this section, we provide the proofs for the results in Section 4. While our main focus is on general types of soft interventions, our results also apply to hard interventions which include do-interventions as a special case.

**Notation.** We let $e_i$ denote the indicator vector with the $i$-th entry equal to one and all other entries equal to zero. To be consistent with other notation in the paper, let $e_i \in \mathbb{R}^p$ be a row vector. We call $j \in \mathrm{ch}_{\mathcal{G}}(i)$ a *maximal child* of $i$ if $\mathrm{pa}_{\mathcal{G}}(j) \cap \mathrm{de}_{\mathcal{G}}(i) = \varnothing$. Denote the set of all maximal children of $i$ as $\mathrm{mch}_{\mathcal{G}}(i)$. For node $i$, define $\overline{\mathrm{de}}_{\mathcal{G}}(i) := \mathrm{de}_{\mathcal{G}}(i) \cup \{i\}$. Given a DAG $\mathcal{G}$, we denote the transitive closure of $\mathcal{G}$ by $\mathcal{TS}(\mathcal{G})$, i.e., $i \to j \in \mathcal{TS}(\mathcal{G})$ if and only if there is a directed path from $i$ to $j$ in $\mathcal{G}$.

### B.1   Faithfulness Assumptions

We start by discussing previous interventional faithfulness assumptions. Prior interventional faithfulness assumptions [57, 66, 24] vary by a few technicalities; but they all assume that all causal variables are observed (causal sufficiency), and, more importantly, that intervening on a node will always change the marginal of its descendants. In particular, [57] (Definition 2, called "influentiality") only made this assumption and showed that the causal graph is identifiable up to its transitive closure by detecting marginal changes. [57] showed that their algorithm consistently identifies the full causal graph by assuming additionally that intervening on a node changes the conditional distribution of its direct children giving its neighbors (details can be found in Assumption 4.5 of [66]). A similar notion was also introduced in [24] where they made further assumptions regarding changes in the conditional distributions.

---

[8]Do interventions are a special type of hard interventions where the intervention target collapses to one specific value.

We now show our linear interventional faithfulness (Assumption 2) is satisfied by a large class of nonlinear SCMs and soft interventions. Recall Assumption 2.

**Assumption 2.** *Intervention $I$ with target $i$ satisfies* linear interventional faithfulness *if for every $j \in \{i\} \cup \mathrm{ch}_{\mathcal{G}}(i)$ such that $\mathrm{pa}_{\mathcal{G}}(j) \cap \mathrm{de}_{\mathcal{G}}(i) = \varnothing$, it holds that $\mathbb{P}(U_j + U_S C^\top) \neq \mathbb{P}^I(U_j + U_S C^\top)$ for all constant vectors $C \in \mathbb{R}^{|S|}$, where $S = [p] \setminus (\{j\} \cup \mathrm{de}_{\mathcal{G}}(i))$.*

In Example 2, we discussed a 2-node graph where Assumption 2 is satisfied. This example can be extended in the following way, which subsumes the case in Example 3.

**Example 5.** *Consider an SCM with additive noise, where each mechanism $\mathbb{P}(U_k \mid U_{\mathrm{pa}_{\mathcal{G}}(k)})$ is specified by $U_k = s_k(U_{\mathrm{pa}_{\mathcal{G}}(k)}) + \epsilon_k$, where $\epsilon_k$ for $k \in [p]$ are independent exogenous noise variables. Assumption 2 is satisfied if $I$ only changes the variance of $\epsilon_i$ and $s_j$ is a quadratic function with non-zero coefficient of $U_i^2$ for each $j \in \mathrm{mch}_{\mathcal{G}}(i)$.*

*Proof.* If $j = i$ in Assumption 2, then $S = [p] \setminus \overline{\mathrm{de}}_{\mathcal{G}}(i) \supset \mathrm{pa}_{\mathcal{G}}(i)$. Since $U_S \perp\!\!\!\perp \epsilon_i$, we have

$$\mathrm{Var}(U_i + U_S C^\top) = \mathrm{Var}(\epsilon_i) + \mathrm{Var}(s_i(U_{\mathrm{pa}_{\mathcal{G}}(i)}) + U_S C^\top).$$

Note that $\mathbb{P}^I$ does not change the joint distribution of $U_S$, and therefore

$$\mathrm{Var}_{\mathbb{P}}(s_i(U_{\mathrm{pa}_{\mathcal{G}}(i)}) + U_S C^\top) = \mathrm{Var}_{\mathbb{P}^I}(s_i(U_{\mathrm{pa}_{\mathcal{G}}(i)}) + U_S C^\top).$$

By $\mathrm{Var}_{\mathbb{P}}(\epsilon_i) \neq \mathrm{Var}_{\mathbb{P}^I}(\epsilon_i)$, we then know that$\mathrm{Var}_{\mathbb{P}}(U_i + U_S C^\top) \neq \mathrm{Var}_{\mathbb{P}^I}(U_i + U_S C^\top)$. Thus $\mathbb{P}(U_i + U_S C^\top) \neq \mathbb{P}^I(U_i + U_S C^\top)$.

If $j \neq i$ in Assumption 2, then by linearity of expectation $\mathbb{E}(U_j + U_S C^\top) = \mathbb{E}(U_j) + \mathbb{E}(U_S)C^\top$. Note that $S = [p] \setminus (\{j\} \cup \mathrm{de}_{\mathcal{G}}(i)) = [p] \setminus \mathrm{de}_{\mathcal{G}}(i)$, and therefore $\mathbb{E}_{\mathbb{P}}(U_S) = \mathbb{E}_{\mathbb{P}^I}(U_S)$. Next we show that $\mathbb{E}_{\mathbb{P}}(U_j) \neq \mathbb{E}_{\mathbb{P}^I}(U_j)$. Once this is proven, then we have that $\mathbb{E}_{\mathbb{P}}(U_j + U_S C^\top) \neq \mathbb{E}_{\mathbb{P}^I}(U_j + U_S C^\top)$, which concludes the proof for $\mathbb{P}(U_j + U_S C^\top) \neq \mathbb{P}^I(U_j + U_S C^\top)$.

Since $s_j$ is a quadratic function of $U_i$, suppose the coefficient of $U_i^2$ in $s_j$ is $\beta \neq 0$. Then

$$
\begin{aligned}
\mathbb{E}(U_j) - \mathbb{E}(\epsilon_j) &= \mathbb{E}(U_j - \epsilon_j) \\
&= \mathbb{E}\big(s_{j,0}(U_{\mathrm{pa}_{\mathcal{G}}(j)\setminus\{i\}}) + s_{j,1}(U_{\mathrm{pa}_{\mathcal{G}}(j)\setminus\{i\}}) \cdot U_i + \beta U_i^2\big) \\
&= \mathbb{E}\big(s_{j,0}(U_{\mathrm{pa}_{\mathcal{G}}(j)\setminus\{i\}}) + s'_{j,1}(U_{\mathrm{pa}_{\mathcal{G}}(j)\setminus\{i\}}, U_{\mathrm{pa}_{\mathcal{G}}(i)}) \cdot \epsilon_i + \beta \epsilon_i^2\big) \\
&= \mathbb{E}\big(s_{j,0}(U_{\mathrm{pa}_{\mathcal{G}}(j)\setminus\{i\}})\big) + \mathbb{E}\big(s'_{j,1}(U_{\mathrm{pa}_{\mathcal{G}}(j)\setminus\{i\}}, U_{\mathrm{pa}_{\mathcal{G}}(i)}) \cdot \epsilon_i\big) + \beta \mathbb{E}(\epsilon_i^2),
\end{aligned}
\tag{6}
$$

for some functions $s_{j,0}, s_{j,1}$ and $s'_{j,1}$. Since $\mathrm{pa}_{\mathcal{G}}(j) \cap \mathrm{de}_{\mathcal{G}}(i) = \varnothing$, we know that $\mathbb{P}^I$ will not change the joint distribution of $U_{\mathrm{pa}_{\mathcal{G}}(j)\setminus\{i\}}$ and that $\epsilon_i \perp\!\!\!\perp U_{\mathrm{pa}_{\mathcal{G}}(j)\setminus\{i\}}, U_{\mathrm{pa}_{\mathcal{G}}(i)}$. Therefore we have

$$
\begin{aligned}
\mathbb{E}_{\mathbb{P}}\big(s_{j,0}(U_{\mathrm{pa}_{\mathcal{G}}(j)\setminus\{i\}})\big) &= \mathbb{E}_{\mathbb{P}^I}\big(s_{j,0}(U_{\mathrm{pa}_{\mathcal{G}}(j)\setminus\{i\}})\big), \\
\mathbb{E}_{\mathbb{P}}\big(s'_{j,1}(U_{\mathrm{pa}_{\mathcal{G}}(j)\setminus\{i\}}, U_{\mathrm{pa}_{\mathcal{G}}(i)}) \cdot \epsilon_i\big) &= \mathbb{E}_{\mathbb{P}}\big(s'_{j,1}(U_{\mathrm{pa}_{\mathcal{G}}(j)\setminus\{i\}}, U_{\mathrm{pa}_{\mathcal{G}}(i)})\big) \cdot \mathbb{E}_{\mathbb{P}}(\epsilon_i) \\
&= \mathbb{E}_{\mathbb{P}^I}\big(s'_{j,1}(U_{\mathrm{pa}_{\mathcal{G}}(j)\setminus\{i\}}, U_{\mathrm{pa}_{\mathcal{G}}(i)})\big) \cdot \mathbb{E}_{\mathbb{P}^I}(\epsilon_i) \\
&= \mathbb{E}_{\mathbb{P}^I}\big(s'_{j,1}(U_{\mathrm{pa}_{\mathcal{G}}(j)\setminus\{i\}}, U_{\mathrm{pa}_{\mathcal{G}}(i)}) \cdot \epsilon_i\big).
\end{aligned}
$$

By $\mathbb{E}_{\mathbb{P}}(\epsilon_j) = \mathbb{E}_{\mathbb{P}^I}(\epsilon_j)$, $\mathbb{E}_{\mathbb{P}}(\epsilon_i^2) \neq \mathbb{E}_{\mathbb{P}^I}(\epsilon_i^2)$ and Eq. (6), we have $\mathbb{E}_{\mathbb{P}}(U_j) \neq \mathbb{E}_{\mathbb{P}^I}(U_j)$, which concludes the proof. $\square$

This example shows how we may check $\mathbb{P}(U_j + U_S C^\top) \neq \mathbb{P}^I(U_j + U_S C^\top)$ by examining the mean and variance of $U_j + U_S C^\top$. In general, this can be extended to checking any finite moments of $U_j + U_S C^\top$ as stated in the following lemma.

**Lemma 3.** *Assumption 2 is satisfied if for each $i \in [p]$ one of the following conditions holds:*

*(1) if $\mathbb{E}_{\mathbb{P}}(U_i \mid U_{\mathrm{pa}_{\mathcal{G}}(i)}) = \mathbb{E}_{\mathbb{P}^I}(U_i \mid U_{\mathrm{pa}_{\mathcal{G}}(i)})$, then there exits an integer $m > 1$ such that*

$$\mathbb{E}_{\mathbb{P}}(U_i^m \mid U_{\mathrm{pa}_{\mathcal{G}}(i)}) \neq \mathbb{E}_{\mathbb{P}^I}(U_i^m \mid U_{\mathrm{pa}_{\mathcal{G}}(i)}),$$

*and the smallest $m$ that satisfies this also satisfies $\mathbb{E}_{\mathbb{P}}(U_i^m) \neq \mathbb{E}_{\mathbb{P}^I}(U_i^m)$. In addition, for all $j \in \mathrm{mch}_{\mathcal{G}}(i)$, it holds that $\mathbb{E}_{\mathbb{P}}(U_j) \neq \mathbb{E}_{\mathbb{P}^I}(U_j)$;*

*(2)* if $\mathbb{E}_{\mathbb{P}}(U_i) \neq \mathbb{E}_{\mathbb{P}^I}(U_i)$, *then for all* $j \in \mathrm{mch}_{\mathcal{G}}(i)$, *there exists an integer* $m > 1$ *such that*
$$\mathbb{E}_{\mathbb{P}}((U_j + c_j U_i)^m \mid U_{S \setminus \{i\}}) \neq \mathbb{E}_{\mathbb{P}^I}((U_j + c_j U_i)^m \mid U_{S \setminus \{i\}}),$$
*where $S$ is as defined in Assumption 2, and the smallest $m$ that satisfies this also satisfies* $\mathbb{E}_{\mathbb{P}}((U_j + c_j U_i)^m) \neq \mathbb{E}_{\mathbb{P}^I}((U_j + c_j U_i)^m)$, *where*
$$c_j = -\frac{(\mathbb{E}_{\mathbb{P}}(U_j) - \mathbb{E}_{\mathbb{P}^I}(U_j))}{(\mathbb{E}_{\mathbb{P}}(U_i) - \mathbb{E}_{\mathbb{P}^I}(U_i))}.$$

*Proof.* Suppose (1) holds true. If $j = i$ in Assumption 2, then $\mathbb{P}(U_S) = \mathbb{P}^I(U_S)$ for $S = [p] \setminus \overline{\mathrm{de}}_{\mathcal{G}}(i)$, and

$$\mathbb{E}_{\mathbb{P}}\big((U_i + U_S C^\top)^m\big)$$
$$= \mathbb{E}_{\mathbb{P}}(U_i^m) + \sum_{l=0}^{m-1} \binom{m}{l} \mathbb{E}_{\mathbb{P}}\big(U_i^l (U_S C^\top)^{m-l}\big)$$
$$= \mathbb{E}_{\mathbb{P}}(U_i^m) + \sum_{l=0}^{m-1} \binom{m}{l} \mathbb{E}_{\mathbb{P}}\Big(\mathbb{E}_{\mathbb{P}}(U_i^l | U_S) \cdot (U_S C^\top)^{m-l}\Big) \quad \text{(law of total expectation)}$$
$$= \mathbb{E}_{\mathbb{P}}(U_i^m) + \sum_{l=0}^{m-1} \binom{m}{l} \mathbb{E}_{\mathbb{P}}\Big(\mathbb{E}_{\mathbb{P}}(U_i^l | U_{\mathrm{pa}_{\mathcal{G}}(i)}) \cdot (U_S C^\top)^{m-l}\Big) \quad \text{(since } U_i \perp\!\!\!\perp U_{S \setminus \mathrm{pa}_{\mathcal{G}}(i)} \mid U_{\mathrm{pa}_{\mathcal{G}}(i)})$$
$$\neq \mathbb{E}_{\mathbb{P}^I}(U_i^m) + \sum_{l=0}^{m-1} \binom{m}{l} \mathbb{E}_{\mathbb{P}}\Big(\mathbb{E}_{\mathbb{P}^I}(U_i^l | U_{\mathrm{pa}_{\mathcal{G}}(i)}) \cdot (U_S C^\top)^{m-l}\Big)$$
$$= \mathbb{E}_{\mathbb{P}^I}(U_i^m) + \sum_{l=0}^{m-1} \binom{m}{l} \mathbb{E}_{\mathbb{P}^I}\Big(\mathbb{E}_{\mathbb{P}^I}(U_i^l | U_{\mathrm{pa}_{\mathcal{G}}(i)}) \cdot (U_S C^\top)^{m-l}\Big) = \mathbb{E}_{\mathbb{P}^I}\big((U_i + U_S C^\top)^m\big),$$

where the inequality is because of $\mathbb{E}_{\mathbb{P}}(U_i^m) \neq \mathbb{E}_{\mathbb{P}^I}(U_i^m)$ and $\mathbb{E}_{\mathbb{P}}(U_i^l | U_{\mathrm{pa}_{\mathcal{G}}(i)}) = \mathbb{E}_{\mathbb{P}^I}(U_i^l | U_{\mathrm{pa}_{\mathcal{G}}(i)})$ for any $l < m$. Therefore $\mathbb{P}(U_i + U_S C^\top) \neq \mathbb{P}^I(U_i + U_S C^\top)$.

If $j \neq i$ in Assumption 2, then $\mathbb{E}_{\mathbb{P}}(U_j) \neq \mathbb{E}_{\mathbb{P}^I}(U_j)$ implies $\mathbb{E}_{\mathbb{P}}(U_j + U_S C^\top) \neq \mathbb{E}_{\mathbb{P}^I}(U_j + U_S C^\top)$, which proves that $\mathbb{P}(U_j + U_S C^\top) \neq \mathbb{P}^I(U_j + U_S C^\top)$.

Suppose (2) holds true. If $j = i$ in Assumption 2, then $\mathbb{E}_{\mathbb{P}}(U_i) \neq \mathbb{E}_{\mathbb{P}^I}(U_i)$ implies $\mathbb{E}_{\mathbb{P}}(U_i + U_S C^\top) \neq \mathbb{E}_{\mathbb{P}^I}(U_i + U_S C^\top)$, which proves that $\mathbb{P}(U_i + U_S C^\top) \neq \mathbb{P}^I(U_i + U_S C^\top)$.

If $j \neq i$ in Assumption 2, then for $C \in \mathbb{R}^{|S|}$, if the coordinate for $U_i$ is not $c_j$, then $\mathbb{E}_{\mathbb{P}}(U_i + U_S C^\top) = \mathbb{E}_{\mathbb{P}}(U_i) + \mathbb{E}_{\mathbb{P}}(U_S)C^\top \neq \mathbb{E}_{\mathbb{P}^I}(U_i) + \mathbb{E}_{\mathbb{P}^I}(U_S)C^\top = \mathbb{E}_{\mathbb{P}^I}(U_i + U_S C^\top)$, since $\mathbb{E}_{\mathbb{P}}(U_{S \setminus \{i\}}) = \mathbb{E}_{\mathbb{P}^I}(U_{S \setminus \{i\}})$. If the coordinate for $U_i$ in $C$ is $c_j$, denote $U_S C^\top = U_{S \setminus \{i\}} C_{-j}^\top + c_j U_i$, and then similar to above we obtain

$$\mathbb{E}_{\mathbb{P}}\big((U_j + U_S C^\top)^m\big)$$
$$= \mathbb{E}_{\mathbb{P}}\big((U_j + c_j U_i + U_{S \setminus \{i\}} C_{-j}^\top)^m\big)$$
$$= \mathbb{E}_{\mathbb{P}}\big((U_i + c_j U_i)^m\big) + \sum_{l=0}^{m-1} \binom{m}{l} \mathbb{E}_{\mathbb{P}}\big((U_i + c_j U_i)^l (U_{S \setminus \{i\}} C_{-j}^\top)^{m-l}\big)$$
$$= \mathbb{E}_{\mathbb{P}}\big((U_i + c_j U_i)^m\big) + \sum_{l=0}^{m-1} \binom{m}{l} \mathbb{E}_{\mathbb{P}}\Big(\mathbb{E}_{\mathbb{P}}\big((U_i + c_j U_i)^l | U_{S \setminus \{i\}}\big) \cdot (U_{S \setminus \{i\}} C_{-j}^\top)^{m-l}\Big)$$
$$\neq \mathbb{E}_{\mathbb{P}^I}\big((U_i + c_j U_i)^m\big) + \sum_{l=0}^{m-1} \binom{m}{l} \mathbb{E}_{\mathbb{P}}\Big(\mathbb{E}_{\mathbb{P}^I}\big((U_i + c_j U_i)^l | U_{S \setminus \{i\}}\big) \cdot (U_{S \setminus \{i\}} C_{-j}^\top)^{m-l}\Big)$$
$$= \mathbb{E}_{\mathbb{P}^I}\big((U_i + c_j U_i)^m\big) + \sum_{l=0}^{m-1} \binom{m}{l} \mathbb{E}_{\mathbb{P}^I}\Big(\mathbb{E}_{\mathbb{P}^I}\big((U_i + c_j U_i)^l | U_{S \setminus \{i\}}\big) \cdot (U_{S \setminus \{i\}} C_{-j}^\top)^{m-l}\Big)$$
$$= \mathbb{E}_{\mathbb{P}^I}\big((U_j + U_S C^\top)^m\big).$$

Thus $\mathbb{P}(U_j + U_S C^\top) \neq \mathbb{P}^I(U_j + U_S C^\top)$, which completes the proof. $\qquad \square$

This lemma gives a sufficient condition for Assumption 2 to hold. Since it involves only finite moments of the variables, one can easily check if this is satisfied for a given SCM associated with soft interventions. Note that Example 5 satisfies the first condition of Lemma 3 for $m = 2$.

Next we show that Assumption 3 is satisfied on a tree graph if Assumption 2 holds, under mild regularity conditions such as that the interventional support lies within the observational support. Recall Assumption 3.

**Assumption 3.** *For every edge $i \to j \in \mathcal{G}$, there do not exist constants $c_j, c_k \in \mathbb{R}$ for $k \in S$ such that $U_i \perp\!\!\!\perp U_j + c_j U_i \mid \{U_l\}_{l \in \mathrm{pa}_{\mathcal{G}}(j) \setminus (S \cup \{i\})}, \{U_k + c_k U_i\}_{k \in S}$, where $S = \mathrm{pa}_{\mathcal{G}}(j) \cap \deg_{\mathcal{G}}(i)$.*

**Lemma 4.** *Suppose $\mathcal{G}$ is a polytree and Assumption 2 holds for an intervention $I$ targeting node $i$. Then for any edge $i \to j \in \mathcal{G}$, Assumption 3 holds if* [9]

$$\mathbb{P}(U_i = u \mid U_{\mathrm{pa}_{\mathcal{G}}(j) \setminus \{i\}}) = 0 \quad \Rightarrow \quad \mathbb{P}^I(U_i = u \mid U_{\mathrm{pa}_{\mathcal{G}}(j) \setminus \{i\}}) = 0, \tag{7}$$

*for almost every $u$ and all realizations of $U_{\mathrm{pa}_{\mathcal{G}}(j) \setminus \{i\}}$.*

*Proof.* Suppose $\mathcal{G}$ is a tree graph and Assumption 2 holds for $I$ targeting $i$. For any edge $i \to j \in \mathcal{G}$, since there is only one undirected path between $i$ and $j$, we have $S = \mathrm{pa}_{\mathcal{G}}(j) \cap \deg_{\mathcal{G}}(i) = \varnothing$. Therefore we only need to show that $U_i$ and $U_j + c_j U_i$ are not conditionally independent given $U_{\mathrm{pa}_{\mathcal{G}}(j) \setminus \{i\}}$ for any $c_j \in \mathbb{R}$.

The regularity condition in Eq. (7) ensures that

$$\int_{\mathbb{P}(U_i = r \mid U_{\mathrm{pa}_{\mathcal{G}}(j) \setminus \{i\}}) \neq 0} \mathbb{P}^I(U_i = r \mid U_{\mathrm{pa}_{\mathcal{G}}(j) \setminus \{i\}}) dr = 1, \tag{8}$$

for any realization of $U_{\mathrm{pa}_{\mathcal{G}}(j) \setminus \{i\}}$.

Suppose $U_i$ and $U_j + c_j U_i$ are conditionally independent given $U_{\mathrm{pa}_{\mathcal{G}}(j) \setminus \{i\}}$. Then for any $l \in \mathbb{R}$ and realization of $U_{\mathrm{pa}_{\mathcal{G}}(j) \setminus \{i\}}, U_i$ (denote the realization of $U_i$ as $r$),

$$\mathbb{P}(U_j + c_j U_i = l \mid U_{\mathrm{pa}_{\mathcal{G}}(j) \setminus \{i\}}) = \mathbb{P}(U_j + c_j U_i = l \mid U_i = r, U_{\mathrm{pa}_{\mathcal{G}}(j) \setminus \{i\}})$$
$$= \mathbb{P}(U_j = l - c_j r \mid U_i = r, U_{\mathrm{pa}_{\mathcal{G}}(j) \setminus \{i\}}).$$

Since this is true for any $r$ with $\mathbb{P}(U_i = r \mid U_{\mathrm{pa}_{\mathcal{G}}(j) \setminus \{i\}}) \neq 0$, by Eq. (8), we have

$$\mathbb{P}(U_j + c_j U_i = l \mid U_{\mathrm{pa}_{\mathcal{G}}(j) \setminus \{i\}})$$
$$= \int_{\mathbb{P}(U_i = r \mid U_{\mathrm{pa}_{\mathcal{G}}(j) \setminus \{i\}}) \neq 0} \mathbb{P}(U_j + c_j U_i = l \mid U_{\mathrm{pa}_{\mathcal{G}}(j) \setminus \{i\}}) \cdot \mathbb{P}^I(U_i = r \mid U_{\mathrm{pa}_{\mathcal{G}}(j) \setminus \{i\}}) dr$$
$$= \int_{\mathbb{P}(U_i = r \mid U_{\mathrm{pa}_{\mathcal{G}}(j) \setminus \{i\}}) \neq 0} \mathbb{P}(U_j = l - c_j r \mid U_i = r, U_{\mathrm{pa}_{\mathcal{G}}(j) \setminus \{i\}}) \cdot \mathbb{P}^I(U_i = r \mid U_{\mathrm{pa}_{\mathcal{G}}(j) \setminus \{i\}}) dr.$$

Note that $\mathbb{P}(U_j \mid U_i, U_{\mathrm{pa}_{\mathcal{G}}(j) \setminus \{i\}}) = \mathbb{P}^I(U_j \mid U_i, U_{\mathrm{pa}_{\mathcal{G}}(j) \setminus \{i\}})$ since $I$ targets $i$, and we therefore have

$$\mathbb{P}(U_j + c_j U_i = l \mid U_{\mathrm{pa}_{\mathcal{G}}(j) \setminus \{i\}})$$
$$= \int_{\mathbb{P}(U_i = r \mid U_{\mathrm{pa}_{\mathcal{G}}(j) \setminus \{i\}}) \neq 0} \mathbb{P}^I(U_j = l - c_j r \mid U_i = r, U_{\mathrm{pa}_{\mathcal{G}}(j) \setminus \{i\}}) \cdot \mathbb{P}^I(U_i = r \mid U_{\mathrm{pa}_{\mathcal{G}}(j) \setminus \{i\}}) dr$$
$$= \int_{\mathbb{P}(U_i = r \mid U_{\mathrm{pa}_{\mathcal{G}}(j) \setminus \{i\}}) \neq 0} \mathbb{P}^I(U_j = l - c_j r, U_i = r \mid U_{\mathrm{pa}_{\mathcal{G}}(j) \setminus \{i\}}) dr$$
$$= \int_{\mathbb{P}^I(U_i = r \mid U_{\mathrm{pa}_{\mathcal{G}}(j) \setminus \{i\}}) \neq 0} \mathbb{P}^I(U_j = l - c_j r, U_i = r \mid U_{\mathrm{pa}_{\mathcal{G}}(j) \setminus \{i\}}) dr$$
$$= \mathbb{P}^I(U_j + c_j U_i = l \mid U_{\mathrm{pa}_{\mathcal{G}}(j) \setminus \{i\}}),$$

where the second-to-last equality uses the regularity condition in Eq. (7).

---

[9] For simplicity, we assume $U$ is continuous and treat $\mathbb{P}$ as the density. For discrete $U$, the proofs extend by replacing $\int$ with $\sum$.

Since $\mathrm{pa}_{\mathcal{G}}(j) \cap \mathrm{de}_{\mathcal{G}}(i) = \varnothing$, it holds that $\mathbb{P}(U_{\mathrm{pa}_{\mathcal{G}}(j)\backslash\{i\}}) = \mathbb{P}^I(U_{\mathrm{pa}_{\mathcal{G}}(j)\backslash\{i\}})$, and thus $\mathbb{P}(U_j + c_j U_i) = \mathbb{P}^I(U_j + c_j U_i)$, which is a contradiction to linear interventional faithfulness of $I$. Therefore, we must have that $U_i$ and $U_j + c_j U_i$ are not conditionally independent given $U_{\mathrm{pa}_{\mathcal{G}}(j)\backslash\{i\}}$, which completes the proof. $\qquad\square$

Essentially, Assumption 2 guarantees influentiality and Assumption 3 guarantees adjacency faithfulness. These assumptions differ from existing faithfulness conditions (c.f., [57, 58, 67]) due to the fact that we can only observe a linear mixing of the causal variables.

## B.2 Summary of representations

In the remainder of this appendix, we will develop a series of representations which are increasingly related to the underlying representation $U$. These representations are summarized in Table 1.

| Symbol | Definition | | Section |
|---|---|---|---|
| $U$ | | | Section 2 |
| $X$ | $X = U\Lambda + b,$ | $\Lambda \in \mathbb{R}^{p\times p}, b \in \mathbb{R}^p$ | Section 2 |
| $\tilde{U}$ | $\tilde{U} = U\tilde{\Gamma} + \tilde{c},$ | $\tilde{\Gamma} = \Lambda\Pi, \tilde{c} = b\Pi$ for $\Pi \in \mathbb{R}^{p\times p}$ | Appendix B.3.1 |
| $\hat{U}$ | $\hat{U} = U\hat{\Gamma} + \hat{c},$ | $\hat{\Gamma} = \tilde{\Gamma}\hat{R}, \hat{c} = \tilde{c}\hat{R}$ for $\hat{R} \in \mathbb{R}^{p\times p}$ upp. tri. | Appendix B.3.2 |
| $\bar{U}$ | $\bar{U} = U\bar{\Gamma} + \bar{c},$ | $\bar{\Gamma} = \hat{\Gamma}\bar{R}, \bar{c} = \hat{c}\bar{R}$ for $\bar{R} \in \mathbb{R}^{p\times p}$ upp. tri. | Appendix B.4 |

Table 1: **Representations of $U$ that are used in this appendix.** Note that, under Assumption 1, we can assume $X = U\Lambda + b$ without loss of generality, by Lemma 1 and Remark 1.

## B.3 Proof of Theorem 1

In the main text (Section 4.2), we laid out an illustrative procedure to identify the transitive closure of $\mathcal{G}$ when we consider a simpler setting with $K = p$. This process relies on iteratively finding source nodes of $\mathcal{G}$. In the generalized setting with $K \geq p$, the proof works in the reversed way, where we iteratively identify the sink nodes[10] of $\mathcal{G}$.

In Section B.3.1, we introduce the concept of a *topological representation*: a representation $\tilde{U}$ of the data for which marginal distributions change in a way consistent with an assignment $\rho_1, \ldots, \rho_p$ of intervention targets. In Lemma 5, we show that under Assumptions 1 and 2, a topological representation is guaranteed to exist. In Lemma 6, we show that any topological representation is also topologically consistent in a natural way with the underlying representation $U$.

In Section B.3.2, we consider transforming a topological representation $\tilde{U}$ into a different topological representation $\hat{U}$. For any such representation $\hat{U} = \tilde{U}\hat{R}'$, we define an associated graph $\hat{\mathcal{G}}^{\hat{R}'}$. In Lemma 7, we show that picking $\hat{R}$ so that $\hat{\mathcal{G}}^{\hat{R}'}$ has the fewest edges will yield that $\hat{\mathcal{G}}^{\hat{R}} = \mathcal{TS}(\mathcal{G}_\tau)$.

Together, these results are used to prove Theorem 1: that we can identify $\mathcal{G}$ up to transitive closure.

### B.3.1 Topologically ordered representations

We begin by introducing the concept of a topological representation.

**Definition 2.** *Suppose $X = U\Lambda + b$. Let $\Pi \in \mathbb{R}^{p\times p}$ be a non-singular matrix, let $\tilde{U} = X\Pi$, and let $\rho_1, \ldots, \rho_p \in [K]$. We call $\tilde{U}$ a* topological representation *of $X$ with intervention targets $\rho_1, \ldots, \rho_p$ if the following two conditions are satisfied for all $j \in [p]$:*

*(Condition 1)* $\mathbb{P}(\tilde{U}_j) \neq \mathbb{P}^{I_{\rho_j}}(\tilde{U}_j)$.

*(Condition 2)* $\mathbb{P}(\tilde{U}_{1:j-1}C^\top) = \mathbb{P}^{I_{\rho_j}}(\tilde{U}_{1:j-1}C^\top)$ *and for any $C \in \mathbb{R}^{j-1}$.*

*Here, $\mathbb{P}(\tilde{U}), \mathbb{P}^{I_k}(\tilde{U})$ are the induced distributions for $\tilde{U}$ when $X \sim \mathbb{P}_X$ and $X \sim \mathbb{P}_X^{I_k}$, respectively.*

---

[10] A sink node is a node without children

The next result shows that a topological representation always exists. In particular, we show that a topological representation can be recovered simply be re-ordering the nodes of $\mathcal{G}$.

**Lemma 5.** *Suppose that Assumption 2 hold. Then, there exists a topological representation of $X$.*

*Proof.* Assume without loss of generality that $\mathcal{G}$ has topological order $\tau = (1, 2, ..., p)$, i.e., $i \to j \in \mathcal{G}$ only if $i < j$. Let $\Pi = \Lambda^{-1}$, then $\tilde{U} = U + \tilde{c}$ for constant vector $\tilde{c} = b\Pi$. Set $\rho_1, ..., \rho_p$ to be such that $T(I_{\rho_j}) = j$ for $j \in [p]$. Let $j \in [p]$ and $C \in \mathbb{R}^{j-1}$.

*Condition 1.* Since $I_{\rho_j}$ targets $U_j$, by Assumption 2, we have $\mathbb{P}(U_j) \neq \mathbb{P}^{I_{\rho_j}}(U_j)$, and thus $\mathbb{P}(\tilde{U}_j) \neq \mathbb{P}^{I_{\rho_j}}(\tilde{U}_j)$.

*Condition 2.* Since $I_{\rho_j}$ targets $U_j$ and $U_{1:j-1} \subset U_{[p] \setminus \overline{\mathrm{de}}_{\mathcal{G}}(j)}$, we have $\mathbb{P}(U_{1:j-1}C^\top) \neq \mathbb{P}^{I_{\rho_j}}(U_{1:j-1}C^\top)$, and thus $\mathbb{P}(\tilde{U}_{1:j-1}C^\top) \neq \mathbb{P}^{I_{\rho_j}}(\tilde{U}_{1:j-1}C^\top)$. $\square$

Now, we show that any topological representation is also consistent with the underlying representation $U$ up to some linear transformation which respects the topological ordering.

**Lemma 6.** *Suppose that Assumptions 2 hold. Let $\tilde{U} = X\Pi$ be a topological representation of $X$ and denote $\tilde{\Gamma} = \Lambda\Pi$. Then, there exists a topological ordering $\tau$ of $\mathcal{G}$ such that for any $j \in [p]$, we have that*

$$i < j \implies \tilde{\Gamma}_{\tau(j),i} = 0 \quad \text{and} \quad \tilde{\Gamma}_{\tau(j),j} \neq 0. \tag{9}$$

*Proof.* We prove by induction. Let $\tilde{c} = b\Pi$. Note that $\tilde{U} = U\tilde{\Gamma} + \tilde{c}$.

Base case.
Consider $I_{\rho_p}$. Let $T(I_{\rho_p}) = i$. We will show that $i$ must be a sink node in $\mathcal{G}$.

Suppose $i$ is not a sink node, and let $j \in \mathrm{mch}_{\mathcal{G}}(i)$. Since $\tilde{\Gamma}$ is nonsingular, $\mathrm{rank}(\mathrm{span}(\tilde{\Gamma}_{:,1}, ..., \tilde{\Gamma}_{:,p-1})) = p-1$. Therefore, we must have $\mathrm{span}(e_i, e_j) \cap \mathrm{span}(\tilde{\Gamma}_{:,1}, ..., \tilde{\Gamma}_{:,p-1}) \neq \{0\}$. Thus,

$$\tilde{\Gamma}\gamma^\top = ae_i^\top + be_j^\top \quad \text{for some } a, b \in \mathbb{R}, \gamma \in \mathbb{R}^p \text{ such that } a^2 + b^2 \neq 0, \gamma_p = 0$$

By Condition 2, we have that $\mathbb{P}(\tilde{U}\gamma^\top) = \mathbb{P}_{I_{\rho_p}}(\tilde{U}\gamma^\top)$. Since $\tilde{c}\gamma^\top$ is a constant, this implies that $\mathbb{P}(aU_i + bU_j) = \mathbb{P}^{I_{\rho_p}}(aU_i + bU_j)$. However, this contradicts Assumption 2. Thus, $i$ must be a sink node, which we denote by $\tau(p)$.

We also have that $\tilde{\Gamma}_{\tau(p),i} = 0$ for any $i < p$. Otherwise suppose $\tilde{\Gamma}_{\tau(p),i} \neq 0$, then $\tilde{U}_i = U\tilde{\Gamma}_{:,i} + h_i$ can be written as $\tilde{\Gamma}_{\tau(p),i} \cdot U_{\tau(p)} + U_S C^\top + \tilde{c}_i$ with $S = [p] \setminus (\{\tau(p)\}) = [p] \setminus \overline{\mathrm{de}}_{\mathcal{G}}(\tau(p))$. By Assumption 2, we have $\mathbb{P}(\tilde{U}_i) \neq \mathbb{P}^{I_{\rho_p}}(\tilde{U}_i)$, a contradiction to Condition 2.

Induction step.
Suppose that we have proven the statement for $q \leq p$. Denote the intervention targets of $I_{\rho_q}, \ldots, I_{\rho_p}$ as $\tau(q), \ldots, \tau(p)$, respectively. Let $K = [p] \setminus \{\tau(q), \ldots, \tau(p)\}$.

Consider $I_{\rho_{q-1}}$ with $T(I_{\rho_{q-1}}) = i$. Let $\mathcal{G}_q$ denote the graph $\mathcal{G}$ after removing the nodes $\tau(q), \ldots, \tau(p)$. We will show that $i$ must be a sink node in $\mathcal{G}_q$.

Suppose that $i$ is not a sink node $\mathcal{G}_q$ and let $j$ be a maximal child of $i$ in $\mathcal{G}_q$. Since $\tilde{\Gamma}_{[p] \setminus K, [q]} = 0$, $|K| = q$, and $\tilde{\Gamma}$ is nonsingular, we have that $\tilde{\Gamma}_{K, [q]}$ is nonsingular. Thus, as above,

$$\tilde{\Gamma}\gamma^\top = ae_{i(q)}^\top + be_{j(q)}^\top \quad \text{for some } a, b \in \mathbb{R}, \gamma \in \mathbb{R}^p \text{ such that } a^2 + b^2 \neq 0, \gamma_q = \gamma_{q+1} = \ldots = \gamma_p = 0$$

where $e_{i(q)}, e_{j(q)}$ are indicator vectors in $\mathbb{R}^q$ with ones at positions of $i, j$ in $1, ..., p$ after removing $\tau(q+1), ..., \tau(p)$, respectively. Thus, by Condition 2, we have that $\mathbb{P}(\tilde{U}\gamma^\top) = \mathbb{P}^{I_{\rho_q}}(\tilde{U}\gamma^\top)$, which contradicts Assumption 2. Therefore $I_{\rho_q}$ targets a sink node of $\mathcal{G}_q$.

To show that $\tilde{\Gamma}_{\tau(q),i} = 0$ for any $i < q$, use $\tilde{\Gamma}_{\tau(k),i} = 0$ for all $k \geq q$ and write $\tilde{U}_i = U\tilde{\Gamma}_{:,i} + \tilde{c}_i$ as $\tilde{\Gamma}_{\tau(q-1),i} \cdot U_{\tau(q-1)} + U_S C^\top + \tilde{c}_i$ with $S = [p] \setminus \{\tau(q-1), \tau(q), ..., \tau(p)\} \subset [p] \setminus \overline{\mathrm{de}}_{\mathcal{G}}(\tau(q))$. By Assumption 2, we have $\mathbb{P}(\tilde{U}_i) \neq \mathbb{P}^{I_{\rho_q}}(\tilde{U}_i)$ if $\tilde{\Gamma}_{\tau(q),i} \neq 0$, a contradiction to Condition 2.

By induction, we have thus proven that the solution to Condition 1 and Condition 2 satisfies $i < j \Rightarrow \tilde{\Gamma}_{\tau(j),i} = 0$. Therefore $\tilde{\Gamma}_{\tau,:}$ is upper triangular. Since it is also non-singular, it must hold that $\tilde{\Gamma}_{\tau(j),j} \neq 0$. Thus Eq. (9) holds for some unknown $\tau$. Furthermore, the proof shows that $I_{\rho_1}, ..., I_{\rho_p}$ target $U_{\tau(1)}, ..., U_{\tau(p)}$ respectively. $\qquad\square$

### B.3.2 Sparsest topological representation

In the section, we will introduce a graph associated to any topological representation. We consider picking a topological representation such that the associated graph is as sparse is possible, and we show that this choice recovers the underlying graph $\mathcal{G}$ up to transitive closure.

We begin by establishing the following property of a topological representation $\tilde{U}$, which relates ancestral relationships in the underlying graph $\mathcal{G}$ to changes in marginals of $\tilde{U}$.

**Proposition 1.** *Suppose that Assumptions 2 hold. Let $\tilde{U}$ be a topological representation with intervention targets $\rho_1, \ldots, \rho_p$.*

*Then, for any $i < j$ such that $\tau(j) \in \deg_{\mathcal{G}}(\tau(i))$, we must have*

$$\mathbb{P}(\tilde{U}_j) \neq \mathbb{P}^{I_{\rho_k}}(\tilde{U}_j) \quad \text{for some } i \leq k < j \quad \text{such that } \tau(k) \in \overline{\deg}_{\mathcal{G}}(\tau(i)).$$

*Proof.* By Lemma 6, Eq. (9), $\tilde{U}_j$ is a linear combination of $U_{\tau(1)}, ..., U_{\tau(j)}$ with nonzero coefficient of $U_{\tau(j)}$. Let $k_0$ be

> *(Case 1)* the largest such that $i \leq k_0 < j$ where $\tau(k_0) \in \overline{\deg}_{\mathcal{G}}(\tau(i))$ and the coefficient of $U_{\tau(k_0)}$ in $\tilde{U}_j$ is nonzero,
>
> *(Case 2)* $i$, if no $k_0$ satisfies Case 1.

Then let $k = k_0$ if $\tau(j) \notin \deg_{\mathcal{G}}(\tau(k_0))$; otherwise let $k$ be such that $\tau(k) \in \overline{\deg}_{\mathcal{G}}(\tau(k_0))$ and $\tau(j) \in \mathrm{mch}_{\mathcal{G}}(\tau(k))$ (such $k$ exists by considering the parent of $\tau(j)$ on the longest directed path from $\tau(k_0)$ to $\tau(j)$ in $\mathcal{G}$). Figure 8 illustrates the different scenarios for $k_0, k$. Note that we always have $\tau(k) \in \overline{\deg}_{\mathcal{G}}(\tau(i))$.

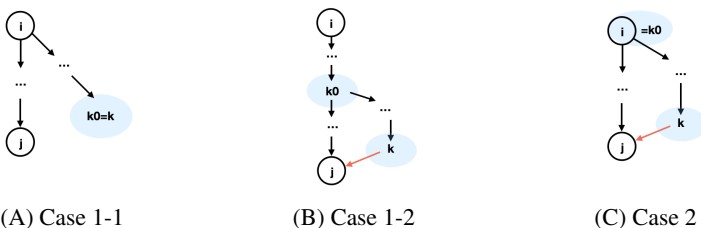

(A) Case 1-1      (B) Case 1-2      (C) Case 2

Figure 8: Illustration of $k_0, k$.

*(Case 1):* We first show that $\tilde{U}_j$ can be written as a linear combination of $U_{\tau(j)}, U_{\tau(k_0)}$ and $U_S$ for $S \subset [p] \setminus \overline{\deg}_{\mathcal{G}}(\tau(k_0))$ with nonzero coefficient for $U_{\tau(j)}, U_{\tau(k_0)}$. Consider an arbitrary $l \in [p]$. If the coefficient for $U_{\tau(l)}$ in $\hat{U}_j$ is nonzero, by Eq. (9), we have $l \leq j$. Also since $k_0$ is the largest, we have $l = k_0$ or $l = j$ or $l < k_0$ or $\tau(l) \notin \overline{\deg}_{\mathcal{G}}(\tau(i))$. If $l < k_0$, then by the topological order, it holds that $\tau(l) \notin \deg_{\mathcal{G}}(\tau(k_0))$. If $\tau(l) \notin \overline{\deg}_{\mathcal{G}}(\tau(i))$, since $\tau(k_0) \in \overline{\deg}_{\mathcal{G}}(\tau(i))$, it also holds that $\tau(l) \notin \overline{\deg}_{\mathcal{G}}(\tau(k_0))$. Therefore $\tilde{U}_j$ can be written as a linear combination of $U_{\tau(j)}, U_{\tau(k_0)}$ and $U_S$ with nonzero coefficient for $U_{\tau(j)}, U_{\tau(k_0)}$. Next, we show that $\mathbb{P}(\tilde{U}_j) \neq \mathbb{P}^{I_{\rho_k}}(\tilde{U}_j)$ by considering two subcases of Case 1.

If $\tau(j) \notin \deg_{\mathcal{G}}(\tau(k_0))$, then $k = k_0$ (illustrated in Figure 8A). Then $S \cup \{\tau(j)\} \subset [p] \setminus \overline{\deg}_{\mathcal{G}}(\tau(k))$. Therefore $\tilde{U}_j$ can be written as a linear combination of $U_{\tau(k)}$ and $U_{S'}$ for $S' \subset [p] \setminus \overline{\deg}_{\mathcal{G}}(\tau(k))$ with nonzero coefficient for $U_{\tau(k)}$. By Assumption 2, we have $\mathbb{P}(\tilde{U}_j) \neq \mathbb{P}^{I_{\rho_k}}(\tilde{U}_j)$.

If $\tau(j) \in \deg_{\mathcal{G}}(\tau(k_0))$, then since $\tau(k) \in \overline{\deg}_{\mathcal{G}}(\tau(k_0))$, we have $\tau(k_0) \in [p] \setminus \deg_{\mathcal{G}}(\tau(k))$ (illustrated in Figure 8B). Then we have $S \subset [p] \setminus \overline{\deg}_{\mathcal{G}}(\tau(k_0)) \subset [p] \setminus \deg_{\mathcal{G}}(\tau(k))$, and thus $S \cup \{\tau(k_0)\} \subset [p] \setminus \deg_{\mathcal{G}}(\tau(k))$. In fact, $S \cup \{\tau(k_0)\}$ is a subset of $[p] \setminus (\deg_{\mathcal{G}}(\tau(k)) \cup \{\tau(j)\})$, since $\tau(j) \in \deg_{\mathcal{G}}(\tau(k))$ by $\tau(k) \in \mathrm{pa}_{\mathcal{G}}(\tau(i))$ (definition of $k$). Therefore $\tilde{U}_j$ can be written as a linear combination of $U_{\tau(j)}$ and $U_{S'}$ for $S' \subset [p] \setminus (\deg_{\mathcal{G}}(\tau(k)) \cup \{\tau(j)\})$ with nonzero coefficient for $U_{\tau(j)}$. Since $\tau(j) \in \mathrm{mch}_{\mathcal{G}}(\tau(k))$ (definition of $k$), by Assumption 2, we have $\mathbb{P}(\tilde{U}_j) \neq \mathbb{P}^{I_{\rho_k}}(\tilde{U}_j)$, as $I_{\rho_k}$ targets $U_{\tau(k)}$.

*(Case 2):* In this case $k_0 = i$ (illustrated in Figure 8C). Then for any $l < j$ such that $\tau(l) \in \deg_{\mathcal{G}}(\tau(k))$, the coefficient of $U_{\tau(l)}$ in $\tilde{U}_j$ is zero. Otherwise since $\deg_{\mathcal{G}}(\tau(k)) \subset \deg_{\mathcal{G}}(\tau(k_0)) = \deg_{\mathcal{G}}(\tau(i))$, it holds that $\tau(l) \in \deg_{\mathcal{G}}(\tau(i))$, which by Eq. (11) implies $i < l < j$. Thus $l$ satisfies Case 1, a contradiction. Therefore, also by Eq. (11), $\tilde{U}_i$ can be written as a linear combination of $U_{\tau(j)}$ and $U_S$ with nonzero coefficient of $U_{\tau(j)}$, where $S \subset [p] \setminus \deg_{\mathcal{G}}(\tau(k))$.

Since $\tau(j) \in \deg_{\mathcal{G}}(\tau(i)) = \deg_{\mathcal{G}}(\tau(k_0))$, by definition of $k$, we have $\tau(j) \in \mathrm{mch}_{\mathcal{G}}(\tau(k))$. Note that $\tilde{U}_i$ can be written as a linear combination of $U_{\tau(j)}$ and $U_{S'}$ with nonzero coefficient of $U_{\tau(j)}$, where $S' \subset [p] \setminus (\deg_{\mathcal{G}}(k) \cup \{\tau(j)\})$. By Assumption 2, $\mathbb{P}(\tilde{U}_j) \neq \mathbb{P}^{I_{\rho_k}}(\tilde{U}_j)$, as $I_{\rho_k}$ targets $U_{\tau(k)}$.

Therefore, in both cases it holds that $\mathbb{P}(\tilde{U}_j) \neq \mathbb{P}^{I_{\rho_k}}(\tilde{U}_j)$. Since $i \leq k < j$ and $\tau(k) \in \overline{\deg}_{\mathcal{G}}(\tau(i))$, the claim is proven. $\qquad\square$

Now, we use marginal changes to define a graph associated to any topologically-ordered representation. We use Proposition 1 to show that picking the the topologically-ordered representation which yields the sparsest graph will recover the transitive closure of $\mathcal{G}$.

**Lemma 7.** *Let $\tilde{U}$ be a topological representation of $X$ with intervention targets $\rho_1, ..., \rho_p$. Let $\hat{R}' \in \mathbb{R}^{p \times p}$ be an invertible upper triangular and let $\hat{U} = \tilde{U}\hat{R}'$. Define the following:*

- *Let $\hat{\mathcal{G}}_0^{\hat{R}'}$ be the DAG such that $i \to j \in \hat{\mathcal{G}}_0$ if and only if $i < j \in [p]$ and $\mathbb{P}(\hat{U}_j) \neq \mathbb{P}^{I_{\rho_i}}(\hat{U}_j)$.*

- *Let $\hat{\mathcal{G}}^{\hat{R}'} = \mathcal{TS}(\hat{\mathcal{G}}_0^{R'})$.*

*Let $\hat{R}$ be such that $\hat{\mathcal{G}}^{\hat{R}}$ has the fewest edges over any choice of $\hat{R}'$. Then $\hat{\mathcal{G}}^{\hat{R}} = \mathcal{TS}(\mathcal{G}_\tau)$. We call $\hat{U}$ a* sparsest topological representation *of $X$.*

*Proof.*
Direction 1.
We first show that for any $\hat{R}'$,
$$\mathcal{TS}(\mathcal{G}_\tau) \subseteq \hat{\mathcal{G}}^{\hat{R}'}. \tag{10}$$

Let $i \to j \in \mathcal{TS}(\mathcal{G}_\tau)$, so $i < j$. By Proposition 1, we have $k$ such that $i \leq k < j$ with $\tau(k) \in \overline{\deg}_{\mathcal{G}}(\tau(i))$. By definition of $\hat{\mathcal{G}}_0^{\hat{R}'}$, we have $k \to j \in \hat{\mathcal{G}}_0^{\hat{R}'}$. Repeating this argument iteratively, we obtain a directed path from $i$ to $j$ in $\hat{\mathcal{G}}_0^{\hat{R}'}$. Thus, by definition of $\hat{\mathcal{G}}^{\hat{R}'}$, we have $\mathcal{TS}(\mathcal{G}_\tau) \subseteq \hat{\mathcal{G}}^{\hat{R}'}$.

Direction 2.
Now we give an example of $\hat{R}$ such that the constructed $\hat{\mathcal{G}}^{\hat{R}}$ satisfies

$$\hat{\mathcal{G}}^{\hat{R}} \subseteq \mathcal{TS}(\mathcal{G}_\tau).$$

Denote $\tilde{\Gamma}\hat{R} = \hat{\Gamma}$ and $\hat{c} = \tilde{c}\hat{R}$. Since $\hat{R}$ is upper-triangular and invertible, by Eq. (9), we have $\hat{U} = U\hat{\Gamma} + \hat{c}$, where
$$i < j \Rightarrow \hat{\Gamma}_{\tau(j),i} = 0 \quad \text{and} \quad \hat{\Gamma}_{\tau(j),j} \neq 0, \tag{11}$$
where $\tau$ is the topological order in Eq. (9). By Eq. (11), there exists an invertible upper-triangular matrix $R \in \mathbb{R}^{p \times p}$ such that $\hat{U} = (U\tilde{\Gamma} + \tilde{c})R = (U_{\tau(1)}, ..., U_{\tau(p)}) + c$ for some constant vector $c$. Now for $i < j \in [p]$, we have $i \to j \in \hat{\mathcal{G}}_0^R \Leftrightarrow \mathbb{P}(U_{\tau(j)}) \neq \mathbb{P}^{I_{\rho_i}}(U_{\tau(j)})$. Since $I_{\rho_i}$ targets $U_{\tau(i)}$, this would only be true when $\tau(j) \in \deg_{\mathcal{G}}(\tau(i))$. Therefore $i \to j \in \hat{\mathcal{G}}_0^{\hat{R}} \Rightarrow \tau(j) \in \deg_{\mathcal{G}}(\tau(i))$. Thus $\hat{\mathcal{G}}_0^{\hat{R}} \subseteq \mathcal{TS}(\mathcal{G}_\tau)$. As $\mathcal{TS}(\mathcal{G}_\tau)$ is a transitive closure, this means $\hat{\mathcal{G}}^{\hat{R}} = \mathcal{TS}(\hat{\mathcal{G}}_0^{\hat{R}}) \subseteq \mathcal{TS}(\mathcal{G}_\tau)$. $\qquad\square$

Analogously to Lemma 6, the following result shows that a sparsest topological representation is topologically consistent with $U$ in a stronger sense than a topological representation.

**Lemma 8.** *Let Assumption 2 hold. For $\tilde{\Gamma} \in \mathbb{R}^{p \times p}$ and $\tilde{c} \in \mathbb{R}^p$, let $\hat{U} = U\tilde{\Gamma} + \tilde{c}$ be a sparsest topological representation of $U$ with intervention targets $\rho_1, \ldots, \rho_p$. Let $\tau$ be a topological ordering of $\mathcal{G}$ such that*

$$i < j \Rightarrow \tilde{\Gamma}_{\tau(j),i} = 0 \quad \text{and} \quad \tilde{\Gamma}_{\tau(j),j} \neq 0. \tag{12}$$

*Then $\tilde{\Gamma}_{\tau(j),l} = 0$ for $\tau(l) \notin \deg_{\mathcal{G}}(\tau(j))$.*

*Proof.* For sake of contradiction, let $l, j \in [p]$. Without loss of generality, let $j$ be the largest value for which $\tilde{\Gamma}_{\tau(j),l} \neq 0$ and $\tau(l) \notin \deg_{\mathcal{G}}(\tau(j))$. By transitivity of $\deg_{\mathcal{G}}$ and the choice of $j$ as the largest value, there is no $j'$ such that $\tau(j') \in \deg_{\mathcal{G}}(\tau(j))$ and $\tilde{\Gamma}_{\tau(j'),l} \neq 0$. Therefore $\tilde{U}_l$ can be written as a linear combination of $U_{\tau(j)}$ and $U_S$ with nonzero coefficient of $U_{\tau(j)}$, where $S \subset [p] \setminus \overline{\deg}_{\mathcal{G}}(\tau(j))$.

By Assumption 2, we have $\mathbb{P}(\tilde{U}_l) \neq \mathbb{P}^{I_{\rho_j}}(\tilde{U}_l)$. Since $\tau(l) \notin \deg_{\mathcal{G}}(\tau(j))$ and $\hat{\mathcal{G}} = \mathcal{TS}(\mathcal{G}_\tau)$, we have $j \to l \notin \hat{\mathcal{G}}$, in which case $\mathbb{P}(\tilde{U}_l) \neq \mathbb{P}^{I_{\rho_{j'}}}(\tilde{U}_l)$ violates Condition 1, a contradiction. $\qquad\square$

### B.3.3 Proof of Theorem 1

**Theorem 1.** *Under Assumption 1 and Assumption 2 for $I_1, ..., I_K$, we can identify $\langle \hat{\mathcal{G}}, \hat{I}_1, ..., \hat{I}_K \rangle$, where $\hat{\mathcal{G}} = \mathcal{TS}(\mathcal{G}_\pi)$, and $\hat{I}_k = (I_k)_\pi$ for some permutation $\pi$.*

Here, we combine the results of the previous two sections to show that we can recover $\mathcal{G}$ up to transitive closure and permutation, and that we recover the intervention targets $I_1, \ldots, I_K$ up to the same permutation.

*Proof.* By Lemma 1 and Remark 1, we can assume, without loss of generality, that $p$ is known and that $X = f(U) = U\Lambda + b$ for some non-singular matrix $\Lambda$, as this can be identified from observational data $\mathcal{D}$.

By Lemma 7, we can identify a topological representation $\hat{U} = U\hat{\Gamma} + \hat{c}$ with intervention targets $\rho_1, ..., \rho_p \in [K]$, where $\hat{\Gamma} \in \mathbb{R}^{p \times p}$ and $\hat{c} \in \mathbb{R}^p$. Further, for some unknown topological ordering $\tau$ of $\mathcal{G}$, $\hat{\Gamma}$ satisfies Eq. (11), $T(I_{\rho_i}) = \tau(i)$ for $i \in [p]$, and we identify $\hat{\mathcal{G}} = \mathcal{TS}(\mathcal{G}_\tau)$.

Identifying additional intervention targets.
So far, we only guarantee that we identify the intervention targets for $I_{\rho_1}, \ldots, I_{\rho_p}$. Now, consider any $k \in [K] \setminus \{\rho_1, ..., \rho_p\}$. Let $l$ be such that $T_{\mathcal{G}}(I_k) = \tau(l)$. We now argue that $l$ can be identified as the smallest $l'$ in $[p]$ such that $\mathbb{P}(\hat{U}_{l'}) \neq \mathbb{P}^{I_k}(\hat{U}_{l'})$.

By Assumption 2, we have $\mathbb{P}(\hat{U}_l) \neq \mathbb{P}^{I_k}(\hat{U}_l)$, since $I_k$ targets $U_{\tau(l)}$ and $\hat{U}_l$ can be written as a linear combination of $U_{\tau(1)}, ..., U_{\tau(l)}$ with nonzero coefficient $U_{\tau(l)}$ (note that $U_{\tau(1)}, ..., U_{\tau(l-1)} \in [p] \subset \overline{\deg}_{\mathcal{G}}(\tau(l))$).

On the other hand, for $l' < l$, we have $\mathbb{P}(\hat{U}_{l'}) = \mathbb{P}^{I_k}(\hat{U}_{l'})$, since $\hat{U}_{l'}$ can be written as a linear combination of $U_{\tau(1)}, ..., U_{\tau(l')}$ and $\tau$ is the topological order. $\qquad\square$

### B.4 Proof of Theorem 2

In this section, we show that by introducing Assumption 3, we can go beyond recovering the transitive closure of $\mathcal{G}$, and we instead recover $\mathcal{G}$. We begin by establishing a basic fact about conditional independences in our setup.

**Claim 1.** *Under Assumption 1, let $\mathbf{A}, \mathbf{B}, \mathbf{C}, \mathbf{D}$ denote (potentially linear combinations of) components of $U$, and assume that $\mathbf{A} \perp\!\!\!\perp \mathbf{B} \mid \mathbf{C}, \mathbf{D}$ and $\mathbf{A} \perp\!\!\!\perp \mathbf{C} \mid \mathbf{B}, \mathbf{D}$. Then $\mathbf{A} \perp\!\!\!\perp \mathbf{B} \mid \mathbf{D}$.*

*Proof.* By Assumption 1, $\mathbb{P}_{\mathbf{A},\mathbf{B},\mathbf{C},\mathbf{D}}$ has positive measure on some full-dimensional set. By Proposition 2.1 of [56], $\mathbb{P}_{\mathbf{A},\mathbf{B},\mathbf{C},\mathbf{D}}$ is a graphoid, i.e., it obeys the intersection property. Invoking this property, we obtain $\mathbf{A} \perp\!\!\!\perp \mathbf{B} \mid \mathbf{D}$, as desired. $\qquad\square$

With this, we are ready to prove Theorem 2, which we recall here.

**Theorem 2.** *Under Assumptions 1,2,3, $\langle \mathcal{G}, I_1, ..., I_K \rangle$ is identifiable up to its CD-equivalence class.*

Note that, from Theorem 1, we have already identified the interventions $I_1, \ldots, I_K$ up to CD-equivalence for a permutation $\tau$. Thus, the only remaining result to show is that we identify $\mathcal{G}$ up to the same permutation.

In particular, we can again characterize the solution in terms of the sparsest solution.

**Theorem 2, Constructive.** *Let $\hat{U}$ be a sparsest topological representation of $X$ with intervention targets $\rho_1, ..., \rho_p$. Let $\bar{R}' \in \mathbb{R}^{p \times p}$ be an invertible upper triangular matrix, and let $\bar{U} = \hat{U}\bar{R}'$. Define the following:*

- *Let $\bar{\mathcal{G}}^{\bar{R}'}$ be the DAG such that $i \to j$ for $i < j \in [p]$, if and only if $\bar{U}_i \not\perp\!\!\!\perp \bar{U}_j \mid \bar{U}_1, \ldots, \bar{U}_{i-1}, \bar{U}_{i+1}, \ldots, \bar{U}_{j-1}$*

*Let $\bar{R}$ be such that $\bar{\mathcal{G}}^{\bar{R}}$ has the fewest edges over any choice of $\bar{R}'$. Then $\bar{\mathcal{G}}^{\bar{R}} = \mathcal{G}_\tau$ for $\tau$ satisfying Eq. (12).*

*Proof.* By Lemma 7, we have $\hat{U} = U\hat{\Gamma}$ for some matrix $\hat{\Gamma} \in \mathbb{R}^{p \times p}$ satisfying Eq. (9) under some topological order $\tau$ of $\mathcal{G}$. Further, we identify $\hat{\mathcal{G}} = \mathcal{TS}(\mathcal{G}_\tau)$ .

Denoting $\hat{\Gamma}\bar{R} = \bar{\Gamma}$ and $\bar{c} = \hat{c}\bar{R}$, by Lemma 8, we have $\bar{U} = U\bar{\Gamma} + \bar{c}$ with

$$
\begin{aligned}
&i < j \Rightarrow \bar{\Gamma}_{\tau(j),i} = 0 \quad \text{and} \quad \bar{\Gamma}_{\tau(j),j} \neq 0, \\
&\tau(l) \notin \deg_{\mathcal{G}}(\tau(j)) \Rightarrow \bar{\Gamma}_{\tau(j),l} = 0.
\end{aligned}
\tag{13}
$$

Direction 1.
First, we show that

$$\mathcal{G}_\tau \subseteq \bar{\mathcal{G}}^{\bar{R}}.$$

Assume on the contrary that there exists $\tau(i) \to \tau(j) \in \mathcal{G}$ such that $i \to j \notin \bar{\mathcal{G}}^{\bar{R}}$. By definition, we have $\bar{U}_i \perp\!\!\!\perp \bar{U}_j \mid \bar{U}_1, \ldots, \bar{U}_{i-1}, \bar{U}_{i+1}, \ldots, \bar{U}_{j-1}$. By Eq. (13), we know that we can retrieve $U_{\tau(1)}, ..., U_{\tau(i-1)}$ by linearly transforming $\bar{U}_1, ..., \bar{U}_{i-1}$; this implies $U_{\tau(i)} \perp\!\!\!\perp \bar{U}_j \mid U_{\tau(1)}, ..., U_{\tau(i-1)}, \bar{U}_{i+1}, ..., \bar{U}_{j-1}$. By subtracting terms in $U_{\tau(1)}, ..., U_{\tau(i-1)}$ from $\bar{U}_{i+1}, ..., \bar{U}_j$ and then subtracting terms $\bar{U}_l$ from $\bar{U}_{l+1}, ..., \bar{U}_j$ for $l = i+1, ..., j-1$, we have that

$$
\begin{aligned}
U_{\tau(i)} \perp\!\!\!\perp U_{\tau(j)} + c_j U_{\tau(i)} \mid U_{\tau(1)}, ..., U_{\tau(i-1)}, \\
U_{\tau(i+1)} + c_{i+1} U_{\tau(i)}, ..., U_{\tau(j-1)} + c_{j-1} U_{\tau(i)},.
\end{aligned}
\tag{14}
$$

for some $c_{i+1}, ..., c_j \in \mathbb{R}$. Since by Eq. (13) there is $\bar{\Gamma}_{\tau(i),l} = 0$ for any $\tau(l) \notin \deg_{\mathcal{G}}(\tau(i))$, this subtraction gives us $c_l = 0$ if $\tau(l) \notin \deg_{\mathcal{G}}(\tau(i))$.

Therefore let

$$
\begin{aligned}
\mathbf{A} &= U_{\tau(j)} + c_j U_{\tau(i)}, & \mathbf{B} &= U_{\tau(i)} \\
\mathbf{C} &= \{U_{\tau(l)} + c_l U_{\tau(i)}\}_{l \leq j-1, \tau(l) \notin \mathrm{pa}_{\mathcal{G}}(\tau(j))}, \text{and} & \mathbf{D} &= \{U_{\tau(l)} + c_l U_{\tau(i)}\}_{\tau(l) \in \mathrm{pa}_{\mathcal{G}}(\tau(j)) \setminus \{\tau(i)\}},
\end{aligned}
$$

where $c_1 = ... = c_{i-1} = 0$. There is $\mathbf{A} \perp\!\!\!\perp \mathbf{B} \mid \mathbf{C}, \mathbf{D}$.

On the other hand, since $\tau(i) \to \tau(j) \in \mathcal{G}$, i.e., $\tau(i) \in \mathrm{pa}_{\mathcal{G}}(\tau(j))$. We will now show that this implies $\mathbf{A} \perp\!\!\!\perp \mathbf{C} \mid \mathbf{B}, \mathbf{D}$. Starting with the local Markov property, we have for any $c_1, \ldots, c_{i-1}, c_{i+1}, \ldots, c_j$ that

$$
\begin{aligned}
&U_{\tau(j)} \perp\!\!\!\perp \{U_{\tau(l)}\}_{l \leq j-1, \tau(l) \notin \mathrm{pa}_{\mathcal{G}}(\tau(j))} \mid \{U_{\tau(l)}\}_{\tau(l) \in \mathrm{pa}_{\mathcal{G}}(\tau(j))} \\
\implies & U_{\tau(j)} + c_j U_{\tau(i)} \perp\!\!\!\perp \{U_{\tau(l)} + c_l U_{\tau(i)}\}_{l \leq j-1, \tau(l) \notin \mathrm{pa}_{\mathcal{G}}(\tau(j))} \mid \{U_{\tau(l)}\}_{\tau(l) \in \mathrm{pa}_{\mathcal{G}}(\tau(j))} \\
\implies & U_{\tau(j)} + c_j U_{\tau(i)} \perp\!\!\!\perp \{U_{\tau(l)} + c_l U_{\tau(i)}\}_{l \leq j-1, \tau(l) \notin \mathrm{pa}_{\mathcal{G}}(\tau(j))} \\
& \qquad \mid U_{\tau(i)}, \{U_{\tau(l)} + c_l U_{\tau(i)}\}_{\tau(l) \in \mathrm{pa}_{\mathcal{G}}(\tau(j)) \setminus \{\tau(i)\}}
\end{aligned}
\tag{15}
$$

where the first implication follows from the definition of conditional independence, and the second implication follows since $\{U_{\tau(l)}\}_{\tau(l)\in\mathrm{pa}_{\mathcal{G}}(\tau(j))}$ is a deterministic function of $U_{\tau(i)}, \{U_{\tau(l)} + c_l U_{\tau(i)}\}_{\tau(l)\in\mathrm{pa}_{\mathcal{G}}(\tau(j))\backslash\{\tau(i)\}}$.

Thus, by Claim 1, if $i \to j \notin \bar{\mathcal{G}}^{\bar{R}}$ and $\tau(i) \to \tau(j) \in \mathcal{G}$, then $\mathbf{A} \perp\!\!\!\perp \mathbf{B} \mid \mathbf{D}$, i.e.,

$$U_{\tau(i)} \perp\!\!\!\perp U_{\tau(j)} + c_j U_{\tau(i)} \mid \{U_{\tau(l)} + c_l U_{\tau(i)}\}_{l\in\mathrm{pa}_{\mathcal{G}}(\tau(j))\backslash\{\tau(i)\}}.$$

Since $c_l = 0$ for any $\tau(l) \notin \mathrm{de}_{\mathcal{G}}(\tau(i))$ and $\tau$ is the topological order, this can be further written as

$$U_{\tau(i)} \perp\!\!\!\perp U_{\tau(j)} + c_j U_{\tau(i)} \mid \{U_{\tau(l)}\}_{l\in\mathrm{pa}_{\mathcal{G}}(\tau(j))\backslash(S\cup\{\tau(i)\})}, \{U_{\tau(l)} + c_l U_{\tau(i)}\}_{l\in S},$$

where $S = \mathrm{pa}_{\mathcal{G}}(\tau(j)) \cap \mathrm{de}_{\mathcal{G}}(\tau(i))$, which violates Assumption 3. Therefore we must have $\mathcal{G}_\tau \subseteq \bar{\mathcal{G}}^{\bar{R}}$.

Direction 2.
There exists an invertible upper-triangular matrix $\bar{R} \in \mathbb{R}^{p\times p}$ such that $\bar{U} = \hat{U}\bar{R} = (U_{\tau(1)}, ..., U_{\tau(p)}) + \bar{c}$ for some constant vector $\bar{c}$. Note that clearly $\bar{U}$ satisfies Condition 1. Also for $i < j \in [p]$ such that $\tau(i) \to \tau(j) \notin \mathcal{G}$, by the Markov property and $\tau$ being the topological order, we have $\bar{U}_i \perp\!\!\!\perp \bar{U}_j \mid \bar{U}_1, ..., \bar{U}_{i-1}, \bar{U}_{i+1}, ..., \bar{U}_{j-1}$. Thus $\tau(i) \to \tau(j) \notin \mathcal{G} \Rightarrow i \to j \notin \bar{\mathcal{G}}$, and hence $\bar{\mathcal{G}} \subseteq \mathcal{G}_\tau$, which completes the proof. $\qquad\square$

**Remark 2.** *These proofs (Lemma 1, Theorem 1,2) together indicate that under Assumptions 1,2,3, we can identify $\langle\mathcal{G}, I_1, ..., I_K\rangle$ up to its CD-equivalence class by solving for the smallest $\hat{p}$, an encoder $\hat{g}: \mathbb{R}^n \to \mathbb{R}^{\hat{p}}, \hat{\mathcal{G}}$ and $\hat{I}_1, ..., \hat{I}_{\hat{K}}$ that satisfy*

(1) *there exists a full row rank polynomial decoder $\hat{f}(\cdot)$ such that $\hat{f} \circ \hat{g}(X) = X$ for all $X \in \mathcal{D} \cup \mathcal{D}^{I_1} \cup ... \cup \mathcal{D}^{I_K}$;*

(2) *the induced distribution on $\hat{U} := \hat{g}(X)$ by $X \in \mathcal{D}$ factorizes with respect to $\hat{\mathcal{G}}$;*

(3) *the induced distribution on $\hat{U}$ by $X \in \mathcal{D}^{I_k}$ where $k \in [K]$ changes the distribution of $\hat{U}_{T_{\hat{\mathcal{G}}}(\hat{I}_k)}$ but does not change the joint distribution of non-descendants of $\hat{U}_{T_{\hat{\mathcal{G}}}(\hat{I}_k)}$ in $\hat{\mathcal{G}}$;*

(4) *$[\hat{p}] \subseteq T_{\hat{\mathcal{G}}}(\hat{I}_1) \cup ... \cup T_{\hat{\mathcal{G}}}(\hat{I}_{\hat{K}})$;*

(5) *$\hat{\mathcal{G}}$ has topological order $1, ..., \hat{p}$;*

(6) *the transitive closure $\mathcal{TS}(\hat{\mathcal{G}})$ of the DAG $\hat{\mathcal{G}}$ is the sparsest amongst all solutions that satisfy (1)-(5);*

(7) *the DAG $\hat{\mathcal{G}}$ is the sparsest amongst all solutions that satisfy (1)-(6);*

*We will use these observations in Appendix E to develop a discrepancy-based VAE and show that it is consistent in the limit of infinite data.*

*Proof.* We first show that there is a solution to (1)-(7). For this, it suffices to show that there is a solution to (1)-(5). Then since $\hat{p}$ and $\hat{\mathcal{G}}$ are discrete, one can find the solution to (1)-(7) by searching amongst all solutions to (1)-(5) such that $\hat{p}$ is the smallest and (6)-(7) are satisfied. Assume without loss of generality that $\mathcal{G}$ has topological order $1, ..., p$. Then $\hat{p} = p$, $\hat{g} = f^{-1}$, $\hat{\mathcal{G}} = \mathcal{G}$, and $\hat{I}_k = I_k$ for $k \in [K]$ satisfy (1)-(5).

Next we show that any solution must recover $\hat{p} = p$ and $\langle\hat{\mathcal{G}}, \hat{I}_1, ..., \hat{I}_K\rangle$ that is in the same CD equivalence class as $\langle\mathcal{G}, I_1, ..., I_K\rangle$. Since we solve for the smallest $\hat{p}$, the former paragraph also implies that $\hat{p} \leq p$. By the proof of Lemma 1, (1) guarantees that $\hat{p} \geq p$. Therefore it must hold that $\hat{p} = p$.

Since we solve for the sparsest transitive closure, the first paragraph implies that $\mathcal{TS}(\hat{\mathcal{G}}) \subset \mathcal{TS}(\mathcal{G})$. Also by the proof of Lemma 1, $\hat{U}$ can be written as an invertible linear mixing of $U$. Then (3)-(5) guarantee that Condition 1 and Condition 2 in Step 1 in the proof of Theorem 1 are satisfied. Then by the proof of Step 2 in the proof of Theorem 1, we have $\mathcal{TS}(\mathcal{G}) \subset \mathcal{TS}(\hat{\mathcal{G}})$. Therefore, it must hold that $\mathcal{TS}(\hat{\mathcal{G}}) = \mathcal{TS}(\mathcal{G})$.

Lastly, by (2) and (5), we obtain that $\hat{\mathcal{G}}$ satisfies Condition 1 and Condition 2 in Theorem 2. Therefore by the proof of Theorem 2, we obtain $\mathcal{G} \subset \hat{\mathcal{G}}$. Again, by the first paragraph and the fact that the sparsest transitive closure satisfies $\mathcal{TS}(\hat{\mathcal{G}}) = \mathcal{TS}(\mathcal{G})$, we obtain that the sparsest $\hat{\mathcal{G}}$ with this transitive closure must satisfy $\hat{\mathcal{G}} \subset \mathcal{G}$, and thus $\hat{\mathcal{G}} = \mathcal{G}$. With this result, it is easy to see that $\hat{I}_k = I_k$ for all $k \in [K]$, as $I_k$ changes the distribution of $\hat{U}_{T_{\mathcal{G}}(I_k)}$ but does not change the joint distribution of $\hat{U}_{[p] \setminus \overline{\mathrm{de}}_{\mathcal{G}}(T_{\mathcal{G}}(I_k))}$.

Therefore we can recover $p$ and the CD equivalence class of $\langle \mathcal{G}, I_1, ..., I_K \rangle$ by solving (1)-(7). Note that this proof assumes the topological order of $\mathcal{G}$ is $1, ..., p$, and therefore it does not violate the fact that $\mathcal{G}, I_1, ..., I_K$ cannot be recovered exactly. $\qquad\square$

## B.5   Proof of Theorem 3

Now, we will show that recovering $\langle U, \mathcal{G}, I_1, \ldots, I_K \rangle$ up to Theorem 1 is sufficient for predicting the effect of combinatorial interventions..

**Theorem 3.** *Letting $\langle \hat{U}, \hat{\mathcal{G}}, \hat{I}_1, ..., \hat{I}_K \rangle$ be the solution identified in the proof of Theorem 1. Then the interventional distribution $\mathbb{P}^{\mathcal{I}}$ for any combinatorial intervention $\mathcal{I} \subset \{I_1, ..., I_K\}$ is given by Eq. (2), i.e., we can generate samples $X$ from the distribution $X = f(U), U \sim \mathbb{P}^{\mathcal{I}}$.*

*Proof.* Since $\mathcal{I}$ contains interventions with different intervention targets, for each $i \in [p]$, we can define $\mathbb{P}^{\hat{\mathcal{I}}}(\hat{U}_i \mid \hat{U}_{\mathrm{pa}_{\hat{\mathcal{G}}}(i)})$ as $\mathbb{P}^{\hat{I}_k}(\hat{U}_i \mid \hat{U}_{\mathrm{pa}_{\hat{\mathcal{G}}}(i)})$ if $i = T_{\hat{\mathcal{G}}}(\hat{I}_k)$ for some $I_k \in \mathcal{I}$ and otherwise $\mathbb{P}(\hat{U}_i \mid \hat{U}_{\mathrm{pa}_{\hat{\mathcal{G}}}(i)})$. Using this definition, we define the joint distribution of $\hat{U}$ as $\mathbb{P}^{\hat{\mathcal{I}}}(\hat{U}) = \prod_{i=1}^{p} \mathbb{P}^{\hat{\mathcal{I}}}(\hat{U}_i \mid \hat{U}_{\mathrm{pa}_{\hat{\mathcal{G}}}(i)})$. In the following we show that $\mathbb{P}^{\hat{\mathcal{I}}}(\hat{U}) = \mathbb{P}^{\mathcal{I}}(U)$ in the sense that $\mathbb{P}^{\hat{\mathcal{I}}}(\hat{U} = \hat{f}^{-1}(x)) = \mathbb{P}^{\mathcal{I}}(U = f^{-1}(x))$ for all $x \in \mathbb{R}^n$.

Our proof combines the following equalities. For any $i \in [p]$, we have

*Equality 1:* $\mathbb{P}^{\hat{\mathcal{I}}}(\hat{U}_i \mid \hat{U}_{\mathrm{pa}_{\hat{\mathcal{G}}}(i)}) = \mathbb{P}^{\hat{\mathcal{I}}}(\hat{U}_i \mid \hat{U}_{\mathrm{an}_{\hat{\mathcal{G}}}(i)})$,

*Equality 2:* $\mathbb{P}^{\hat{\mathcal{I}}}(\hat{U}_i \mid \hat{U}_{\mathrm{an}_{\hat{\mathcal{G}}}(i)}) = \mathbb{P}^{\mathcal{I}}(U_i \mid U_{\mathrm{an}_{\mathcal{G}}(i)})$,

*Equality 3:* $\mathbb{P}^{\mathcal{I}}(U_i \mid U_{\mathrm{an}_{\mathcal{G}}(i)}) = \mathbb{P}^{\mathcal{I}}(U_i \mid U_{\mathrm{pa}_{\mathcal{G}}(i)})$.

*Proof of Equality 1.* This follows by definition of $\hat{\mathcal{G}}$, since it is transitively closed, we have $\mathrm{pa}_{\hat{\mathcal{G}}}(i) = \mathrm{an}_{\hat{\mathcal{G}}}(i)$.

*Proof of Equality 2.* By similar arguments below Eq. (13), we have $\hat{U} = U\hat{\Gamma} + \hat{c}$ for an invertible matrix $\hat{\Gamma}$, where $\hat{\Gamma}_{\tau(j),l} = 0$ for any $\tau(l) \notin \mathrm{de}_{\mathcal{G}}(\tau(j))$. Therefore we can recover $U_{\mathrm{an}_{\mathcal{G}}(i)}$ by linear transforming $U\hat{\Gamma}_{:,\mathrm{an}_{\mathcal{G}}(i)}$ and vise versa. We can also recover $U_i$ by subtracting linear terms of $U_{\mathrm{an}_{\mathcal{G}}(i)}$ from $U\hat{\Gamma}_{:,i}$.

Note also, since $\mathcal{TS}(\mathcal{G}) = \mathcal{TS}(\hat{\mathcal{G}})$, there must be $\mathrm{an}_{\hat{\mathcal{G}}}(i) = \mathrm{an}_{\mathcal{G}}(i)$. Thus

$$
\begin{aligned}
\mathbb{P}^{\hat{\mathcal{I}}}(\hat{U}_i \mid \hat{U}_{\mathrm{an}_{\hat{\mathcal{G}}}(i)}) &= \mathbb{P}^{\mathcal{I}}(U\hat{\Gamma}_{:,i} \mid U\hat{\Gamma}_{:,\mathrm{an}_{\hat{\mathcal{G}}}(i)}) \\
&= \mathbb{P}^{\mathcal{I}}(U\hat{\Gamma}_{:,i} \mid U\hat{\Gamma}_{:,\mathrm{an}_{\mathcal{G}}(i)}) \\
&= \mathbb{P}^{\mathcal{I}}(U\hat{\Gamma}_{:,i} \mid U_{\mathrm{an}_{\mathcal{G}}(i)}) = \mathbb{P}^{\mathcal{I}}(U_i \mid U_{\mathrm{an}_{\mathcal{G}}}(i)).
\end{aligned}
$$

*Proof of Equality 3.* Follows from the Markov property on $U$.

Combining these equalities, we have $\mathbb{P}^{\hat{\mathcal{I}}}(\hat{U}_i \mid \hat{U}_{\mathrm{pa}_{\hat{\mathcal{G}}}(i)}) = \mathbb{P}^{\mathcal{I}}(U_i \mid U_{\mathrm{pa}_{\mathcal{G}}(i)})$ for all $i \in [p]$. Thus $\mathbb{P}^{\hat{\mathcal{I}}}(\hat{U}) = \prod_{i=1}^{p} \mathbb{P}^{\hat{\mathcal{I}}}(\hat{U}_i \mid \hat{U}_{\mathrm{pa}_{\hat{\mathcal{G}}}(i)}) = \prod_{i=1}^{p} \mathbb{P}^{\mathcal{I}}(U_i \mid U_{\mathrm{pa}_{\mathcal{G}}(i)}) = \mathbb{P}^{\mathcal{I}}(U)$. Therefore the procedure in Section 4.4 generates $X$ from the same distribution as $X = f(U), U \sim \mathbb{P}^{\mathcal{I}}$. $\qquad\square$

# C  Details on Discrepancy-based VAE

In previous sections, we have shown that the data-generating process in Section 2 is identifiable up to equivalence classes. However, the proofs (Appendix A, B) do not lend themselves to an algorithmically efficient approach to learning the latent causal variables from data. Therefore, we propose a discrepancy-based VAE in Section 5, which inherits scalable tools of VAEs that can in principle learn flexible deep latent-variable models. In this framework, Eq. (3) can be computed and optimized efficiently using the reparametrization trick [29] and gradient-based optimizers.

## C.1  Maximum Mean Discrepancy

We recall the definition of the maximum mean discrepancy measure between two distributions, and its empirical counterpart.

**Definition 3.** *Let $k$ be a positive definite kernel function and let $\mathcal{H}$ be the reproducing kernel Hilbert space defined by this kernel. Given distributions $\mathbb{P}$ and $\mathbb{P}'$, we define*

$$\mathrm{MMD}(\mathbb{P}, \mathbb{P}') := \sup_{f \in \mathcal{H}} \left( \mathbb{E}_{\mathbb{P}}[f(X)] - \mathbb{E}_{\mathbb{P}'}[f(X)] \right)$$

The following empirical counterpart is an unbiased estimate of the squared MMD, see Lemma 6 of [18].

**Definition 4.** *Let $k$ be a positive definite kernel. Let $\{X_{(i)}\}_{i=1}^m$ be samples from $\mathbb{P}$ and $\{X'_{(i)}\}_{i=1}^m$ be samples from $\mathbb{P}'$. We define*

$$\widehat{\mathrm{MMD}}^2(\{X_{(i)}\}_{i=1}^m, \{X'_{(i)}\}_{i=1}^m) = \frac{1}{m(m-1)} \sum_{i=1}^m \sum_{j \neq i} k(X_i, X_j) + \frac{1}{m(m-1)} \sum_{i=1}^m \sum_{j \neq i} k(X'_i, X'_j)$$
$$- \frac{2}{m^2} \sum_{i=1}^m \sum_{j=1}^m k(X_i, X'_j)$$

## C.2  Discrepancy VAE Details

We walk through the details of this model in this section, where we illustrate it using two types of interventions, namely do interventions and shift interventions.

**Noiseless vs. Noisy Measurement Model with General SCMs.** Recall that each latent causal variable $U_i$ is a function of its parents in $\mathcal{G}$ and an exogenous noise term $Z_i$. All the $Z_i$'s are mutually independent. The overall model can be defined (recursively) as

$$\begin{aligned} U_j &= s_j(U_{\mathrm{pa}_{\mathcal{G}}(j)}, Z_j), \\ X &= f(U_1, ..., U_p). \end{aligned} \tag{16}$$

In particular, there exists a function $s_{\varnothing}^{\mathrm{full}}$ such that $U = s_{\varnothing}^{\mathrm{full}}(Z)$. We model each intervention $I$ as a set of intervention targets $T(I)$ and a vector $a^I$. Under $I$, the observations $X$ are generated by

$$\begin{aligned} U_j^I &= \begin{cases} s_j(U_{\mathrm{pa}_{\mathcal{G}}(j)}^I, Z_j) \mathbb{1}_{j \notin T(I)} + a_j^I \mathbb{1}_{j \in T(I)}, & \text{for do intervention}, \\ s_j(U_{\mathrm{pa}_{\mathcal{G}}(j)}^I, Z_j) + a_j^I \mathbb{1}_{j \in T(I)}, & \text{for shift intervention}, \end{cases} \\ X^I &= f(U_1^I, ..., U_p^I). \end{aligned} \tag{17}$$

As above, there exists a function $s_I^{\mathrm{full}}$ such that $U^I = s_I^{\mathrm{full}}(Z)$. Note that here we assume that the measurements (sometimes called "observations" in the literature[11]) $X$ are *noiseless*. Our theoretical results are built upon noiseless measurements. In practice, however, one can consider the *noisy* measurement model in which $X = f(U) + \epsilon$ (resp. $X^I = f(U^I) + \epsilon$), where $\epsilon$ is some measurement noise independent of $U$.

We leave as future work to prove consistency under the noisy measurement model. [28] established identifiability results of the noisy measurement model, when the latent variables conditioned on

---

[11] We use "measurements" to distinguish from the observational distribution defined for $U$.

additionally observed variables follow a factorized distribution in an exponential family. Their techniques can be potentially used to generalize our results to the noisy measurement model; however, further assumptions on the mechanisms $s_i$'s will be needed.

**Discrepancy-based VAE.** We use one *decoder*

$$p_\theta(X|U)$$

parameterized by $\theta$ to approximate both $X = f(U)$ and $X^I = f(U^I)$ in the noiseless measurement model (or $X = f(U) + \epsilon$ and $X^I = f(U^I) + \epsilon$ in the noisy measurement model). As for the encoder, we do not directly learn the posteriors $\mathbb{P}(U|X)$ and $\mathbb{P}(U^I|X^I)$. Instead, we approximate one posterior $\mathbb{P}(Z|X)$ and then use Eq. (16), (17) to transform $Z$ into $U, U^I$ respectively. This is done by two *encoders* for $Z$ and $(T(I), a^I)$ parameterized by $\phi$ and denoted as

$$q_\phi(Z|X), (T_\phi(I), a_\phi(I))$$

The dimension $p$ of $Z$ is set as a hyperparameter. Note that the procedure of learning a posterior $\mathbb{P}(Z|X)$ in the observational distribution and then mapping to $U^I$ using Eq. (17) can be regarded as learning the counterfactual posterior of $\mathbb{P}(U^I|X)$.

In the following, to better distinguish data from observational and interventional distributions, we use $X^\varnothing, U^\varnothing$ instead of $X, U$ to denote samples generated by Eq. (16). After encoding $X^\varnothing$ and $I$ into $Z$ and $(T(I), a^I)$ respectively, we parameterize the causal mechanisms $s_j$'s in Eq. (16), (17) as neural networks (e.g., multi-layer perceptrons or linear layers). We absorb the paramterizations of $s_j$'s into $\theta$ and denote

$$p_{\theta,\varnothing}(X^\varnothing|Z) = p_\theta\big(X^\varnothing \mid U^\varnothing = s_\varnothing^{\text{full}}(Z)\big),$$
$$p_{\theta,I}(X^I|Z) = p_\theta\big(X^I \mid U^I = s_I^{\text{full}}(Z)\big).$$

Note that in implementation, to make sure $U_j$ only depends on its parents $U_{\text{pa}_\mathcal{G}(j)}$, one can train an adjacency matrix $A$ that is upper-triangular up to permutations and then apply any layers after individual rows of matrix $U \otimes A$[12]. Since identifiability can be only up to permutations of latent nodes, one can simply use an upper-triangular adjacency matrix $A$.

## D  Lower Bound to Paired Log-Likelihood

In this section, we consider the *paired* setting, in which we have access to samples from the joint distribution $\mathbb{P}(X^\varnothing, X^I)$. To discuss counterfactual pairs, we must introduce structure beyond the structure described in Section 2. In particular, in the observational setting, assume that the latent variables $U^\varnothing$ are generated from a structural causal model with exogenous noise terms $Z$. This implies that there is a function $g_\varnothing$ such that $U^\varnothing = g_\varnothing(Z)$. Similarly, under intervention $I$, assume there is a function $g_I$ such that $U^I = g_I(Z)$. Then, given a distribution $\mathbb{P}(Z)$, the joint distribution $\mathbb{P}(X^\varnothing, X^I)$ is simply the induced distribution under the maps $X^\varnothing = f(U^\varnothing)$ and $X^I = f(U^I)$.

Since $X^\varnothing$ and $X^I$ are independent conditioned on $Z$, we have

$$\log \mathbb{P}(X^\varnothing, X^I) \geq \mathbb{E}_{\mathbb{P}(X^\varnothing, X^I)}\Big[\mathbb{E}_{q_\phi(Z|X^\varnothing)} \log p_{\theta,\varnothing}(X^\varnothing \mid Z) + \mathbb{E}_{q_\phi(Z|X^\varnothing)} \log p_{\theta,I}(X^I|Z)$$
$$- D_{\text{KL}}\big(q_\phi(Z|X^\varnothing)\|p(Z)\big)\Big] \quad (18)$$

We have the following result on the loss function in Eq. (3).

**Proposition 2.** *Let $k$ be a Gaussian kernel with width $\epsilon$, i.e., $k(x,y) = \exp\left(-\frac{\|x-y\|_2^2}{2\epsilon^2}\right)$. Let $p_{\theta,I}(X^I \mid U)$ be Gaussian with mean $\mu_\theta^I(U)$ and a fixed variance $\sigma^2$. Then, for $\epsilon$ sufficiently large, for $\alpha$ given in the proof, and for some constant $c$ depending only on $\sigma$ and data dimension $d$,*

$$\mathbb{E}_{\mathbb{P}(X^\varnothing, X^I)}\big[\mathbb{E}_{q_\phi(Z|X^\varnothing)} \log p_{\theta,I}(X^I|Z)\big] \geq -\alpha \cdot \text{MMD}\big(p_{\theta,I}(X^I), \mathbb{P}^I(X^I)\big) + c.$$

*Thus, up to an additive constant, $\mathcal{L}_{\theta,\phi}^{\alpha,1,0}$ lower bounds the paired-data ELBO in Eq.(18) and by extension the paired-data log-likelihood $\log \mathbb{P}(X^\varnothing, X^I)$.*

---

[12]Here $\otimes$ denotes the Kronecker product.

*Proof.* By the choice of a Gaussian distribution for $p_{\theta,I}(X^I \mid U)$, we have

$$\log p_{\theta,I}(X^I \mid Z) = \log p_\theta(X^I \mid U^I = s_I^{\text{full}}(Z)) = c - \frac{1}{2\sigma^2}\|X^I - \mu_\theta^I(U)\|_2^2, \tag{19}$$

where $c$ is a constant depending only on $\sigma$ and data dimension $d$. Let $\{(X_{(i)}^\varnothing, X_{(i)}^I)\}_{i=1}^m$ be independent and identically distributed according to $\mathbb{P}(X^\varnothing, X^I)$. Then

$$\mathbb{E}_{\mathbb{P}(X^\varnothing, X^I)}\left[\mathbb{E}_{q_\phi(Z|x^{(0)})}[\log p_{\theta,I}(X^I|Z)]\right]$$

$$= \mathbb{E}_{\mathbb{P}(X^\varnothing, X^I)}\left[\mathbb{E}_{q_\phi(Z_i|X_i^\varnothing)}\left[\frac{1}{m}\sum_{i=1}^m \log p_{\theta,I}(X_i^I|Z_i)\right]\right]$$

$$= c - \frac{1}{2\sigma^2}\mathbb{E}_{\mathbb{P}(X^\varnothing, X^I)}\left[\mathbb{E}_{q_\phi(Z_{(i)}|X_{(i)}^\varnothing)}\left[\frac{1}{m}\sum_{i=1}^m \|X_{(i)}^I - \mu_\theta^I(U_{(i)})\|_2^2\right]\right]$$

Now, for the empirical MMD, we have

$$\widehat{\text{MMD}}^2\left(\{X_{(i)}^I\}_{i=1}^m, \{\hat{X}_{(i)}^I\}_{i=1}^m\right)$$

$$= \frac{1}{m(m-1)}\sum_{i=1}^m\sum_{j\neq i}\exp\left(-\frac{\|X_{(i)}^I - X_{(j)}^I\|_2^2}{2\epsilon^2}\right) + \frac{1}{m(m-1)}\sum_{i=1}^m\sum_{j\neq i}\exp\left(-\frac{\|\hat{X}_{(i)}^I - \hat{X}_{(j)}^I\|_2^2}{2\epsilon^2}\right)$$

$$- \frac{2}{m^2}\sum_{i=1}^m\sum_{j=1}^m\exp\left(-\frac{\|X_{(i)}^I - \hat{X}_{(j)}^I\|_2^2}{2\epsilon^2}\right)$$

$$\geq -\frac{2}{m^2}\sum_{i=1}^m\sum_{j=1}^m\exp\left(-\frac{\|X_{(i)}^I - \hat{X}_{(j)}^I\|_2^2}{2\epsilon^2}\right)$$

$$\geq -2 + \frac{1}{2m^2\epsilon^2}\sum_{i=1}^m\sum_{j=1}^m\|X_{(i)}^I - \hat{X}_{(j)}^I\|_2^2$$

$$\geq -2 + \frac{1}{2m^2\epsilon^2}\sum_{i=1}^m\|X_{(i)}^I - \hat{X}_{(i)}^I\|_2^2,$$

where we have used the positivity of the exponential function and for the penultimate inequality used the fact that $\epsilon$ is large enough and that $e^{-x} \leq 1 - x/2$ for $x$ sufficiently small. Substituting into (20) yields the theorem, with $\alpha = \frac{1}{2m\sigma^2\epsilon^2}$. $\qquad\square$

# E  Consistency of Discrepancy-based VAE

We consider Discrepancy-based VAE described in the last section. Suppose the conditions in Theorem 2 is satisfied by the ground-truth model, i.e., it is possible to identify CD-equivalence class in theory.

## E.1  CD-Equivalence Class

**Theorem 4.** *Let $X^\varnothing$, $X^{I_1}$, ..., $X^{I_K}$ be generated as in Section 2. Suppose that Assumptions 1, 2, and 3 hold. Define*

$$M_1 = argmin_{\theta,\phi}\mathcal{L}_{\theta,\phi}$$

$$M_2 = argmin_{\theta,\phi\in M_1}|\mathcal{TS}(\mathcal{G}_\theta)|$$

$$\hat{\theta}, \hat{\phi} \in argmin_{\theta,\phi\in M_2}|\mathcal{G}_\theta|$$

*for $\mathcal{L}_{\theta,\phi}$ defined in Equation 3. Further, suppose that the VAE prior $p(Z)$ is equal to the true distribution over $Z$, that $p_\theta(X \mid U)$ and $q_\phi(Z \mid X)$ are Dirac distributions. Let $\langle \hat{U}, \hat{\mathcal{G}}, \hat{I}_1, \ldots, \hat{I}_K \rangle$ be the solution induced by $\hat{\theta}, \hat{\phi}$.*

*Then $\langle \hat{U}, \hat{\mathcal{G}}, \hat{I}_1, ..., \hat{I}_K \rangle$ is CD-equivalent to $\langle U, \mathcal{G}, I_1, ..., I_K \rangle$.*

*Proof.* Note that the parameterization of $s_j$, $\mathcal{G}$, and the induced distributions of $U$ through prior $p(Z)$ using Eq. (16), (17) satisfy (2),(3) and (5) in Remark 2.

The first two terms in combined in Eq. (18) satisfy

$$\mathbb{E}_{\mathbb{P}(X^\varnothing)} \left[ \mathbb{E}_{q_\phi(Z|X^\varnothing)} \log p_{\theta,\varnothing}(X^\varnothing|Z) - D_{KL}\big(q_\phi(Z|X^\varnothing)\|p(Z)\big) \right]$$

$$= \mathbb{E}_{\mathbb{P}(X^\varnothing)} \left[ \log p_{\theta,\varnothing}(X^\varnothing) - D_{KL}\big(q_\phi(Z|X^\varnothing)\|p_{\theta,\varnothing}(Z|X^\varnothing)\big) \right]$$

$$\leq \mathbb{E}_{\mathbb{P}(X^\varnothing)} \log p_{\theta,\varnothing}(X^\varnothing)$$

$$= \mathbb{E}_{\mathbb{P}(X^\varnothing)} \log \mathbb{P}(X^\varnothing) - D_{KL}\big(p_{\theta,\varnothing}(X^\varnothing)\|\mathbb{P}(X^\varnothing)\big)$$

$$\leq \mathbb{E}_{\mathbb{P}(X^\varnothing)} \log \mathbb{P}(X^\varnothing),$$

where the equality holds if and only if $q_\phi(Z|X^\varnothing) = p_{\theta,\varnothing}(Z|X^\varnothing)$ and $p_{\theta,\varnothing}(X^\varnothing) = \mathbb{P}(X^\varnothing)$. On the other hand, since $\mathrm{MMD}(\cdot, \cdot)$ is a valid measure between distributions, we have

$$-\mathrm{MMD}\left( \mathbb{P}_{\theta,\phi}\left( \hat{X}^{\hat{I}_k} \right), \mathbb{P}(X^{I_k}) \right) \leq 0,$$

where the inequality is satisfied with equality if and only if $\hat{X}^{\hat{I}_k}$ and $X^{I_k}$ are equal in distribution.

Therefore if the learned intervention targets of $I_1, ..., I_K$ cover $[\hat{p}]$ and the minimum loss function is not larger that for $\hat{p} = K$, we have the solution satisfy (1)-(5) in Remark 2. Since $\mathcal{G}$ has the sparsest transitive closure and $\mathcal{G}$ is the sparsest with this transitive closure, (6)-(7) in Remark 2 are also satisfied. Therefore Remark 2 guarantees the smallest $\hat{p} \leq K$ satisfying the conditions recovers the CD-equivalence class. $\square$

Note that in practice, it can be hard to ensure that the gradient-based approach returns a DAG $\mathcal{G}$ that has the sparsest transitive closure and is simultaneously the sparsest DAG with this transitive closure. We instead search for sparser DAGs $\mathcal{G}$ by penalizing its corresponding adjacency in Eq. (3).

### E.2 Consistency for Multi-Node Interventions

Theorem 3 guarantees that in an SCM with additive noises where interventions modify the exogenous noises, if the CD equivalence can be identified, we can extrapolate to unseen combinations of interventions with different intervention targets. In fact, for certain types of interventions, extrapolation to unseen combinations of any interventions is possible. We illustrate this for shift interventions in an SCM with additive Gaussian noises, where an intervention changes the mean of the exogenous noise variable.

For single-node intervention $I$, let $a^I$ denote the corresponding changes in the mean of the exogenous noise variables, i.e.,

$$a_i^I = \begin{cases} \mathbb{E}(\epsilon_i^I) - \mathbb{E}(\epsilon_i), & i \in T(I), \\ 0, & i \notin T(I). \end{cases}$$

We encode it as $\hat{I}$ with $T(\hat{I})$ containing one element and $\hat{\mathbf{a}}^{\hat{I}}$ being a one-hot vector, where

$$\hat{a}_i^{\hat{I}} = \begin{cases} \hat{a}_i, & i \in T(\hat{I}), \\ 0, & i \notin T(\hat{I}). \end{cases}$$

We extend this notation for $I$ with potentially multiple intervention targets (i.e., sets $I, \hat{I}$ that contain multiple elements) where $\mathbf{a}^I, \hat{\mathbf{a}}^{\hat{I}}$ can be a multi-hot vector.

In the shift intervention case, from Theorem 3, we know that the encoded $\hat{\mathbf{a}}^{\hat{I}_1}, ..., \hat{\mathbf{a}}^{\hat{I}_K}$ satisfy $\hat{\mathbf{a}}^{\hat{I}_k} = M(\mathbf{a}^{I_k})$ in the limit of infinite data, where $M$ is a linear operation with $M(\mathbf{a})_i = \Upsilon_{\tau(i),i} a_{\tau(i)}$. Thus for single-node interventions $I_{t(1)}, ..., I_{t(k)}$ amongst $I_1, ..., I_K$, the multi-node intervention $\mathcal{I} = I_{t(1)} \cup ... \cup I_{t(k)}$[13] corresponds multi-hot vector $\mathbf{a}^{\mathcal{I}}$ that satisfies $M(\mathbf{a}^{\mathcal{I}}) = M(\mathbf{a}^{I_{t(1)}} + ... +$

---

[13]Note that we allow overlapping intervention targets among $I_{t(1)}, ..., I_{t(k)}$, where $I_{t(1)} \cup ... \cup I_{t(k)}$ adds up all the shift values for intervention target $i$.

$\mathbf{a}^{I_{t(k)}}) = \hat{\mathbf{a}}^{\hat{I}_{t(1)}} + ... + \hat{\mathbf{a}}^{\hat{I}_{t(k)}}$. Thus if we encode $\mathcal{I}$ as $\hat{\mathbf{a}}^{\hat{\mathcal{I}}} := \hat{\mathbf{a}}^{\hat{I}_{t(1)}} + ... + \hat{\mathbf{a}}^{\hat{I}_{t(k)}}$, we can also generate $\hat{X}^{\hat{\mathcal{I}}}$ from the ground-truth distribution of $X = f(U)$ where $U \sim \mathbb{P}_U^{\mathcal{I}}(U)$ following the encoding-decoding process of Fig. 4.

# F  Discrepancy-based VAE Implementation Details

We summarize our hyperparameters in Table 2. Below, we describe where they are used in more detail. We use a linear structural equation with shift interventions. In practice, due to the nonlinear encoding from the latent $U$ to observed $X$, not much expressive power is lost. Code for our method is at https://github.com/uhlerlab/discrepancy_vae.

| Loss function | |
|---:|:---:|
| Kernel width (MMD) | 200 |
| Number of kernels (MMD) | 10 |
| $\lambda$ | 0.1 |
| $\beta_{\max}$ | 1 |
| $\alpha_{\max}$ | 1 |
| **Training** | |
| $t_{\max}$ | 100 |
| Learning rate | 0.001 |
| Batch size | 32 |

Table 2: Hyper-Parameters

**VAE Parameterization.** As is standard with VAEs, our encoder and decoder are parameterized as neural networks, and the exogenous variables are described via the reparameterization trick. We use a standard isotropic normal prior for $p(Z)$. To encode interventions, the function $T_\phi(\cdot)$ is parameterized as a fully connected neural network, where for differentiable training $T_\phi(C)$ is encoded as a one-hot vector via a softmax function, i.e., $T_\phi(C)_i = \exp(tT'_\phi(C)_i)/\sum_{j=1}^p \exp(tT'_\phi(C)_j)$ for some fully connected $T'_\phi$ and temperature $t > 0$. During training, we adopt an annealing temperature for $t$. In particular, $t = 1$ until half of the epochs elapse, and $t$ is linearly increased to $t_{\max}$ over the remaining epochs. At test time, the temperature of the softmax is set to a large value, recovering a close-to-true one-hot encoding.

**Loss Functions.** We use a mixture of MMD discrepancies, each with a Gaussian kernel with widths that are dyadically spaced [18]. This helps prevent numerical issues and vanishing gradient issues in training. The coefficient $\alpha$ of the discrepancy loss term $\mathcal{L}_{\theta,\phi}^{\text{discrep}}$ is given the following schedule: $\alpha = 0$ for the first 5 epochs, then $\alpha$ is linearly increased to $\alpha_{\max}$ until half of the epochs elapse, at which point it remains at $\alpha_{\max}$ for the rest of training. Similarly, the coefficient $\beta$ of the KL regularization term is given the following schedule: $\beta = 0$ for the first 10 epochs, then $\beta$ is linearly increased to $\beta_{\max}$ until half of the epochs elapse, at which point it remains at $\beta_{\max}$ for the rest of training.

**Optimization.** We train using the Adam optimizer, with the default parameters from PyTorch and a learning rate of 0.001.

**Biological Data.** For the experiments described in Section 6, the encoder $q_\phi$ was implemented as a 2-layer fully connected network with leaky ReLU activations and 128 hidden units. The intervention encoder $T_\phi$ uses 128 hidden units. To account for interventions with less samples, we use a batch size of 32. We train for 100 epochs in total, which takes less than 45 minutes on a single GPU.

# G  Extended Results on Biological Dataset

In this section, we provide additional evaluations of the experiments on the Perturb-seq dataset. The computation of RMSE are computed for individual interventional distributions. The computation of $R^2$ (we capped the minimum by 0 to avoid overflow) records the coefficient of determination by regressing the mean of the generated samples on the ground-truth distribution mean.

## G.1 Single-node interventions

Figure 9 shows the same visualization as Figure 5 in the main text for the remaining $11 = 14 - 3$ single target-gene interventions with more than 800 cells. Figure 10 presents this side-by-side for the training samples. For the entire 105 single interventions, we visualize for each individual intervention the empirical MMD between the generated populations and ground-truth populations in Figure 11, where the bars record the MMD in different batches.

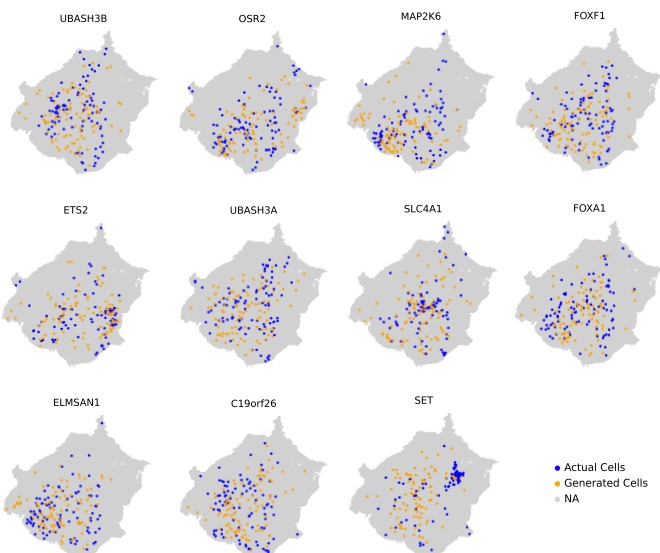

Figure 9: **For single-node interventions, the distribution of generated test samples visually mirrors the distribution of the actual samples.** A UMAP visualization of 11 single target interventions shows that the generated and the actual distributions closely match.

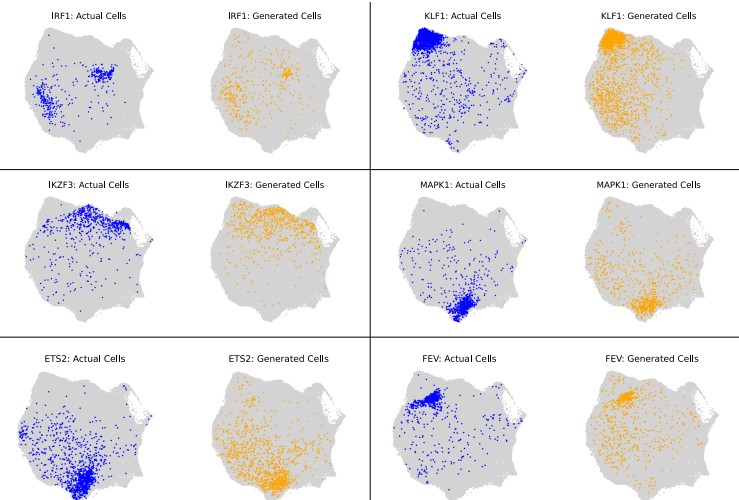

Figure 10: **For single-node interventions, the distribution of generated training samples visually mirrors the distribution of the actual samples.** As with the test samples, the distributions of the generated training samples closely match the actual distributions.

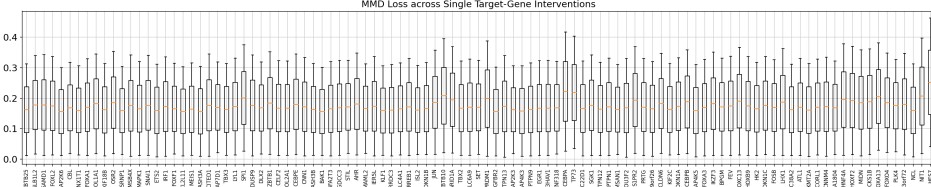

Figure 11: **For single-node interventions, the distribution of generated training samples quantitatively mirrors the distribution of the actual samples.** The figure shows the empirical MMD, defined in Appendix C.1, between the generated populations and ground-truth populations for 105 single target-node interventions.

## G.2 Double-node interventions

We plot the generated samples for 11 random double target-gene interventions in Figure 12. In Figure 13, we highlight two interventions for which the generated samples differ from the actual samples. The plots for all 112 interventions are provided at `https://github.com/uhlerlab/discrepancy_vae`.

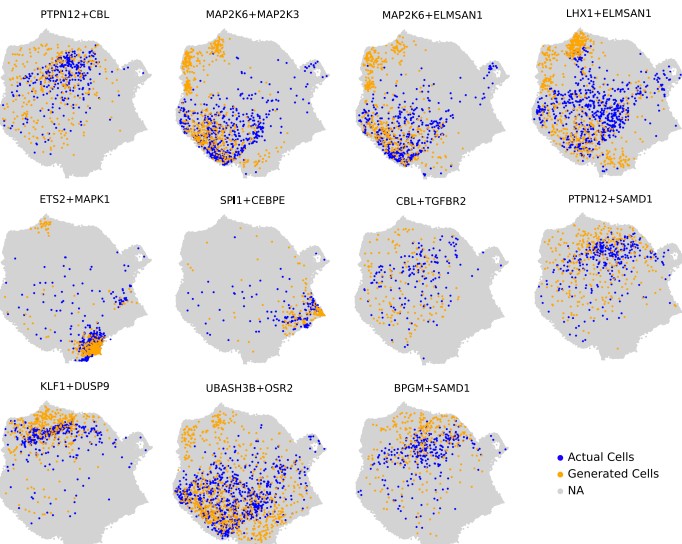

Figure 12: **UMAP visualization for a random sampling of double-node interventions.** Compared to single-node interventions, the generated samples of the double-node interventions match only for certain pairs.

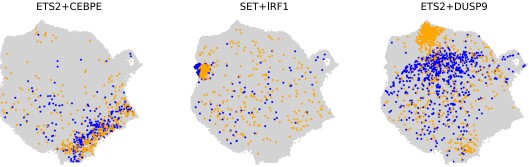

Figure 13: **For some double-node interventions, the generated samples match the actual samples, and for some combinations they do not.** The model accurately predicts the effect of the combinations ETS2+CEBPE and SET+IRF1, but does not accurately predict the effect of ETS2+DUSP9.

The MMD losses for all 112 interventions are summarized in Figure 14. Similar to Figure 6 in the main text, Figure 15 shows the distribution of RMSE and $R^2$ of the 112 interventions.

We remark here that this task has also been studied in previous works (e.g., [39, 8, 68, 49]) with different setups. Formally benchmarking the empirical results under a unified setting would be of interest in future works.

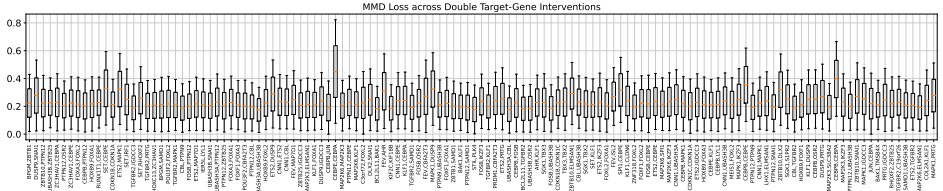

Figure 14: **For some double-node interventions, the distribution of generated samples quantitatively mirrors the distribution of the actual samples.** The figure shows the empirical MMD, defined in Appendix C.1, between the generated populations and ground-truth populations for 105 single target-node interventions.

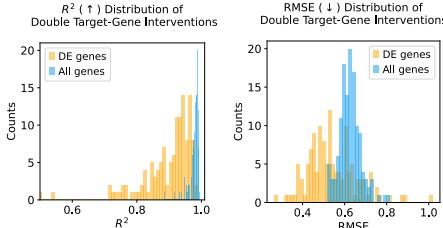

Figure 15: **Our model accurately predicts the effect of many double-node interventions.** 'All genes' indicates measurements using the entire 5000-dimensional vectors; 'DE genes' indicates measurements using the 20-dimensional vectors for the top 20 most differentially expressed genes.

### G.3 Structure Learning

In Figure 16, we show the learned latent structure between gene programs, along with descriptions of each gene program.

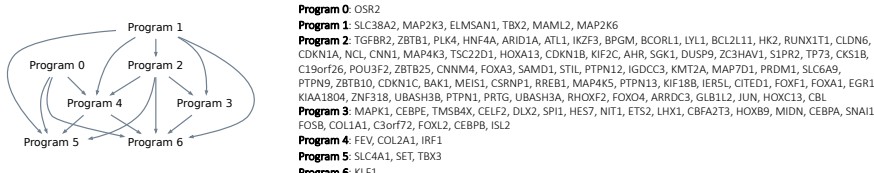

Figure 16: Regulatory relationships between programs learned in $\mathcal{G}$ and full list of genes in each program.

## H Extended Experiments

In this section, we provide additional experimental results. First, we perform ablation studies of different components of the proposed architecture on biological data. Then, we provide a simple simulation study to examine the performance of the framework on different tasks.

### H.1 Ablation Studies

For the ablation studies of different components, we compared the performance of our final model (depicted in Figure 4) against three alternative versions. All models are trained with the same setting (data split, schedule, learning rate, etc). In particular, we compared against

- **Models without the discrepancy loss.** These models learn the distributions similar to conditional VAE [51], where both an interventional sample and its interventional label are fed in to learn the exogenous $Z$. Then inside the latent space, we use the same causal layer as our model to generate a virtual sample. During inference, we can generate interventional samples via two approaches. One is sampling the exogenous $Z$ from $p(Z)$ and decoding. The other is sampling an observational sample, obtaining its exogenous $Z$ using the encoder

then decoding. These two approaches correspond to the second and third rows of Table 3 respectively.

- **A model without the causal layer.** This model uses a similar workflow as our final model in Figure 4, where we do not use a causal-based decoder but a simple MLP decoder. This corresponds to the fourth row of Table 3.

We note that the encoder, decoder, DSCM, and intervention encoder are needed to learn distributions and the latent causal graph from this setting where observational and interventional data are present.

For the metrics, we report both MMD and $R^2$ in Table 3. However, MMD is more meaningful as we are assessing the quality of generating a distribution. We observe that models without discrepancy perform much worse due to mode collapses, whereas the sampling approach using observational data performs slightly better. Our final model works the best in general; however on the MMD for double-node interventions, the version without a causal layer seems to work slightly better. This is potentially because some double-node interventions that act non-additively can be captured better without imposing the structure.

| Method | MMD (single) | $R^2$ (single) | MMD (double) | $R^2$ (double) |
|---|---|---|---|---|
| ours | **0.324±0.007** | **0.986±0.001** | 0.432±0.006 | **0.978±0.001** |
| ours w/o discrepancy | 2.966±0.054 | 0.984±0.003 | 3.358±0.031 | 0.972±0.002 |
| ours w/o discrepancy (obs) | 2.965±0.054 | 0.984±0.002 | 3.355±0.030 | 0.972±0.002 |
| ours w/o causal layer | 0.348±0.009 | 0.982±0.002 | **0.427±0.006** | **0.978±0.002** |

Table 3: **Ablation studies.** We report testing metrics and their standard error on the biological datasets. The results on single-node interventions are computed over 14 interventions. The results on double-node interventions are computed over all 112 interventions.

## H.2 Simulation

For the simulation study, as a proof-of-concept, we tested on a simple 5-node graph, where we generate 2048 samples in each of the 5 interventional datasets. We map this to a 10-dimensional observation space, where we pad zeros to the additional dimensions. This ensures clear visualization of the generated samples in Figure 17, where we compare the zero-shot learned double-node interventional samples against ground truth. In Table 4, we report the quantitative metrics. In addition to the MMD on left-out single and double-node interventions, we also report the training MMD and Structural Hamming Distance (SHD) of the learned graph.

Due to the combinatorial nature of learning a DAG and the small sample sizes in this setting, we observe that the learned intervention targets can be quite sensitive to initializations. Therefore during evaluation, we report the metrics while fixing the intervention targets to be of different transposition distances to the true targets. For single-node generations, different transposition distances return similar results, meaning that the model is expressive enough to learn these distributions, although we observe that the result with zero transposition distance is marginally better. This also holds during training, which can potentially be used as model selection to overcome the initialization issue. For double-node extrapolation, the result with zero transposition distance shows a larger benefit, as expected from our theory.

| Transposition Distance | MMD (training) | MMD (single) | MMD (double) | SHD |
|---|---|---|---|---|
| 0 | 0.030±0.007 | 0.047±0.008 | 0.041±0.004 | 2 |
| 1 | 0.057±0.028 | 0.058±0.030 | 0.181±0.048 | 6 |
| 10 | 0.042±0.007 | 0.041±0.009 | 0.119±0.023 | 11 |

Table 4: **A simple simulation study.** On a 5-node DAG, we test the model performance with varying transposition distances of the identified intervention targets. For sample generations, we report MMD and its standard error. The training metric is evaluated on all single-node interventions, where the third and forth rows are evaluated based on held-out samples of single and double-node interventions.

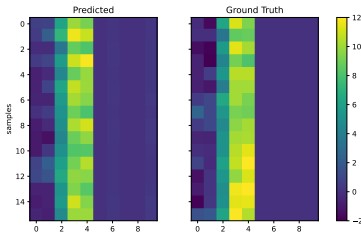

Figure 17: **An illustration of double-node intervention extrapolation in simulation.** We visualize 16 samples of the double-node intervention on nodes $2, 3$. The generated samples are shown on the left, where the ground-truth samples are shown on the right.

# I  Extended Discussion

## I.1  Limitations and Future Work

This paper opens up several direction for future theoretical and empirical work, which we now discuss.

**Theoretical Perspective.** We have focused on the setting where a single-node intervention on each latent node is available, similar to prior works on causal disentanglement [3, 53]. However, we highlight three issues in this setup and discuss potential remedies. First, by assuming access to data from intervening on every single latent node, we inherently possess partial knowledge of all the latent variables, even though we are unaware of their specific values or whether multiple interventions act on the same variable. The setups that do not assume interventions but the existence of anchored observed variables (i.e., variables with only one latent parent) [19, 9, 64, 65] face the same issue. This assumption can be unsatisfying in the context of causal representation learning, where the causal variables are assumed to be entirely unknown. Second, it may be impossible to intervene on all latent causal variables, especially in scenarios involving latent confounding. For instance, in climate research, it might be impossible to intervene on a variable like the precipitation level in a particular region. Finally, the assumption of single-node interventions can be overly optimistic in many applications. For example, in the case of chemical perturbations on cells, it is known that drugs often target multiple variables.

Nevertheless, the results obtained in the current setup can serve as a foundation and stepping stone towards the ultimate goal of general causal representation learning. On one hand, our analysis showed what can be learned from each intervention. This is helpful when considering cases where only a subset of the latent causal variables can be intervened on. On the other hand, the key techniques employed in our proofs can be extended to the multi-node setting. Specifically, in the latent space, one should expect only the marginals of variables downstream of a multi-node intervention to change.

Moreover, we have primarily focused on the infinite data regime for analyzing identifiability. Considering the expensive nature of obtaining interventional samples in practice, there is ample room for further investigation concerning sample complexity. Aside from the feasibility of identifiability, many applications are concerned with specific downstream tasks. Full identification of the underlying causal representations provides a comprehensive understanding of the system and would be beneficial for multiple downstream tasks. However, in certain cases, full identification may be unnecessary or inefficient for a particular task. Therefore, it is of interest to develop task-specific identifiability criteria for causal representation learning.

**Empirical Perspective.** We make two remarks on the VAE framework proposed in this work. First, as shown in our experiments in Section 6, our proposed framework can still be applied in settings with multi-node interventions and fewer single-node interventions. For instance, one can model multi-node interventions by reducing the temperature in the softmax layer. Second, due to the permutation symmetry of CD-equivalence, we impose an upper-triangular structure on the adjacency matrix in the deep SCM and learn the intervention targets. Alternatively, when there is exactly one intervention available for each latent node, one can instead prefix the intervention targets and learn the adjacency matrix. Specifically, we can set the intervention targets of $I_1, ..., I_p$ to be

a random permutation of $[p]$. Subsequently, the adjacency matrix can be learned for example via the nontears penalty [70] to enforce acyclicity. However, both methods inherit the combinatorial nature of learning a DAG, and therefore their performance may require large sample sizes and can be sensitive to initialization [27]. Consequently, endeavors to improve the optimization process and robustness of such models would be valuable.

## I.2  Discussion of Contemporaneous Works.

This work is concurrent with a number of other works in interventional causal representation learning. Unless otherwise noted, all of these works consider single-node interventions, as we do in this paper. Most similar to our setting is [59], which studies identifiability of nonparametric latent SCMs under linear mixing. They consider the case where exactly one intervention per latent node is available, which is an easier setting as we discussed in Section 2. In that setting, they provide a characterization of the learned causal variables. On the other hand, [7] studies identifiability of a linear latent SCM under nonparametric mixing. They also consider both hard and soft interventions, but in the form of linear SCM with additive Gaussian noises. Three concurrent works [25, 60, 35] consider *both* nonparametric SCMs and nonparametric mixing functions: [60] prove identifiability for the case of $p = 2$ latent variables when there is one intervention per latent variable. They provide an extension to arbitrary $p$ for settings where there are paired interventions on each latent variable. Meanwhile, [25] consider arbitrary $p$, without paired interventions. However, they use only conditional independence statements over the observed variables $X$ to recover the latent causal graph. As a result, their identifiability guarantees place restrictions on the latent causal graph, unlike the other works discussed here. The third work [35] studies the *Causal Component Analysis* problem, where the latent causal graph is assumed to be known. Finally, we note that other concurrent works study causal representation learning without interventional data [40, 32] or with vector-valued contexts instead of interventions [31].

