# OpenReview forum: "Identifiability Guarantees for Causal Disentanglement from Soft Interventions"
_NeurIPS.cc/2023/Conference — NeurIPS 2023 poster_

### Official Review · Reviewer_8AvV · 2023-06-23

**Soundness:** 3 good
**Presentation:** 4 excellent
**Contribution:** 2 fair
**Rating:** 6
**Confidence:** 3

**Summary:**

This paper focuses on the problem of causal discovery with unobserved causal variables. The authors show that under necessary assumptions, the latent causal model can be recovered by iteratively identifying the interventions that target the source nodes and removing these source nodes. Based on this finding, the authors propose a VAE-based generative model for recovering the causal mechanisms from observational and interventional data. The model shows promising performance on a gene mutation dataset in terms of capturing the interventional distribution and learning the causal DAG between target genes, which demonstrates the effectiveness of the proposed method.

**Strengths:**

• The authors clearly list the steps of identifying the ancestral relationships of nodes and direct edges in a causal DAG with soft interventions and give concrete examples to help readers better understand the assumptions.

• The presented experimental results show the proposed model’s capability of learning the causal mechanism in the training data as well as generalizing to unseen interventional data.

• The paper is organized and presented in good shape.


**Weaknesses:**

• It is better to briefly discuss the meaning of interventional faithfulness in previous studies. Does it simply mean “all causal variables are observed”?

• I think encourage the authors to discuss the validity of assumptions 2 and 3, especially assumption 2, in real-world scenarios as they claim that these are stronger versions of faithfulness assumptions.

Minor comments:

• In the equation on line 278, should the subscripts be $\hat{I} \notin \mathcal{I}$ and $\hat{I} \in \mathcal{I}$ instead of $I \notin \mathcal{I}$ and $I \in \mathcal{I}$?

• What does $s_i(\cdot)$ represent on line 297, the causal mechanism corresponding to the $i^{th}$ node in the causal DAG?

• The fonts in Figure 4 are too small. It might be better to change the orientation of the diagram (e.g. using a left-right flow instead of a top-down flow) and make the fonts larger.


**Questions:**

• Why do the authors choose MMD over Wasserstein distance for the distance measure between the generated and true interventional distribution?

• Can the proposed method handle the case where there exist **imbalanced interventions**? For example, there is only one sample in each of $\mathcal{D}^1, …, \mathcal{D}^{K-1}$ and 1000 samples in $\mathcal{D}^K$. I expect this situation to be possible in biological data since some genetic mutations can be extremely rare while some are frequently observed.

• Besides biological settings, can authors think of other scenarios where the proposed model can be applied?


**Limitations:**

The limitations are appropriately discussed. I’m not aware of any direct negative societal impacts of this work.

---

> ### Author Rebuttal · Authors · 2023-08-09
>
> Thank you for the encouraging comments. We appreciate that you think the identifiability proof is well laid out and the experiment shows promising results. We would like to address your remaining questions as follows:
>
> > **"It is better to briefly discuss the meaning of interventional faithfulness in previous studies. Does it simply mean “all causal variables are observed”?”**
>
> Prior interventional faithfulness assumptions [8-10] vary by a few technicalities; but they all assume that all causal variables are observed (causal sufficiency), and, more importantly, that intervening on a node will always change the marginal of its descendants. In particular, [8] (Definition 2, called “influentiality”) only made this assumption and showed that the causal graph is identifiable up to its transitive closure by detecting marginal changes. [8] showed that their algorithm consistently identifies the full causal graph by assuming additionally that intervening on a node changes the conditional distribution of its direct children giving its neighbors (details can be found in Assumption 4.5 of [9]).  A similar notion was also introduced in [10] where they made further assumptions regarding changes in the conditional distributions.
>
> We introduced this briefly between line 182-183. We will add this discussion to Appendix B.1 before presenting our faithfulness results and a pointer to this discussion after line 183.
>
> > **”I encourage the authors to discuss the validity of assumptions 2 and 3, especially assumption 2, in real-world scenarios as they claim that these are stronger versions of faithfulness assumptions.”**
>
> We appreciate this suggestion by the reviewer. Appendix B.1 discusses how these faithfulness assumptions can be satisfied.
>
> For Assumption 2, we extended Example 2 in the main text by providing a wider class of nonlinear SCMs that satisfies it, as demonstrated in Example 5. Then in Lemma 3, we showed in general how Assumption 2 can be satisfied by checking the moments of a finite number of variables. This boils Assumption 2 down to satisfying a finite set of inequalities. This is similar to the interventional faithfulness assumptions, but stronger, since the set of inequalities required pose more constraints than the inequalities required by interventional faithfulness. In both cases, the set of inequalities required ensure that any variable that remains unchanged under an intervention is not due to coincidence but rather the causal structure.
>
> As for Assumption 3, reductions can be made by assuming structural forms. We provided a brief discussion of how it can be satisfied in Lemma 4, where we showed it can be satisfied in a tree graph that satisfies Assumption 2. This is a stronger version of the faithfulness assumption in causal models [4], which ensures that any independence in the data arises not from coincidence but rather from the causal structure.
>
> > **”... should the subscripts be $\hat{I}\in\mathcal{I}$ and $\hat{I}\in\mathcal{I}$ instead of $I\in\mathcal{I}$ and $I\in\mathcal{I}$?”**
>
> Thank you for pointing this out. It should be $\hat{I}\in\mathcal{I}$ and $\hat{I}\notin \mathcal{I}$. We will revise it accordingly.
>
> > **”What does $s_i(\cdot)$ represent... the causal mechanism corresponding to the $i$th node...?"**
>
> Yes. $s_i(\cdot)$ corresponds to the causal mechanism that generates $U_i$ from its parents $U_{pa_{\mathcal{G}}(i)}$ and the exogenous noise $Z_i$. We will clarify this by adding a footnote stating its definition.
>
> > **”The fonts in Figure 4 are too small...”**
>
> Thank you for this suggestion! We were able to make the fonts larger by changing the orientation of the diagram. This can be found in the PDF attached to the general response.
>
> > **”Why do the authors choose MMD over Wasserstein distance for the distance...?”**
>
> We chose to use MMD over Wasserstein distance mainly because the kernel mean embedding used by MMD makes it efficiently computable, unlike the Wasserstein distance. In addition, MMD has several desirable properties. For instance, it can be estimated well with empirical samples, where the expected square error does not grow with the dimension. Further, when using a characteristic kernel (e.g., the Gaussian kernel that we used), MMD is a “strong metric”, i.e. the MMD between two distributions is equal to zero if and only they are equal almost everywhere.
>
> > **”Can the proposed method handle the case where there exist imbalanced interventions?...”**
>
> In the case of biological data, the interventions are imbalanced, but not very severely. Each intervention out of the 105 interventions comprises 50 to 2,000 cells. The proposed method seems to fit well to the interventions with smaller sample sizes. However, in case of severe imbalance, e.g. with just one sample in some interventions, we cannot estimate the distribution distance well.
>
> > **”Besides biological settings, can authors think of other scenarios where the proposed model can be applied?”**
>
> To apply the proposed model, we need data corresponding to various interventions to ensure theoretical guarantees. Therefore, in addition to observational data, we also need several interventional dataset to evaluate the algorithm. The biological domain provides a suitable setting, as it allows for large-scale interventional experiments. Besides the biological domain, we also see other potential applications that we hope to explore in future work. For example, the synthetic RL suite in [11] could be a suitable test case where interventions correspond to actions on the objects. However, the evaluation metric needs to be adapted in this case to maximizing a reward. Advanced image generative models such as [12] may also provide an opportunity to potentially generate interventional data by engineering prompts. However, it is debatable whether prompt engineering is more similar to intervening or conditioning.

---

> > ### Comment · Reviewer_8AvV · 2023-08-18
> > **Response to author rebuttal**
> >
> > Thank you for the comprehensive discussion regarding the faithfulness assumptions, which I believe will significantly enhance the theoretical robustness of this paper. The authors have adequately addressed most of my questions, and I'm inclined to keep the score unchanged.

---

> > > ### Author Response · Authors · 2023-08-18
> > > **Response**
> > >
> > > Thank you for the reply! We appreciate the comments which help improve the paper. We would be happy to answer any additional questions if there are any.

---

### Official Review · Reviewer_jsuJ · 2023-06-28

**Soundness:** 3 good
**Presentation:** 2 fair
**Contribution:** 2 fair
**Rating:** 5
**Confidence:** 4

**Summary:**

This paper studies the task of learning causal variables from high-dimensional observations. This is done under the assumption that single-node, soft interventions are available for all causal variables, as well as the observation function being a full row rank polynomial, e.g. linear. Based on prior work, the paper shows that this reduces to identifying causal variables from a linear combination, and determines challenges to identify the causal variables further in interventional distributions. The paper derives a result to identify causal variables up to linear combinations of its ancestors, and for identifying the causal graph among them. Finally, a practical algorithm, CausalDiscrepancyVAE, is derived and evaluated on genetic data with CRISPR interventions.

**Strengths:**

The paper is overall well written. The paper uses a clear and consistent notation throughout the paper. The setting is well introduced. Further, examples are provided to guide the understanding of the theory. The closest related works for the presented paper are clearly discussed and put into perspective with the goal of this paper.

The paper presents both theoretical and empirical results. The theoretical results are backed up by detailed proofs in the appendix. The proofs appear sound and the results are intuitive, but I have not checked all proofs in detail. The setting of unpaired observations with causal relations between variables is challenging. Hence, while not ideal, an assumption for single-node intervention on all variables is reasonable, considering the current stage of causal representation learning (CRL) in general. The empirical results are based on a newly proposed algorithm, that takes advantage of a variational autoencoder setup with additional losses for guidance.

Another strength of the paper is its application to real-world data, specifically gene expression data. Many CRL works currently work on purely synthetic benchmarks, which this paper tries to go a bit beyond. The results on the genetic data are encouraging, suggesting that future work in CRL can become potentially used in this application domain.

**Weaknesses:**

The main focus of the paper is its theoretical identifiability results. However, for that, the theoretical contribution of this work is quite limited. The main task of the identifiability theory is to determine the causal variables $U$ from a linear mixing $X=U\Lambda +b$. The problem with identifying $U$ from interventions is then discussed as the effect of interventions potentially being canceled out by linear combinations of the causal variables. Assumption 2 is then stated as simply ruling out this case. However, this assumption is done without much of simplification of this linear combination statement and requires checking all possible linear combinations in $P(U\_j+U\_SC^T)\neq P^I(U\_j+U\_SC^T)$. This is very likely difficult or impossible to verify in any real-world setting given that (a) the causal variables are not known, and (b) one needs to search over all $C$. Similarly, with the identifiability of the graph $G$, the assumption that it is the sparsest within its transitive closure is relatively restrictive and difficult to verify in real-world applications. Finally, it's unfortunate to see that the causal variables are then only identified up to linear combinations of its ancestors. The paper would be strengthened in discussing whether hard interventions can give stronger identifiability in these cases, which is likely the case as in [Lippe et al., 2023].

The proposed model architecture is quite involved and has several components (encoder, decoder, DSCM, MMD, intervention encoder) and hyperparameters ($\lambda, \alpha, \beta$, temperature of intervention encoder, MMD kernels) interacting. The tuning of the hyperparameters is not well discussed and it is only mentioned that delayed linear schedules are used for both $\alpha$ and $\beta$. It is unclear how sensitive the model is to those, and how much tuning is necessary. No ablation study on individual elements of the architecture have been performed, overall making it unclear what each component is contributing and whether parts are necessary. Overall, with such an involved setup without ablation studies, it is unlikely that the architecture will be adopted in future work.

Moreover, the theory has as integral assumption that the mixing function is limited to full row rank polynomial (Assumption 1). Nonetheless, the practical algorithm itself uses non-linear neural networks covering a much larger space of possible mixing functions. This appears to violate the assumptions and break its relation to the theoretical results. A discussion on how this affects the algorithm or intended datasets is largely missing.

In terms of empirical evaluation, it is great to see experiments on real-world data. However, no baselines like a simple VAE are compared to. This makes it difficult to put the results into perspective. A synthetic dataset on which the identifiability of the causal variables and baselines can be precisely compared is crucial to gain insights in the proposed architecture. Further, considering other modalities like images would greatly improve the range of the empirical evaluation. Finally, it is unclear if the experiments have been performed for multiple seeds, and how the trained models vary over the seeds.

Regarding the writing and presentation of the paper, it is overall well written. Still, it clearly targets researchers in the CRL community and is likely difficult to fully understand for researchers outside. In Section 2, 3 and 4.1, it is not always clear which parts are novel for this paper and which builds up on previous results. Overall, assumptions like almost-linear mixing function and single-node interventions on all variables should be clearly stated in the introduction. Additionally, the term 'unobserved causal variables' is confusing with latent variables in common causal discovery tasks, e.g. latent confounders. In this setup, the information of the causal variables is still observed through a mixing function, just the separation and individual variables are not identified. This should be clarified when using this term, e.g. in the introduction (line 52).

### Typos

- Line 306: 'to to'

### References

[Lippe et al., 2023] Lippe, P., Magliacane, S., Löwe, S., Asano, Y. M., Cohen, T., & Gavves, E. (2023). Causal representation learning for instantaneous and temporal effects in interactive systems. In The Eleventh International Conference on Learning Representations.

**Questions:**

### Review Summary

The paper presents an identifiability result for causal representation learning from soft interventions. It's strengths includes the detailed proofs for the presented theoretical results and its application to real-world datasets. However, the theoretical contribution is limited, considering its basis on prior works and extensions on them. The empirical evaluation is further insufficient, requiring more experiments to validate the claims and components of the architecture. Thus, I consider the current version of the paper below the acceptance threshold.

### Questions

Main aspects:
* How would you verify Assumption 2 and 3 in real-world settings?
* Do you have theoretical results on whether hard interventions strengthening your identifiability class or not?
* How does your practical algorithm relate to the theory given that the mixing functions have different classes?
* Do you have ablation studies on all components of your architecture?

Minor points:
* Example 1, Line 190: the interventional distribution of $U\_2$ is presented as identical to its observational distribution. I assume this is a typo where the interventional distribution should be $P(U\_2|U\_1)=\mathcal{N}(U\_1+1,1)$ instead? If not, can you explain how the interventional faithfulness is satisfied despite an intervention having no effect?
* Line 222: why do you consider here $p$ interventional settings ($I\in \{I\_1,...,I\_p\}$) instead of $K$ ($I\in \{I\_1,...,I\_K\}$) as in Sec. 2 before?

---

### Post-rebuttal update

The rebuttal addressed some of my concerns. Considering an improved presentation as discussed in the rebuttal, and the main contribution of this paper being theoretical, I raise my score to '5: Borderline Accept'.

**Limitations:**

Limitations of the theoretical setting have been shortly discussed in the conclusion.

---

> ### Author Rebuttal · Authors · 2023-08-09
>
>
> Thank you for the detailed review! We appreciate that reviewer found the application to real-world data encouraging. Due to character limits, we’d address the reviewer’s main points below.
>
> > **“Assumption 2  is done without much of simplification...” | “$G$ ... is the sparsest within its transitive closure” | "unfortunate that causal variables are only identified up to...” | "verify Assumption 2 and 3...”**
>
> There are many misunderstandings of the theoretical result that we would like to clarify as follows.
>
> Due to the points below, we strongly disagree with the assertion that the theoretical contribution is limited. Compared to other works on CRL, which have shown identifiability under the simpler cases of hard interventions, this paper considers the much more challenging case of soft interventions. Further, this paper provides a novel way of proving identifiability, which is clearly different from previous works. In addition, our detailed analysis show where non-identifiability arises and how this would affect real-world applications.
>
> **Assumption 2.** This assumption is not stated without much simplification of the linear combination statement. We require that the interventional effect is not canceled out *only for the interventional target itself and its immediate children*, which is a major reduction from requiring such non-cancellation for all the nodes downstream of the interventional target.
>
> More importantly, we did not require changed marginals for all possible linear combinations of $U$, but only linear combinations of nodes that are not downstream of the interventional target. Since the unknown linear mixing function can be arbitrary, we need to assume changed marginals for all linear combinations of the reduced set. This is required to avoid the existence of a mixing function such that the effect is canceled out which leads to a valid but wrong causal representation.
>
> These simplifications are necessary for us to show how it can be satisfied in Appendix B (Example 5 and Lemma 3). In Lemma 3, we reduce this assumption to a finite number of inequalities (instead of all possible linear combinations) that the causal model needs to satisfy, which is similar to previously assumed faithfulness (e.g. [8-10]).
>
> **Testability of faithfulness.** Similar to previous faithfulness, Assumption 2 is not a testable assumption, but rather a belief that any independence/unchanged relationships present in the data are due to the causal structure rather than coincidence. Although it seems like a hefty assumption, it really isn’t (see [4] for a discussion of faithfulness in prior work). Importantly, without some faithfulness or parsimony assumption, it is well-known that it is impossible to infer causal structure from data.
>
> **Graph sparsity.** We did not assume that the graph is sparsest within its transitive closure to guarantee its identifiability. See Theorem 2: no such assumption is made. Assumptions 1 and 2 suffice for identifying the graph up to transitive closure (Theorem 1). To also identify the direct edges, we need a type of “adjacency faithfulness” (Assumption 3), similar to traditional causal structure learning setups. We also show a special case of how Assumption 3 is satisfied if the ground-truth DAG is a tree graph.
>
> **Non-identifiability of variables.** The non-identifiability of causal variables themselves is an inherent result of this setup under soft interventions. This is not a limitation of our proof. Similar non-identifiability was also pointed out in [1,5]. Understanding it is crucial when we apply such methods in applications for interpreting the learned information correctly. Furthermore, even with this, we can still draw causal explanations and predict the effect of unseen combinations of interventions, as shown in Section 4.4.
>
> > **”theoretical results on hard interventions...”**
>
> In Appendix A.2, we discuss how the model is identifiable up to its finest level - CD equivalence class - with hard interventions. Since this is a result of prior work, we only present a short discussion.
>
> > **”tuning of the hyperparameters..." | "No ablation study...” | "synthetic dataset..." | "modalities like images..." | "multiple seeds...”**
>
> We provided the hyperparameters in detail in Appendix E (Table 1). We observe that the model is not sensitive to hyperparameters and not much tuning is necessary. The generation results reported are not performed with multiple seeds and should not vary significantly. However, the learned DAG structure varies across runs, likely due to its inherent combinatorial nature. For this, we run our algorithm with the identified number of modules multiple times and take the one with the least number of edges. We apologize for forgetting to add this sentence and will add it after line 382.
>
> In terms of empirical evaluation, we appreciate the reviewer’s suggestion. We performed additional ablation studies and a simple simulation study during the rebuttal. These results and details can be found in the general response and its attached PDF.
>
> For other modalities, we agree it would be interesting but this work is primarily a theoretical contribution. We see our algorithms and experiments as a way to show how our theory can be useful in applications. Further empirical evaluations should remain in future work.
>
> >**”practical algorithm relate to the theory... different classes?”**
>
> In Appendix D, we give consistency results if the algorithm uses parameterizations that are in line with Assumption 1. However, the assumptions are only required for providing theoretical guarantees. To apply the proposed algorithm in practice, it can easily adopt any neural network as the mixing function (e.g., MLPs that are easy to code up). However, such a method will not come with theoretical results such as consistency, unless proven in future works.
>
> ---
> All references are provided under the general response. We will make responses to minor points during the discussion period.

---

> > ### Author Response · Authors · 2023-08-10
> > **Response to minor points raised by the reviewer**
> >
> > We thank the reviewer again for the detailed review. Below, we provide responses to the minor points for completeness:
> >
> >
> > > **”In Section 2, 3 and 4.1 ... not always clear which parts are novel ... which builds on previous results.”**
> >
> > Section 2 only introduced the model setup with citations to where the definition comes from. Papers in CRL that study similar setups are discussed in the previous Section 1.1 on related work. In Section 3, we laid out the definition of equivalence classes (for our main results), which many previous papers implicitly used but may not explicitly define. In Section 4.1, we stated its title as “Preliminaries”, so this is mainly built on previous work. In addition, when introducing the Assumptions and Lemmas, we stated the references in the paragraph above. To make this more clear, we will add the citation to the assumption themselves.
> >
> > > **"almost-linear mixing function and single-node interventions... should be clearly stated..." | "'unobserved causal variables confusing...”**
> >
> > We thank the reviewer for this suggestion. We will revise the introduction by adding the following sentence to the last paragraph where we introduced the content of this paper: “We provide theoretical guarantees when the latent variables are observed through a class of (potentially non-linear) polynomial mixing function proposed by [1].” We will also clarify the term 'unobserved causal variables' in line 52.
> >
> >
> > > **”Example 1, Line 190: the interventional distribution of $U_2$ is presented as identical to its observational distribution. I assume this is a typo where the interventional distribution should be $P(U_2|U_1)=\mathcal{N}(U_1+1,1)$ instead?”**
> >
> > Yes, it should be $P(U_2|U_1)=\mathcal{N}(U_1+1,1)$ in line 190. Thank you for pointing this out; we will revise that accordingly.
> >
> > > **”Line 222: why do you consider here $p$ interventional settings ($I\in I_1,...,I_p$) instead of $K (I\in I_1,...,I_K)$ as in Sec. 2 before?”**
> >
> > This is for simplicity of illustrating the proof sketch of Theorem 1, as we stated in line 216. The formal proof in Appendix B.2 considers the general setting where there are $K\geq p$ interventions.

---

> > ### Comment · Reviewer_jsuJ · 2023-08-11
> > **Response to Rebuttal**
> >
> > Thank you for your response and clarifications. The rebuttal clarified the theoretical contribution of the paper better, with the suggestion to also take these clarifications into account for the final paper version. The empirical part of the paper remains a weakness, e.g. given its several components necessary. Still, since I acknowledge the contribution to be mainly theoretical, I raise my score by one.

---

> > > ### Author Response · Authors · 2023-08-11
> > > **Response**
> > >
> > > Thank you for the discussion and updates on reviews! We were glad to provide the clarifications on the theoretical contributions. We would also be happy to answer any additional questions if there are any.
> > >
> > > We want to make another comment on why several components are necessary for the proposed model. In this empirical framework, several components are necessary because of 1). the setup with multiple interventional datasets (discrepancy) and unknown interventions (intervention encoder) 2). needing to model the latent graph (DSCM).

---

### Official Review · Reviewer_xRy1 · 2023-07-07

**Soundness:** 2 fair
**Presentation:** 3 good
**Contribution:** 2 fair
**Rating:** 6
**Confidence:** 3

**Summary:**

This paper proves identifiability of causal disentaglement in latent space in the presence of interventional data. In more details, consider data X generated as X = f(U) where the distribution of U encodes a causal graph G (or a Bayesian network), identifiability is the question of whether f and U can be recovered uniquely. This is known to be not true in general but recent works have shown that this is possible under additional information or biases. This work assumes access to extra data of the form X' = f(U') is available where U' is a single-node soft intervention on U and under various other assumptions shows that identifiability holds.

Specifically, if we assume
1. f is a full-rank polynomial
2. the causal variables are linear interventional faithful (which means interventional effets don't cancel each other out),

then this work shows that the intervention targets and the transitive closure of the causal graph can be recovered up to an equivalence class. Additionally, with

3. an adjacency faithfulness condition,
it's shown that the underlying causal graph can also be recovered up to an equivalence class.

Identifiability is a statistically desirable notion that a unique latent variable model could have generated the data. A flurry of recent works have studied identifiability of latent causal graphs under interventions and this work continues this line of work for soft interventions under polynomial mixing. For the proof, the work by Ahuja et al. 2022 is first applied to directly recover the polynomial f up to a linear transformation. What follows next is the main contribution of this work, which is a sort of a peeling procedure. Basically, interventions that target the source nodes are recovered, after which we can recursively extract the transitive closure of the graph.

The authors also propose a variational autoencoder framework to estimate the latent causal representations
from interventional datasets in practice. The standard Evidence Lower Bound is modified to include a discrepancy term, that roughly measures how well the interventional samples have been modeled. The model is then trained via the standard reparameterization trick. Experiments on a biological dataset, with around 100 latent variables, are performed. The model is shown to predict certain expected causal relationships as well as good R^2 values for unseen double-node interventions. The target audience are people interested in identifiable representation learning. The theory seems good but the assumptions are a bit strong and the experiments should have been more comprehensive. It's possible that similar experimental techniques will adapt for other datasets however it may require more work, since assumptions maybe limiting (see weaknesses below).

#### References:
- [1] A. Seigal, C. Squires, and C. Uhler. Linear causal disentanglement via interventions. arXiv preprint arXiv:2211.16467, 2022
- [2] K. Ahuja, Y. Wang, D. Mahajan, and Y. Bengio. Interventional causal representation learning. arXiv preprint arXiv:2209.11924, 2022.
- [3] Varici, B., Acarturk, E., Shanmugam, K., Kumar, A., and Tajer, A. Score-based causal representation learning with interventions. arXiv preprint arXiv:2301.08230, 2023.

**Strengths:**

- As opposed to the do-interventions that Ahuja et al. 2022 consider, soft interventions addressed in this work are more realistic. Similarly, unpaired datasets and unknown intervention targets are also more natural.

- The illustrative examples shown in section 4.2 are useful to understand non-identifiability situations.

- The intervention targets are learned on-the-fly, as part of the autoencoding variational bayes formulation. This allows the work to generate virtual counterfactual samples. This is shown also in the experimental section.

**Weaknesses:**

- It's not mentioned anywhere in the introduction that the function f is a polynomial. Since this is a significant assumption both in theory and especially in practice (where we use neural networks), I find this very misleading to the readers.

- Assumption 2 as it reads seems a bit strong. The authors show that certain quadratic SEMs satisfy this assumption in Appendix B, but do the more simpler case of linear SEMs satisfy them? If not, it's a bit strange that the theorem does not cover the case of linear SEMs with linear f, that's covered in [1].

- Image and genomics datasets are used as motivation, but it's not clear why the crucial assumptions such as interventional faithfulness or polynomial f, should be true in those settings.

- Footnote 4 says that interventions are chosen to be shifts, why do the authors make this choice, when the theory seems to hold in general? This seems too restrictive and may limit potential applications.

- Additional experiments on synthetic datasets would have been more illustrative of the technique's performance, as well as allowing access to ablation studies which are crucial for such a loss formulation. In other words, it's not clear if the good performance on the biological dataset is because of the model (as we would like) or instead just an artefact of the specific dataset and training technique used.

**Questions:**

Some questions were raised above.

- I think it's fair to also cite
Varici, B., Acarturk, E., Shanmugam, K., Kumar, A., and Tajer, A. Score-based causal representation learning with interventions. arXiv preprint arXiv:2301.08230, 2023.

- A minor remark, the problem studied in this work has also been called "causal representation learning" in the literature, while "causal disentanglement" is used for the special case when the latent variables are jointly independent.

**Limitations:**

Limitations have been discussed.

---

> ### Author Rebuttal · Authors · 2023-08-09
>
> Thank you for the thorough review! We appreciate your recognition of our theoretical result, and the appreciation of our illustrative examples. We’ve ran addition experiments as suggested and we’d like to address your detailed comments below:
>
> > **”It's not mentioned anywhere in the introduction that the function f is a polynomial...”**
>
> We thank the reviewer for pointing this out. We will revise the introduction by adding the following sentence to the last paragraph where we introduced the content of this paper: “We provide theoretical guarantees, when the latent variables are observed through a class of (potentially non-linear) polynomial mixing functions proposed by [2].”
>
> In our original writeup, we introduced it in the main section on identifiability results (section 4) as this assumption is only needed for providing our theoretical guarantees. To apply the proposed algorithm in practice, one can use a neural network as the mixing function. However, this won’t come with nice theoretical results such as consistency. Contemporary work [3] considered the setting when the mixing function f is nonparametric. However, this work assumes that the underlying structural causal model (SCM) is linear additive Gaussian, whereas we work with general nonparametric SCMs. It remains an open problem to study the most general setting where both the mixing function and the SCM are nonparametric.
>
> > **”Assumption 2 as it reads seems a bit strong... do the more simpler case of linear SEMs satisfy them?”**
>
> The simpler linear SEMs with shift interventions will not satisfy Assumption 2. In fact, the transitive closure (and thus the latent causal graph) is not identifiable in this setting. Counter-example 1 in section 4.2 is a linear SEM with $U_2=U_1+\epsilon_2, U_1=\epsilon_1$, where the interventions are shift interventions. Intuitively, linear SEMs pose a harder challenge for identifiability because it is hard to distinguish the linear causal mechanism from the arbitrary linear mixing. In the case of nonlinear SEMs, some nonlinear causal mechanisms could be preserved after the linear mixing and used to identify the causal relationships.
>
> Note that this is only the case for soft interventions. When considering hard interventions (as is the main focus of [1]), identifiability can be achieved with weaker assumptions. The non-identifiability of linear SEMs with soft interventions was also recognized in [1] (Appendix J). The authors obtained identifiability of the transitive closure for a special class of soft interventions (Assumption 1(b) in their paper). This was achieved by fully utilizing the linearity of the underlying SEM, which would not be available in the general case.
>
> > **”Image and genomics datasets are used as motivation, but it's not clear why the crucial assumptions...”**
>
> We thank the reviewer for this comment. By assuming interventional faithfulness, we rule out the cases where the effect of an intervention is canceled out. Although it seems like a strong assumption, we feel it isn’t (see [4] for a discussion of the faithfulness assumption in previous literature). This is because it essentially assumes any variable that remains unchanged under an intervention is not due to coincidence but rather the causal structure. Without it, it is hard to infer causal structure from data, as one can only rule out causal structures that imply independence/unchanged relationships that are not present in the data. In terms of our polynomial assumption, see our response to the first comment of why it is required and ways to relax it. It is also worth noting that polynomial functions are universal approximators [2].
>
> We would also like to clarify that we motivate the scenario we study using image and genomic examples. However, to provide theoretical guarantees, certain assumptions need to be made. On real data, with limited samples, these can only hold true with approximations. We note this as an important future step in the discussions, which we hope to explore further in future work.
>
> > **”Footnote 4 says that interventions are chosen to be shifts, why do the authors make this choice...”**
>
> The theory indeed holds for general nonparametric interventions. In footnote 4, shifts are utilized as a convenient illustration of our algorithm, where the notation is simple. When applying the algorithm, the intervention model can be changed to fit the corresponding application. We appreciate this comment and will clarify the footnote to emphasize this flexibility.
>
> > **”Additional experiments on synthetic datasets... allowing access to ablation studies which are crucial for such a loss formulation...”**
>
> We thank the reviewer for this suggestion. Similar to [1,5], we view this work as primarily a theoretical contribution, where the proposed algorithms and experiments on biological data serve as a proof-of-concept of how theory can be helpful in real-world applications. However, we agree that more compressive experiments are helpful. Therefore, we performed additional ablation studies and a simple simulation study during the rebuttal phase. These results can be found in the PDF attached to the general response. The details of these experiments can also be found in the general response.
>
> > **”...fair to also cite Varici...”**
>
> Thank you for pointing this out! We will add the these contemporaneous works (including [3,5,6]) to Section 7.2.
>
> > **”... the problem studied in this work has also been called "causal representation learning" in the literature...”**
>
> Thank you for this remark. We adopted the term “causal disentanglement” mainly following this literature review [7] and some previous works [1]. This has the advantage of being more specific than “causal representation learning”, which also includes methods such as Invariant Risk Minimization (IRM) which do not completely learn all latent variables. We will clarify this in the related works section.
>
> ---
> All references are provided under the general response.

---

> > ### Comment · Reviewer_xRy1 · 2023-08-16
> > **Response to rebuttal**
> >
> > I thank the authors for the response and clarifications. The additional ablation studies are quite important and the authors should add them to the paper, along with improving the writing as per the reviewers' suggestions. I'm willing to increase my score.

---

> > > ### Author Response · Authors · 2023-08-17
> > > **Thank you so much for the discussion and update on the reviews**
> > >
> > > Thank you so much for the discussion and update on the reviews! We will make sure to include all the additional results and feedbacks from the reviews to the revision. Additionally, please feel free to reach out if there are any further questions.

---

### Official Review · Reviewer_YmWw · 2023-07-27

**Soundness:** 3 good
**Presentation:** 3 good
**Contribution:** 2 fair
**Rating:** 5
**Confidence:** 4

**Summary:**

This paper focuses on latent causal representation learning, with a specific focus on showcasing the identifiability of the causal structure among latent variables. The authors propose an approach based on assuming a full-row rank polynomial generator, soft interventions on each latent variable, a generalized version of faithfulness, and sparsity, so that the causal structure can be identified up to some equivalence.

**Strengths:**

Overall, the presentation is clear, and the study introduces a relatively novel contribution. While previous works in causal representation learning have primarily assumed marginally independent causal variables and hard interventions, this paper explores new dimensions by incorporating soft interventions.

**Weaknesses:**

However, there is one primary concern regarding the interpretation of soft interventions. In line 111, the paper mentions, "We focus on the scenario where we have at least one intervention per latent node." On the other hand, Assumption 2 implies, "Intervention I with target i," which appears to indicate that only one variable is being intervened upon each time. To avoid any confusion, I kindly request the authors to clarify this point in the discussion.


**Questions:**

In line 111, the paper mentions, "We focus on the scenario where we have at least one intervention per latent node." On the other hand, Assumption 2 implies, "Intervention I with target i," which appears to indicate that only one variable is being intervened upon each time. To avoid any confusion, I kindly request the authors to clarify this point in the discussion.

---

> ### Author Rebuttal · Authors · 2023-08-09
>
> Thank you for appreciating our results exploring soft interventions, and for your recognition of our contribution. We would like to clarify the following points in response to your comments:
>
> > **”In line 111, the paper mentions, "We focus on the scenario where we have at least one intervention per latent node." On the other hand, Assumption 2 implies, "Intervention I with target i," which appears to indicate that only one variable is being intervened upon each time.”**
>
> We will clarify the point about single-node interventions, and having one single-node intervention per target. This is best clarified with an example: if we have $d = 3$ latent nodes, then the intervention set $\\{ \\{ 1 \\}, \\{ 2 \\}, \\{ 3 \\} \\}$ is an intervention set consisting of single-node interventions, for which there is at least one intervention per latent node.
>
> Our assumption about single-node interventions is stated on line 91, and the necessity of at least one intervention per latent node is stated on line 111. There could be multiple single-node interventions targeting one node, as we discussed after line 111, e.g. the intervention set  $\\{ \\{ 1 \\}, \\{ 2 \\}, \\{ 3 \\}, \\{ 3 \\} \\}$ is also sufficient for identifiability.
>
> For further context, note that single-node interventions have been the main point of study in previous works (e.g., [1,2]) on causal disentanglement / causal representation learning, which considered the simpler cases of do- and perfect interventions. An extension to multi-node interventions is an important and challenging direction for future work, as we describe on line 398.
>
> ---
> All references are provided under the general response.

---

> > ### Comment · Reviewer_YmWw · 2023-08-16
> >
> >  Thank you for the explanation. Single-node interventions seem to be extremely hard to achieve. I would be excited to see multi-node interventions. Initially, I thought the current work can handle multi-node interventions as well.

---

> > > ### Author Response · Authors · 2023-08-17
> > > **Clarification of multi-node intervention results and contributions (in contrast to previous/contemporary works in this direction)**
> > >
> > > Thank you for the discussion! We do want to stress that some of our results hold for multi-node interventions — these are stated in Section 4.4 and Section 6. In the following, we further clarify this work’s results regarding multi- and single- node interventions (in contrast with previous/contemporary works). As the reviewer thought, the current work can handle multi-node interventions; we apologize for the confusion created in our first response. We hope this clarification elevates the reviewer’s opinion about our work.
> > >
> > > **Results on multi-node intervention.**
> > >
> > > In Theorem 3 of Section 4.4, we show that it is possible to extrapolate and generate samples from multi-node interventions. This theorem consists of our main theoretical results on multi-node interventions, which states that we can predict the effect of a multi-node intervention from its single-node components. This result is very helpful when working with real-world applications (which we describe in the next paragraph). To the best of our knowledge, no previous/contemporary works have shown similar results.
> > >
> > > For the empirical results on multi-node interventions, we demonstrate the extrapolation to predict multi-node interventional effect in our experiment section. In the biological application we considered, one important problem of interest is to predict the effect of combinatorial perturbations. In this context, one have access to perturbing several single genes (which can be modeled as single-node interventions) and the goal is to predict the effect of perturbing the combination of these genes (i.e., multi-node interventions). Our result in Theorem 3 and the proposed discrepancy-based VAE framework provide a solution.
> > >
> > > **Identifiability proof.**
> > >
> > > In our first response, we only describe the setting where we can show identifiability of the latent causal model (Theorem 1, 2). This result holds when we have at least one single-node interventions per latent node.
> > >
> > > The contribution of this works in terms of such identifiabilty proofs lies in considering _general soft_ interventions in _general_ structural causal models. This stands in contrast to previous/contemporary works [1,2,3], which primarily consider _hard_ interventions. It also generalizes previous/contemporary works [1,4], which consider _linear Gaussian_ structural causal models. In comparison, the setup we consider is the most general setup in terms of both intervention and structural equation model types. Two other contemporary works that consider this general setting are [5,6], where [5] consider having exactly one single-node interventions per latent node (which is an easier setting as we discussed in line 112), and [6] deals with the special case of p = 2 latent variables. All these works consider single-node interventions, as we do in this paper.
> > >
> > > Considering this paper works with the much more challenging case of soft interventions in general causal models, it provides a novel way of proving identifiability, which is very different from prior works. This comes with a new set of techniques that we think could be instrumental for future work.
> > >
> > >
> > > ---
> > > [1] Squires, C., Seigal, A., Bhate, S. S., and Uhler, C. (2023). Linear causal disentanglement via intervention.
> > >
> > > [2] Ahuja, K., Wang, Y., Mahajan, D., and Bengio, Y. (2022b). Interventional causal representation learning.
> > >
> > > [3] Jiang, Y. and Aragam, B. (2023). Learning nonparametric latent causal graphs with unknown intervention.
> > >
> > > [4] Buchholz, S., Rajendran, G., Rosenfeld, E., Aragam, B., Sch ̈olkopf, B., and Ravikumar, P. (2023). Learning linear causal representations from interventions under general nonlinear mixing.
> > >
> > > [5] Varici, B., Acarturk, E., Shanmugam, K., Kumar, A., and Tajer, A. (2023). Score-based causal representation learning with interventions.
> > >
> > > [6] von K ̈ugelgen, J., Besserve, M., Liang, W., Gresele, L., Keki ́c, A., Bareinboim, E., Blei, D. M., and Sch ̈olkopf, B. (2023). Nonparametric identifiability of causal representations from unknown intervention.

---

### Author Rebuttal · Authors · 2023-08-09

We thank all the reviewers for their insightful comments and suggestions!

---
In this general response, we attached a pdf of the additional figures and tables that we will add to the manuscript. To summarize the PDF, it includes:

- A modified Figure 4, which now has a larger font with the procedure to generate virtual counterfactual samples highlighted.
- Additional experiments (see below for details):
    - Table 1, which shows ablation studies of different components of the proposed architecture on biological data
    - Table 2 and Figure 1, which shows a simple simulation study.

In addition, we will also provide a discussion on contemporaneous work in discussion (including [3,5,6]).

---
For the ablation studies of different components, we compared the performance of our final model against three alternative versions. All models are trained with the same setting (data split, schedule, learning rate, etc). This can be found in Table 1 of the PDF under general response. In particular, we compared against

- **Models without the discrepancy loss.** These learn the distributions similar to the conditional VAE [13], where both an interventional sample and its interventional label are fed in to learn the exogenous $Z$. Then inside the latent, we use the same causal layer as our model to generate a virtual sample. During inference, we can generate interventional samples via two approaches. One is sampling the exogenous $Z$ from $\mathcal{N}(0,I)$ and decoding. The other is sampling an observational sample, obtaining its exogenous Z using the encoder then decoding. These two approaches correspond to the second and third rows respectively.
- **A model without the causal layer.** This model uses a similar workflow as our final model (illustrated in Figure 4 of our paper), where we do not use a causal-based decoder but a simple MLP decoder. This corresponds to the fourth row of the table.

We also note that the encoder, decoder, DSCM, and intervention encoder are needed to learn distributions and the latent causal graph from this setting where observational and interventional data are present. To make the architecture clearer for future use, we improved the presentation of Figure 4.

The metrics can be found in the PDF, where we report both MMD and $R^2$ (a widely adopted metric in biological applications). However, MMD is more meaningful as we are assessing the quality of generating a distribution. We observe that models without discrepancy perform much worse due to mode collapses, whereas the sampling approach using observational data (see the above bullet points for details) performs slightly better. Our final model works the best in general; however on the MMD for double-node interventions, the version without a causal layer seems to work slightly better. This is potentially because some double-node interventions that act non-additively can be captured better without imposing the structure.

For the simulation study, as a proof-of-concept, we tested on a simple 5-node graph, where we generate $2048$ samples in each of the 5 interventional datasets. We map this to a $10$-dimensional observation space, where we pad zeros to the additional dimensions. This ensures clear visualization of the generated samples in Figure 1, where we compare the zero-shot learned double-node interventional samples against ground truth. In Table 2, we report the quantitative metrics. In addition to the MMD on left-out single and double-node interventions, we also report the training MMD and Structural Hamming Distance (SHD) of the learned graph.

Due to the combinatorial nature of learning a DAG and the small sample sizes in this setting, we observe that the learned intervention targets can be quite sensitive to initializations. Therefore during evaluation, we report the metrics while fixing the intervention targets to be of different transposition distances to the true targets. For single-node generations, different transposition distances return similar results, meaning that the model is expressive enough to learn these distributions, although we observe the result with zero transposition distance is slightly better. This is also true during training, which can potentially be used as model selection to overcome the initialization issue. For double-node extrapolation, the result with zero transposition distance shows a larger benefit, as expected from our theory.

---
We provide the full list of references here:

[1] Squires, Chandler, et al. "Linear Causal Disentanglement via Interventions."

[2] Ahuja, Kartik, et al. "Interventional causal representation learning."

[3] Buchholz, Simon, et al. "Learning Linear Causal Representations from Interventions under General Nonlinear Mixing."

[4] Sobel, Michael E. "An introduction to causal inference."

[5] Varici, Burak, et al. "Score-based causal representation learning with interventions."

[6] Lippe, Phillip, et al. "Causal representation learning for instantaneous and temporal effects in interactive systems."

[7] Kaddour, Jean, et al. "Causal machine learning: A survey and open problems."

[8] Tian, Jin, and Judea Pearl. "Causal discovery from changes."

[9] Yang, Karren, et al."Characterizing and learning equivalence classes of causal DAGs under interventions."

[10] Jaber, Amin, et al. "Causal discovery from soft interventions with unknown targets: Characterization and learning."

[11] Ke, Nan Rosemary, et al. "Systematic evaluation of causal discovery in visual model based reinforcement learning."

[12] Ramesh, Aditya, et al. "Hierarchical text-conditional image generation with clip latents."

[13] Sohn, Kihyuk, et al. "Learning structured output representation using deep conditional generative models."

---

### Author Response · Authors · 2023-08-15
**Opportunity to engage in discussions and answer further questions**

Dear reviewers, we thank you again for taking the time to review our paper. We appreciate all the feedback and comments that help improve the work.

As the discussion end date approaches, we would love to have an opportunity to engage in discussions, and provide further clarifications if there are additional questions. Once again, thank you for the considerations!

---

### Decision · Program_Chairs · 2023-09-21

**Decision:**

Accept (poster)

**Comment:**

This paper continues a recent line of work on causal representation learning under interventions, which has been mostly studied in the special case of atomic, hard interventions, and is largely theoretical. The authors contribute a series of theoretical results to cover more general soft interventions, as well as a variational Bayes algorithm that can be used in practice. Although they consider polynomial decoders, the results in this paper are essentially linear since polynomial functions can be trivially reduced to linear models. There is an additional limitation in that some of the results actually rule out linear models (as pointed out in the discussion by the authors), which is surprising but opens the door for future investigations. In my view, the main contribution is the treatment of soft interventions.

Overall, there is a weak consensus amongst reviewers to accept this paper. Causal representation learning is a very difficult problem, and this work contributes notable theoretical results that improve the state of the art.